# Faster Rates for Private Adversarial Bandits

Hilal Asi [* 1]   Vinod Raman [* 2]   Kunal Talwar [* 1]

## Abstract

We design new differentially private algorithms for the problems of adversarial bandits and bandits with expert advice. For adversarial bandits, we give a simple and efficient conversion of any non-private bandit algorithm to a private bandit algorithm. Instantiating our conversion with existing non-private bandit algorithms gives a regret upper bound of $O\left(\frac{\sqrt{KT}}{\sqrt{\varepsilon}}\right)$, improving upon the existing upper bound $O\left(\frac{\sqrt{KT\log(KT)}}{\varepsilon}\right)$ for all $\varepsilon \leq 1$. In particular, our algorithms allow for sublinear expected regret even when $\varepsilon \leq \frac{1}{\sqrt{T}}$, establishing the first known separation between central and local differential privacy for this problem. For bandits with expert advice, we give the first differentially private algorithms, with expected regret $\quad O\left(\frac{\sqrt{NT}}{\sqrt{\varepsilon}}\right), O\left(\frac{\sqrt{KT\log(N)}\log(KT)}{\varepsilon}\right)$, and $\tilde{O}\left(\frac{N^{1/6}K^{1/2}T^{2/3}\log(NT)}{\varepsilon^{1/3}} + \frac{N^{1/2}\log(NT)}{\varepsilon}\right)$, where $K$ and $N$ are the number of actions and experts respectively. These rates allow us to get sublinear regret for different combinations of small and large $K, N$ and $\varepsilon$.

## 1. Introduction

In the adversarial bandit problem, a learner plays a sequential game against nature over $T \in \mathbb{N}$ rounds. In each round $t \in \{1, \ldots, T\}$, nature picks a loss function $\ell_t : [K] \to [0, 1]$, hidden to the learner. The learner, using the history of the game up to time point $t - 1$, selects a potentially random action $I_t \in \{1, \ldots, K\}$ and nature reveals only the loss $\ell_t(I_t)$ of the selected action. For any sequence of loss functions $\ell_1, \ldots, \ell_T$, the goal of the learner is to select a sequence of actions $I_1, \ldots, I_T$, while only observing

the loss of selected actions, such that its expected *regret*

$$\mathbb{E}\left[\sum_{t=1}^{T} \ell_t(I_t)\right] - \arg\min_{i \in [K]} \sum_{t=1}^{T} \ell_t(i)$$

is minimized, where the expectation is taken with respect to the randomness of the learner.

Bandit algorithms, and in particular adversarial bandit algorithms (Auer et al., 2002), have been of significant interest for over two decades (Bubeck et al., 2012) due to their applications to online advertising, medical trials, and recommendation systems. In many of these settings, one would like to publish the actions selected by bandit algorithms without leaking sensitive user information. For example, when predicting treatment options for patients with the goal of maximizing the number of cured patients, one may want to publish results about the best treatment without leaking sensitive patient medical history (Lu et al., 2021). In online advertising, a goal is to publish the recommended ads without leaking user preferences. In light of such privacy concerns, we study adversarial bandits under the constraint of differential privacy (Dwork, 2006). Surprisingly, unlike the stochastic setting (Azize & Basu, 2022), the price of privacy in adversarial bandits is not well understood. Existing work by Agarwal & Singh (2017) and Tossou & Dimitrakakis (2017) give $\varepsilon$-differentially private bandit algorithms with expected regret at most $O\left(\frac{\sqrt{KT\log(K)}}{\varepsilon}\right)$ [1]. However, their algorithms satisfy the stronger notion of local differential privacy and become vacuous for tasks with high privacy requirements, where one might take $\varepsilon < \frac{1}{\sqrt{T}}$. In fact, it was not known how large $\varepsilon$ needs to be in order to obtain sublinear expected worst-case regret.

**Main Contributions.** Motivated by this gap, we provide new, differentially private algorithms for adversarial bandits and bandits with expert advice with better trade-offs between privacy and regret. In the adversarial bandits setting, we provide a simple and efficient conversion of any non-private bandit algorithm into a private bandit algorithm. By instantiating this conversion with existing (non-private)

---

*Equal contribution [1]Apple [2]Department of Statistics, University of Michigan, Ann Arbor. Work done while interning at Apple. Correspondence to: Vinod Raman <vkraman@umich.edu>.

*Proceedings of the 42nd International Conference on Machine Learning*, Vancouver, Canada. PMLR 267, 2025. Copyright 2025 by the author(s).

---

[1]Tossou & Dimitrakakis (2017) also claim to give an algorithm with regret $\tilde{O}\left(\frac{T^{2/3}\sqrt{K\ln(K)}}{\varepsilon^{1/3}}\right)$, however, we are unable to verify its correctness. See Appendix G.1.

bandit algorithms, we get $\varepsilon$-differentially private bandit algorithms with expected regret at most $O\left(\frac{\sqrt{KT}}{\sqrt{\varepsilon}}\right)$, improving upon the best known upper bounds for *all* $\varepsilon \leq 1$. In particular, this result shows that sublinear regret is possible for any $\varepsilon \in \omega\left(\frac{1}{T}\right)$. As corollaries, we establish separations in the achievable regret bounds between:

(1) **Oblivious and Adaptive adversaries.** In particular, while we show that sublinear regret is possible for all choices of $\varepsilon \in \omega(\frac{1}{T})$ under an oblivious adversary, this is not the case for an adaptive adversary, where one cannot achieve sublinear regret if $\varepsilon \in o(\frac{1}{\sqrt{T}})$ (Asi et al., 2023).

(2) **Central and Local differential privacy.** While our results show that sublinear regret is possible if $\varepsilon \in o(\frac{1}{\sqrt{T}})$ under central differential privacy, it is well known that this is not the case for local differential privacy.

For bandits with expert advice (Auer et al., 2002), we give the first differentially private algorithms. In particular, we give three different $(\varepsilon, \delta)$-differentially private bandit algorithms, obtaining expected regret $O\left(\frac{\sqrt{NT}}{\sqrt{\varepsilon}}\right), O\left(\frac{\sqrt{KT\log(N)}\log(KT)}{\varepsilon}\right)$, and $\tilde{O}\left(\frac{N^{1/6}K^{1/2}T^{2/3}\log(NT)}{\varepsilon^{1/3}} + \frac{N^{1/2}\log(NT)}{\varepsilon}\right)$ respectively. These regret guarantees cover regimes with high-privacy requirements and regimes with a large number of experts $N$. In both settings, our techniques involve combining the Laplace mechanism with batched losses.

## 1.1. Related Works

**Adversarial Bandits and Bandits with Expert Advice.** We refer the reader to the excellent book by Bubeck et al. (2012) for a history of stochastic and adversarial bandits. The study of the adversarial bandit problem dates back at least to the seminal work of Auer et al. (2002), who show that a modification to the Multiplicative Weights Algorithm, known as EXP3, achieves worst-case expected regret $O\left(\sqrt{TK\log(K)}\right)$. Following this work, there has been an explosion of interest in designing better adversarial bandit algorithms, including, amongst others, the work by Audibert & Bubeck (2009), who establish that the minimax regret for adversarial bandits is $\Theta\left(\sqrt{TK}\right)$. More recently, there has been interest in unifying existing adversarial bandit algorithms through the lens of Follow-the-Regularized Leader (FTRL) and Follow-the-Perturbed-Leader (FTPL) (Abernethy et al., 2015). Surprisingly, while it was known since the work of Audibert & Bubeck (2009) that an FTRL-based approach can lead to minimax optimal regret bounds, it was only recently shown that this is also the case for FTPL-based bandit algorithms (Honda et al., 2023).

The first works for bandits with expert advice also date back at least to that of Auer et al. (2002), who propose EXP4 and bound its expected regret by $O\left(\sqrt{TK\log(N)}\right)$, where $N$ is the number of experts. When $N \geq K$, Seldin & Lugosi (2016) prove a lower bound of $\Omega\left(\sqrt{\frac{K}{\log(K)}T\log(N)}\right)$ on the expected regret, showing that EXP4 is already near optimal. As a result, EXP4 has become an important building block for related problems, like online multiclass classification (Daniely & Helbertal, 2013; Raman et al., 2024) and sleeping bandits (Kleinberg et al., 2010), among others.

**Private Online Learning.** Dwork et al. (2010a) initiated the study of differentially private online learning. Jain et al. (2012) extend these results to broad setting of online convex programming by using gradient-based algorithms to achieve differential privacy. Following this work, Guha Thakurta & Smith (2013) privatize the Follow-the-Approximate-Leader template to obtain sharper guarantees for online convex optimization. In the special case of learning with expert advice, Dwork et al. (2014) and Jain & Thakurta (2014) give private online learning algorithms with regret bounds of $O\left(\frac{\sqrt{T\log(N)}}{\varepsilon}\right)$. More recently, Agarwal & Singh (2017) design private algorithms for online linear optimization with regret bounds that scale like $O(\sqrt{T}) + O(\frac{1}{\varepsilon})$. In particular, for the setting of learning with expert advice, they show that it is possible to obtain a regret bound that scales like $O\left(\sqrt{T\log(N)} + \frac{N\log(N)\log^2 T}{\varepsilon}\right)$, improving upon the work by Dwork et al. (2014) and (Jain & Thakurta, 2014). For large $N$, this upper bound was further improved to $O\left(\sqrt{T\log(N)} + \frac{T^{1/3}\log(N)}{\varepsilon}\right)$ and $O\left(\sqrt{T\log(N)} + \frac{T^{1/3}\log(N)}{\varepsilon^{2/3}}\right)$ by Asi et al. (2023) and Asi et al. (2024) respectively in the oblivious setting.

**Private Bandits.** The majority of existing work on differentially private bandits focus on the stochastic setting (Mishra & Thakurta, 2015; Tossou & Dimitrakakis, 2016; Sajed & Sheffet, 2019; Hu et al., 2021; Azize & Basu, 2022), linear contextual bandits (Shariff & Sheffet, 2018; Neel & Roth, 2018), or adjacent notions of differential privacy (Zheng et al., 2020; Tenenbaum et al., 2021; Ren et al., 2020). To our knowledge, there are only three existing works that study differentially private *adversarial* bandits. The first is by Guha Thakurta & Smith (2013) who give an $(\varepsilon, \delta)$-differentially private bandit algorithm with expected regret $O\left(\frac{KT^{3/4}}{\varepsilon}\right)$. Finally, and in parallel, Agarwal & Singh (2017) and Tossou & Dimitrakakis (2017) improve the upper bound to $O\left(\frac{\sqrt{KT\log(K)}}{\varepsilon}\right)$. We note that the private algorithms given by Agarwal & Singh (2017) and Tossou & Dimitrakakis (2017) satisfy the even stronger notion of

*Table 1.* Summary of upper bounds with constant factors and dependencies on $\log \frac{1}{\delta}$ suppressed. The three rows for Bandits with Experts represent different algorithms with incomparable guarantees.

| | **Existing Work** | **Our Work** |
|---|---|---|
| **Adversarial Bandits** | $\frac{\sqrt{KT \log(KT)}}{\varepsilon}$ | $\frac{\sqrt{KT}}{\sqrt{\varepsilon}}$ |
| **Bandits with Experts** | NA | $\frac{\sqrt{NT}}{\sqrt{\varepsilon}}$ |
| **Bandits with Experts** | NA | $\frac{\sqrt{KT \log(N)} \log(KT)}{\varepsilon}$ |
| **Bandits with Experts** | NA | $\frac{N^{1/6} K^{1/2} T^{2/3} \log(NT)}{\varepsilon^{1/3}} + \frac{N^{1/2} \log(NT)}{\varepsilon}$ |

local differential privacy (Duchi et al., 2013).

## 2. Preliminaries

### 2.1. Notation

Let $K \in \mathbb{N}$ denote the number of actions and $\ell : [K] \mapsto [0, 1]$ denote an arbitrary loss function that maps an action to a bounded loss. For an abstract sequence $z_1, \ldots, z_n$, we abbreviate it as $z_{1:n}$ and $(z_s)_{s=1}^n$ interchangeably. For a measurable space $(\mathcal{X}, \sigma(\mathcal{X}))$, we let $\Pi(\mathcal{X})$ denote the set of all probability measures on $\mathcal{X}$. We let $\mathrm{Lap}(\lambda)$ denote the Laplace distribution with mean zero and scale $\lambda$ such that its probability density function is $f_\lambda(x) = \frac{1}{2\lambda} \exp\left(\frac{-|x|}{\lambda}\right)$. Finally, we let $[N] := \{1, \ldots, N\}$ for $N \in \mathbb{N}$.

### 2.2. The Adversarial Bandit Problem

In the adversarial bandit problem, a learner plays a sequential game against nature over $T \in \mathbb{N}$ rounds. In each round $t \in [T]$, the learner selects (potentially randomly) an action $I_t \in [K]$ and observes *only* its loss $\ell_t(I_t)$. The goal of the learner is to adaptively select actions $I_1, \ldots, I_T \in [K]$ such that its cumulative loss is close to the best possible cumulative loss of the best fixed action $i^\star \in [K]$ in hindsight. Crucially, we place no assumptions on the sequence of losses $\ell_1, \ldots, \ell_T$, and thus they may be chosen adversarially.

Before we quantify the performance metric of interest, we provide a formal definition of a bandit online learning algorithm. This definition will be useful for precisely formalizing the notion of privacy (Section 2.4) and describing our generic transformation of non-private bandit algorithms to private ones (Section 3).

**Definition 2.1** (Bandit Algorithm). A bandit algorithm is a deterministic map $\mathcal{A} : ([K] \times \mathbb{R})^\star \to \Pi([K])$ which, for every $t \in \mathbb{N}$, maps a history of actions and observed losses $(I_s, \ell_s(I_s))_{s=1}^{t-1} \in ([K] \times \mathbb{R})^{t-1}$ to a distribution $\mu_t \in \Pi([K])$. The learner then samples an action $I_t \sim \mu_t$.

We will slightly abuse notation by using $\mathcal{A}((I_s, \ell_s(I_s))_{s=1}^{t-1})$ to denote the random action $I_t$ drawn from $\mu_t$, the distribu-

tion that $\mathcal{A}$ outputs when run on $(I_s, \ell_s(I_s))_{s=1}^{t-1}$. In addition, we will sometimes use $\mathcal{H}_t := (I_s, \ell_s(I_s))_{s=1}^{t-1}$ to denote the history of selected actions and observed losses induced by running $\mathcal{A}$ up to, but not including, timepoint $t \in \mathbb{N}$. Note that $\mathcal{H}_t$ is a random variable and we may write the action selected by algorithm $\mathcal{A}$ on round $t \in \mathbb{N}$ as $\mathcal{A}(\mathcal{H}_t)$. It will also be helpful to think about $\mathcal{H}_t$ as the View of $\mathcal{A}$ as a result of its interaction with the adversary up to, but not including, timepoint $t$.

Given a bandit online learner $\mathcal{A}$, we define its *worst-case* expected regret as

$$\mathrm{R}_{\mathcal{A}}(T, K) = \sup_{\ell_{1:T}} \left( \mathbb{E}\left[\sum_{t=1}^T \ell_t(\mathcal{A}(\mathcal{H}_t))\right] - \inf_{i \in [K]} \sum_{t=1}^T \ell_t(i) \right),$$

where the expectation is taken only with respect to the randomness of the learner. Our goal is to design a bandit algorithm $\mathcal{A}$ such that $\mathrm{R}_{\mathcal{A}}(T, K) = o(T)$. Note that our definition of regret means that we are assuming an *oblivious* adversary, one that selects the entire sequence of losses $\ell_1, \ldots, \ell_T$ before the game begins. This assumption is in contrast to that of an adaptive adversary which, for every $t \in \mathbb{N}$, may select the loss $\ell_t$ based on $\mathcal{H}_t$. We leave quantifying the rates for private adversarial bandits under adaptive adversaries for future work. That said, we do note that the lower bounds for adaptive adversaries established in full-information setting by Asi et al. (2023) also carry over to the bandit feedback setting. Accordingly, Corollary 3.2 and Theorems 4 and 5 in Asi et al. (2023) show that the strong separation in the possible rates for oblivious and adaptive adversaries also holds under bandit feedback.

### 2.3. The Bandits with Expert Advice Problem

In adversarial bandits with expert advice (Auer et al., 2002), we distinguish between a set of experts $[N]$ and the set of available actions $[K]$. In each round $t \in [T]$, each expert $j \in [N]$ predicts a distribution $\mu_t^j \in \Pi([K])$. The learner uses these predictions to compute its own distribution $\hat{\mu}_t \in \Pi([K])$, after which it samples $I_t \sim \hat{\mu}_t$ and observes the loss $\ell_t(I_t)$. The goal of the learner is to compete against the best fixed expert in hindsight while observing bandit feedback. We need a new definition of a bandit with expert

advice algorithm to account for the fact that the learner has access to expert advice.

**Definition 2.2** (Bandits with Expert Advice Algorithm). A bandit with expert advice algorithm is a deterministic map $\mathcal{A} : ([K] \times \mathbb{R})^{\star} \times (\Pi([K])^N)^{\star} \to \Pi([K])$ which, for every $t \in \mathbb{N}$, maps the history of actions and observed losses $(I_s, \ell_s(I_s))_{s=1}^{t-1} \in ([K] \times \mathbb{R})^{t-1}$ as well the sequence of expert advice $\mu_{1:t}^{1:N} \in ((\Pi([K])^N)^t$ to a distribution $\hat{\mu}_t \in \Pi([K])$. The learner then samples an action action $I_t \sim \hat{\mu}_t$.

One can now take an analogous definition of worst-case expected regret to be

$$
\begin{aligned}
\mathrm{R}_{\mathcal{A}}(T, K, N) := \sup_{\ell_{1:T}} \sup_{\mu_{1:T}^{1:N}} & \left( \mathbb{E} \left[ \sum_{t=1}^{T} \ell_t(\mathcal{A}(\mathcal{H}_t, \mu_{1:t}^{1:N})) \right] \right. \\
& \left. - \inf_{j \in [N]} \sum_{t=1}^{T} \sum_{i=1}^{K} \mu_t^i(j) \cdot \ell_t(i) \right)
\end{aligned}
$$

where the expectation is taken only with respect to the randomness of the learner. As for adversarial bandits, our definition of minimax regret for bandits with experts advice implicitly assumes an oblivious adversary.

### 2.4. Differential Privacy

In this work, we are interested in designing bandit algorithms that have low expected regret while satisfying the constraint of *differential privacy*. Roughly speaking, differential privacy quantifies the following algorithmic property: an algorithm $\mathcal{A}$ is a *private* bandit algorithm if, for any two sequences of losses that differ in exactly one position, the distributions over actions induced by running $\mathcal{A}$ on the two loss sequences are close. Definition 2.3 formalizes this notion of privacy in adversarial bandits.

**Definition 2.3** (($\varepsilon, \delta$)-Differential Privacy in Adversarial Bandits (Dwork et al., 2014)). A bandit algorithm $\mathcal{A}$ is ($\varepsilon, \delta$)-differentially private if for every $T \in \mathbb{N}$, any two sequences of loss functions $\ell_{1:T}$ and $\ell'_{1:T}$ differing at exactly one time point $t' \in [T]$, and any $E \subset [K]^T$, we have that $\mathbb{P}[(\mathcal{A}(\mathcal{H}_1), \mathcal{A}(\mathcal{H}_2), \ldots, \mathcal{A}(\mathcal{H}_T)) \in E] \leq e^{\varepsilon} \mathbb{P}[(\mathcal{A}(\mathcal{H}'_1), \mathcal{A}(\mathcal{H}'_2), \ldots, \mathcal{A}(\mathcal{H}'_T)) \in E] + \delta$, where we let $\mathcal{H}_t = (I_s, \ell_s(I_s))_{s=1}^{t-1}$ and $\mathcal{H}'_t = (I'_s, \ell'_s(I_s))_{s=1}^{t-1}$.

We note that the our notion of differential privacy in Definition 2.3 is inherently for an *oblivious* adversary. A different definition of privacy is required if the adversary is allowed to be *adaptive* i.e., having the ability to pick the loss $\ell_t$ using the realized actions $I_1, \ldots, I_{t-1}$ played by the learner (see Definition 2.1 in Asi et al. (2023) for more details). While the utility guarantees of our bandit algorithms hold only for oblivious adversaries, their privacy guarantees hold against adaptive adversaries.

We use an analogous definition of differential privacy for bandits with expert advice.

**Definition 2.4** (($\varepsilon, \delta$)-Differential Privacy in Bandits with Expert Advice (Dwork et al., 2014)). A bandit with expert advice algorithm $\mathcal{A}$ is ($\varepsilon, \delta$)-differentially private if for every $T \in \mathbb{N}$, any two sequences of loss functions $\ell_{1:T}$ and $\ell'_{1:T}$ differing at exactly one time point $t' \in [T]$, and any $E \subset [K]^T$, we have that $\mathbb{P}[(\mathcal{A}(\mathcal{H}_1), \mathcal{A}(\mathcal{H}_2), \ldots, \mathcal{A}(\mathcal{H}_T)) \in E] \leq e^{\varepsilon} \mathbb{P}[(\mathcal{A}(\mathcal{H}'_1), \mathcal{A}(\mathcal{H}'_2), \ldots, \mathcal{A}(\mathcal{H}'_T)) \in E] + \delta$, where we let $\mathcal{H}_t = (I_s, \ell_s(I_s))_{s=1}^{t-1}$ and $\mathcal{H}'_t = (I'_s, \ell'_s(I_s))_{s=1}^{t-1}$.

Note that Definition 2.4 implicitly assumes that only the sequence of losses is sensitive information and that expert predictions are public.

Our main focus in this work will be on designing bandit algorithms that satisfy *pure* differential privacy (i.e. when $\delta = 0$). In Appendix A, we review several fundamental properties of privacy and privacy-preserving mechanisms that serve as important building blocks.

## 3. Faster Rates for Private Adversarial Bandits

In this section, we establish a connection between non-private bandit algorithms that can handle negative losses and $\varepsilon$-differentially private bandit algorithms. Let $\mathcal{B}$ be any bandit algorithm and define

$$
\begin{aligned}
\tilde{\mathrm{R}}_{\mathcal{B}}(T, K, \lambda) := \sup_{\ell_{1:T}} & \left( \mathbb{E} \left[ \sum_{t=1}^{T} \tilde{\ell}_t(\mathcal{B}(\tilde{\mathcal{H}}_t)) \right] - \inf_{i \in [K]} \mathbb{E} \left[ \sum_{t=1}^{T} \tilde{\ell}_t(i) \right] \right) \\
= \sup_{\ell_{1:T}} & \left( \mathbb{E} \left[ \sum_{t=1}^{T} \ell_t(\mathcal{B}(\tilde{\mathcal{H}}_t)) \right] - \inf_{i \in [K]} \sum_{t=1}^{T} \ell_t(i) \right),
\end{aligned}
$$

where $\tilde{\ell}_t(i) = \ell_t(i) + Z_t(i)$ with $Z_t(i) \sim \mathrm{Lap}(\lambda)$, $\tilde{\mathcal{H}}_t = (I_s, \tilde{\ell}_s(I_s))_{s=1}^{t-1}$, and the expectation is taken with respect to both the randomness of $\mathcal{B}$ and the losses $\tilde{\ell}_{1:T}$. Theorem 3.1 states that one can always convert $\mathcal{B}$ into an $\varepsilon$-differentially private bandit algorithm $\mathcal{A}$ whose regret guarantees can be written in terms of $\tilde{\mathrm{R}}_{\mathcal{B}}(T, K, \lambda)$.

**Theorem 3.1** (Generic Conversion). *Let $\mathcal{B}$ be any bandit algorithm. Then, for every $\tau \geq 1$ and $\varepsilon \leq 1$, there exists an $\varepsilon$-differentially private bandit algorithm $\mathcal{A}_{\tau}$ such that*

$$
\mathrm{R}_{\mathcal{A}_{\tau}}(T, K) \leq \tau \tilde{\mathrm{R}}_{\mathcal{B}} \left( \frac{T}{\tau}, K, \frac{1}{\varepsilon \tau} \right) + \tau.
$$

*In particular, picking $\tau = \lceil \frac{1}{\varepsilon} \rceil$ means that there exists a $\varepsilon$-differentially private bandit algorithm $\mathcal{A}$ such that*

$$
\mathrm{R}_{\mathcal{A}}(T, K) \leq \frac{2}{\varepsilon} \tilde{\mathrm{R}}_{\mathcal{B}} (\varepsilon T, K, 1) + \frac{2}{\varepsilon}.
$$

As a corollary of Theorem 3.1, we establish new upper bounds on the expected regret under the constraint of $\varepsilon$-differential privacy that improves on existing work *in all*

*regimes* of $\varepsilon > 0$. In particular, Corollary 3.2 follows by letting $\mathcal{B}$ be the HTINF algorithm from Huang et al. (2022), which modifies Follow-the-Regularized-Leader (FTRL) for heavy-tailed losses.

**Corollary 3.2** (FTRL Conversion)**.** *For every* $\varepsilon \in [\frac{1}{T}, 1]$, *if $\mathcal{B}$ is* HTINF *with $\alpha = 2$ and $\sigma = \sqrt{6}$, then Algorithm 1, when run with $\mathcal{B}$ and $\tau = \lceil \frac{1}{\varepsilon} \rceil$, is $\varepsilon$-differentially private and suffers worse-case expected regret at most*

$$O\left( \frac{\sqrt{TK}}{\sqrt{\varepsilon}} + \frac{1}{\varepsilon} \right).$$

In Appendix E, we instantiate Theorem 3.1 with EXP3 and FTPL to obtain two other upper bounds. In every case, our upper bounds establish the first known separation in rates between central differential privacy and local differential privacy (see Appendix A for definition) for this problem. Namely, while the lower bounds from Basu et al. (2019) show that any local $\varepsilon$-differentially private bandit algorithm must suffer linear $\Omega(T)$ expected regret when $\varepsilon < \frac{1}{\sqrt{T}}$, Corollary 3.2 gives an algorithm satisfying $\varepsilon$-central differential privacy (i.e. Definition 2.3) whose expected regret is sublinear $o(T)$ even when $\varepsilon < \frac{1}{\sqrt{T}}$. The remainder of this section is dedicated to proving Theorem 3.1. Corollary 3.2 is proven in Appendix D.

### 3.1. Proof of Theorem 3.1

The conversion behind Theorem 3.1 is remarkably simple. At a high-level, it requires simulating the non-private bandit algorithm on noisy batched losses. That is, instead of passing every loss to the non-private bandit algorithm, we play the same arm for a batch size $\tau$, average the loss across this batch, add independent Laplace noise to the batched loss, and then pass this noisy batched loss to the non-private bandit algorithm. By adding Laplace noise to batched losses as opposed to the original losses (as is done by Tossou & Dimitrakakis (2017) and Agarwal & Singh (2017)), the magnitude of the required noise is reduced by a multiplicative factor of the batch size.

However, a key issue that needs to be handled when adding noise (whether to batched or un-batched losses) is the fact that the losses fed to the non-private bandit algorithm can now be negative and unbounded. Accordingly, in order to get any meaningful utility guarantees, Theorem 3.1 effectively requires our non-private bandit algorithm to handle unbounded, negative (but still unbiased) losses. Fortunately, there are several existing adversarial bandit algorithms that can achieve low expected regret while observing negative losses. Three of these are presented in Corollary 3.2, E.1, and E.3. To the best of our knowledge, this is the first work to establish a connection between handling negative losses (for example in works that handle heavy-tailed losses) and

---

**Algorithm 1** Non-Private to Private Conversion

1: **Input:** Non-private bandit algorithm $\mathcal{B}$, batch size $\tau$, privacy parameter $\varepsilon \in (0, 1]$
2: **Initialize:** $j = 1$
3: **for** $t = 1, \ldots, T$ **do**
4:     **if** $t = (j-1)\tau + 1$ **then**
5:         Receive action $I_j$ from $\mathcal{B}$.
6:     **end if**
7:     Play action $I_t := I_j$
8:     Observe loss $\ell_t(I_t)$.
9:     **if** $t = j\tau$ **then**
10:        Define $\hat{\ell}_j(i) := \frac{1}{\tau} \sum_{s=(j-1)\tau+1}^{j\tau} \ell_s(i)$
11:        Pass $\hat{\ell}_j(I_j) + Z_j$ to $\mathcal{B}$, where $Z_j \sim \text{Lap}(\frac{1}{\tau\varepsilon})$.
12:        Update $j \leftarrow j + 1$.
13:     **end if**
14: **end for**

---

(non-local) differential privacy.

Algorithm 1 provides the pseudo code for converting a non-private bandit algorithm to a private bandit algorithm.

**Lemma 3.3** (Privacy guarantee)**.** *For every bandit algorithm $\mathcal{B}$, batch size $\tau \geq 1$, and $\varepsilon \leq 1$, Algorithm 1 is $\varepsilon$-differentially private.*

*Proof.* (sketch of Lemma 3.3) Observe that Algorithm 1 applies the bandit algorithm $\mathcal{B}$ on the losses $\hat{\ell}_1, \ldots, \hat{\ell}_{\lfloor \frac{T}{\tau} \rfloor}$ in a black box fashion. Accordingly, the privacy guarantee of Algorithm 1 follows from the privacy guarantee of $\hat{\ell}_1(I_1), \ldots, \hat{\ell}_{\lfloor \frac{T}{\tau} \rfloor}(I_{\lfloor \frac{T}{\tau} \rfloor})$ and post-processing. The privacy of each $\hat{\ell}_j(I_j)$ follows from the Laplace mechanism. ∎

A rigorous proof of Lemma 3.3 is in Appendix C.

**Lemma 3.4** (Utility guarantee)**.** *For every bandit algorithm $\mathcal{B}$, batch size $\tau \geq 1$, and $\varepsilon \leq 1$, the worst-case expected regret of Algorithm 1 is at most $\tau \tilde{R}_{\mathcal{B}}(\frac{T}{\tau}, K, \frac{1}{\varepsilon\tau}) + \tau$.*

The proof of Lemma 3.4 follows from the following result by Arora et al. (2012).

**Theorem 3.5** (Theorem 2 in (Arora et al., 2012))**.** *Let $\mathcal{B}$ be any bandit algorithm. Let $\tau \geq 1$ be a batch size and let $\mathcal{A}_\tau$ be the batched version of $\mathcal{B}$. That is, the bandit algorithm $\mathcal{A}_\tau$ groups the rounds $1, \ldots, T$ into consecutive and disjoint batches of size $\tau$ such that the $j$'th batch begins on round $(j-1)\tau + 1$ and ends on round $j\tau$. At the start of each batch $j$ the algorithm $\mathcal{A}_\tau$ calls $\mathcal{B}$ and receives an action $I_j$ drawn from $\mathcal{B}$'s internal distribution. Then, $\mathcal{A}_\tau$ plays this action for $\tau$ rounds. At the end of the batch, $\mathcal{A}_\tau$ feeds $\mathcal{B}$ with the average loss value $\frac{1}{\tau} \sum_{s=(j-1)\tau+1}^{j\tau} \ell_s(I_j)$. For such an algorithm $\mathcal{A}_\tau$, its worst-case expected regret is at most $\tau R_{\mathcal{B}}\left( \frac{T}{\tau}, K \right) + \tau$.*

Note that Algorithm 1 is precisely the batched version of its input $\mathcal{B}$. Accordingly, Theorem 3.5 immediately implies that on any sequence $\ell_{1:T}$, the expected regret of Algorithm 1 is at most $\tau \tilde{R}_\mathcal{B}(\frac{T}{\tau}, K, \frac{1}{\varepsilon\tau}) + \tau$. We provide a complete proof of Lemma 3.4 in Appendix C.

## 4. Upper bounds for Bandits with Expert Advice

Theorem 3.1 also allows us to give guarantees for bandits with expert advice. To do so, we need Theorem 4.1, due to Auer et al. (2002), which shows that any bandit algorithm can be converted into a bandit with expert advice algorithm in a black-box fashion. For completeness, we provide this conversion and the proof of Theorem 4.1 in Appendix F.

**Theorem 4.1** (Bandit to Bandit with Expert Advice). *Let $\mathcal{B}$ be any bandit algorithm and $R_\mathcal{B}(T, K)$ denote its worst-case expected regret. Then, the worst-case expected regret of Algorithm 5 when initialized with $\mathcal{B}$ is at most $R_\mathcal{B}(T, N)$.*

By treating each expert as a meta-action, Theorem 3.1 and Theorem 4.1 can be used to convert a non-private bandit algorithm $\mathcal{B}$ into a private bandit with expert advice algorithm $\mathcal{A}$ in the following way: given a non-private bandit algorithm $\mathcal{B}$, use Theorem 3.1 to convert it into a private bandit algorithm $\mathcal{B}'$. Then, use Theorem 4.1 to convert $\mathcal{B}'$ into a private bandit with expert advice algorithm $\mathcal{A}$. By post-processing, the corresponding actions played by $\mathcal{A}$ are also private. In fact, this conversion also satisfies a stronger notion of privacy where the expert advice is also taken to be sensitive information. Theorem 4.2 formalizes this conversion.

**Theorem 4.2** (Generic Conversion). *Let $\mathcal{B}$ be any bandit algorithm. Then, for every $\tau \geq 1$ and $\varepsilon \leq 1$, there exists an $\varepsilon$-differentially private bandit with expert advice algorithm $\mathcal{A}_\tau$ such that*

$$R_{\mathcal{A}_\tau}(T, K, N) \leq \tau \tilde{R}_\mathcal{B}\left(\frac{T}{\tau}, N, \frac{1}{\varepsilon\tau}\right) + \tau.$$

*In particular, by setting $\tau = \lceil \frac{1}{\varepsilon} \rceil$, there exists an $\varepsilon$-differentially private bandit with expert advice algorithm $\mathcal{A}$ such that*

$$R_\mathcal{A}(T, K, N) \leq \frac{2}{\varepsilon} \tilde{R}_\mathcal{B}(\varepsilon T, N, 1) + \frac{2}{\varepsilon}.$$

The proof of Theorem 4.2 is deferred to Appendix F since it closely follows that of Theorem 3.1. Using HTINF for $\mathcal{B}$ in Theorem 4.2 gives the following corollary.

**Corollary 4.3** (FTRL Conversion). *For every $\varepsilon \in [\frac{1}{T}, 1]$, if $\mathcal{B}$ is HTINF with $\alpha = 2$ and $\sigma = \sqrt{6}$, then Theorem 4.2 guarantees the existence of an $\varepsilon$-differentially private*

*algorithm whose worst-case expected regret at most*

$$O\left(\frac{\sqrt{TN}}{\sqrt{\varepsilon}} + \frac{1}{\varepsilon}\right).$$

The upper bound in Corollary 4.3 is non-vacuous for constant or small $N$ (i.e. $N \leq K$). However, this bound is vacuous when $N$ grows with $T$. To address this, we consider EXP4 which enjoys expected regret $O\left(\sqrt{KT \log(N)}\right)$ in the non-private setting, exhibiting only a poly-logarithmic dependence on $N$ (Auer et al., 2002). The following theorem shows that by adding independent Laplace noise to each observed loss, a similar improvement over Corollary 4.3 can be established for large $N$, at the cost of a worse dependence on $\varepsilon$.

**Theorem 4.4** (Locally Private EXP4). *For every $\varepsilon \leq 1$, Algorithm 6 when run with $\eta = \sqrt{\frac{\log(N)}{3TK\left(1 + \frac{10\log^2(KT)}{\varepsilon^2}\right)}}$ and $\gamma = \frac{4\eta K \log(KT)}{\varepsilon}$ is $\varepsilon$-differentially private and suffers worst-case expected regret at most*

$$O\left(\frac{\sqrt{TK \log(N)} \log(KT)}{\varepsilon}\right).$$

Due to space constraints, we defer Algorithm 6, which just adds independent Laplace noise to each observed loss, to Appendix F. Note that when $N \leq K$, the upper bound in Corollary 4.3 is still superior to that of Theorem 4.4 for all ranges of $\varepsilon \leq 1$. The proof of Theorem 4.4 is also deferred to Appendix F.

Algorithm 6 provides a stronger privacy guarantee than what is actually necessary. Indeed, by adding independent Laplace noise to each observed loss, Algorithm 6 actually satisfies $\varepsilon$-*local* differential privacy (see Appendix A for definition). Accordingly, in contrast to Corollary 3.2, the upper bound in Theorem 4.4 is vacuous for $\varepsilon \leq \frac{1}{\sqrt{T}}$. The following algorithm uses the batching technique from Section 3 to improve the dependence in $\varepsilon$ from Theorem 4.4 while also improving the dependence on $N$ from Corollary 4.3.

**Theorem 4.5** (Private, Batched EXP4). *For every $\varepsilon, \delta \in (0, 1]$, Algorithm 2, when run with*

$$\eta = \frac{(N\log(\frac{1}{\delta}))^{1/6}\log^{1/3}(NT)\log^{1/3}(N)}{T^{1/3}K^{1/2}\varepsilon^{1/3}},$$

$$\tau = \frac{(N\log(\frac{1}{\delta}))^{1/3}\log^{2/3}(NT)T^{1/3}}{\varepsilon^{2/3}\log^{1/3}(N)},$$

**Algorithm 2** Private, Batched EXP4

1: **Input:** Action space $[K]$, Number of experts $N$, batch size $\tau$, privacy parameters $\varepsilon, \delta > 0$, learning rate $\eta$, mixing parameter $\gamma > 0$
2: **Initialize:** $r = 1$, $w_1(j) = 1$ for all $j \in [N]$
3: **for** $t = 1, \ldots, T$ **do**
4:   Receive expert advice $\mu_t^1, \ldots, \mu_t^N \in \Pi([K])$
5:   **if** $t = (r-1)\tau + 1$ **then**
6:     Set $P_r(j) \leftarrow \frac{w_r(j)}{\sum_{j \in [N]} w_r(j)}$
7:   **end if**
8:   Set $Q_t(i) \leftarrow (1-\gamma) \sum_{j=1}^N P_r(j)\mu_t^j(i) + \frac{\gamma}{K}$.
9:   Draw $I_t \sim Q_t$
10:   Observe loss $\ell_t(I_t)$ and construct unbiased estimator $\hat{\ell}_t(i) = \frac{\ell_t(i)\mathbb{I}\{I_t = i\}}{Q_t(i)}$
11:   **if** $t = r\tau$ **then**
12:     Define $\tilde{\ell}_r(j) := \frac{1}{\tau}\sum_{s=(r-1)\tau+1}^{r\tau} \hat{\ell}_s \cdot \mu_s^j$ and $\tilde{\ell}_r'(j) := \tilde{\ell}_r(j) + Z_r^j$ where

$$Z_r^j \sim \text{Lap}\left(0, \frac{3K\sqrt{N\log(\frac{1}{\delta})}}{\gamma\tau\varepsilon}\right).$$

13:     Update $w_{r+1}(j) \leftarrow w_r(j) \cdot \exp\{-\eta\tilde{\ell}_r'(j)\}$
14:     Update $r \leftarrow r + 1$.
15:   **end if**
16: **end for**

*and*

$$\gamma = \max\left\{\frac{\eta^{1/3}N^{1/3}K^{2/3}\log^{2/3}(NT)}{\varepsilon^{2/3}\tau^{2/3}}, \frac{12\eta K\sqrt{N\log(\frac{1}{\delta})}\log(NT)}{\varepsilon\tau}\right\},$$

*satisfies $(\varepsilon, \delta)$-differentially privacy and suffers worst-case expected regret at most*

$$O\left(\frac{N^{1/6}K^{1/2}T^{2/3} \cdot \log^{1/6}(\frac{1}{\delta})\log^{1/3}(NT)\log^{1/3}(N)}{\varepsilon^{1/3}} + \frac{N^{1/2} \cdot \log(\frac{1}{\delta})^{1/2}\log(NT)\log(N)}{\varepsilon}\right).$$

The proof of Theorem 4.5 modifies the standard proof of EXP4 to handle the noisy, batched losses. See Appendix F for the full proof. Compared to Theorem 4.3 and 4.4, Theorem 4.5 shows that Algorithm 2 enjoys sublinear regret even when $N \geq T^{1/4}$ and $\varepsilon = \frac{1}{\sqrt{T}}$. Table 2 summarizes the three regimes and the corresponding algorithm that obtains the best regret bound in that regime. Our upper bounds for bandits with expert advice become vacuous when $\varepsilon \leq \frac{1}{\sqrt{T}}$

and $N \geq T$. We leave deriving non-vacuous upper bounds for this regime as an interesting direction for future work.

## 5. Barriers to Even Faster Rates for Private Adversarial Bandits

In this work, we provided new algorithms for the private adversarial bandit problem and its expert advice counterpart. In the adversarial bandits setting, we provided a generic conversion of a non-private bandit algorithm into a private bandit algorithm. Instantiating our conversion with existing bandit algorithms resulted in private bandit algorithms whose worst-case expected regret improve upon all existing work in all privacy regimes. In the bandits with expert advice setting, we provide, to the best of our knowledge, the first private adversarial bandit algorithms by modifying EXP4.

An important direction of future work is answering whether it is possible to achieve an *additive* separation in $\varepsilon$ and $T$. We note that this is possible in the stochastic bandit setting (Azize & Basu, 2022) as well as the the full-information adversarial online setting (Agarwal & Singh, 2017). To this end, we conclude our paper by discussing some road blocks when attempting to derive such guarantees for the adversarial bandit setting.

### 5.1. On the Hardness of Privatizing EXP3

First, we comment on the difficulty of privatizing EXP3. In the full-information setting, a standard privacy analysis for exponential weights shows that for every $t \in [T]$, the per-round privacy loss at time step $t$ is at most $2\eta$, and for $\eta = \frac{\varepsilon}{\sqrt{T}}$ advanced composition yields $(\varepsilon, \delta)$-differential privacy with expected regret $O(\sqrt{T\log(K)}/\varepsilon)$ (Dwork et al., 2014).

Unfortunately, it is not easy to bound the per-round privacy loss of EXP3 uniformly across time. This is because EXP3 uses Inverse-Probability-Weighted estimators (Robins et al., 1994) (see Algorithm 3). Thus the algorithm needs to know not just the arm $I_t$ but also the probability $P_t$ with which it was selected. It is, however, not clear how to account for the privacy cost of releasing $P_t$ and indeed we can construct examples where the per-round privacy loss grows with the time horizon $T$. We provide a more formal analysis of this issue in Appendix G.1.

### 5.2. Limits of a Class of Adversarial Bandit Algorithms

In Section 3, we established upper bounds of $O(\frac{\sqrt{KT}}{\sqrt{\varepsilon}})$ on the expected regret for the private adversarial bandit problem. Unfortunately, by exploiting the ability to pick arbitrary sequences of loss functions, we can show that for a large class of bandit algorithms, including EXP3 and its batched

*Table 2.* Summary of the three regimes for bandits with expert advice and the corresponding algorithm that obtains the best regret bound in that regime.

| Regime | Best Alg. | Guarantee |
|---|---|---|
| Low-dimension, High Privacy ($N \le K$) | Cor. 4.3 | $O(\frac{\sqrt{NT}}{\sqrt{\epsilon}})$ |
| High-dimension, Low Privacy ($N \ge K$, $\epsilon \ge \frac{K}{N}$) | Thm. 4.4 | $O(\frac{\sqrt{KT \log N} \log(KT)}{\epsilon})$ |
| High-dimension, High Privacy ($N \ge K$, $\epsilon \le \frac{1}{\sqrt{T}}$) | Thm. 4.5 | $O(\frac{N^{1/6} K^{1/2} T^{2/3} \log(NT)}{\epsilon^{1/3}} + \frac{N^{1/2} \log(NT)}{\epsilon})$ |

variants, one cannot significantly improve upon this upper bound. Informally, our result holds for any (adaptively) private bandit algorithm that "quickly" reduces the probability of playing a sub-optimal arm. We provide the formal details below, starting with some intuition.

Consider an instance on two arms where arm 1 has loss $\frac{1}{2}$ at each step, while arm 2 has loss 1 at each step. Any algorithm that has regret $R$ must play arm 2 at most $O(R)$ times on this instance. Informally, our lower bound applies to bandit algorithms that drops the probability of playing arm 2 to be about $\frac{R}{T}$ within about $o(T/R)$ steps. We note that EXP3 drops this probability to $O(\frac{R}{T})$ in $O(\log T)$ steps. For algorithms of this kind, our lower bound shows that any $\varepsilon$-differentially private algorithm (for $\varepsilon < 1$) must incur regret $O(\sqrt{T/\varepsilon})$. Intuitively, the lower bound follows from the fact that if the loss of arm 2 falls to 0 at step $\approx T/R$ (while arm 1 is unchanged at $\frac{1}{2}$), then an $\varepsilon$-differentially private algorithm must pull arm 2 at least $\frac{1}{\varepsilon}$ times to "notice" this change. Accounting for the accumulated regret in the time it takes to pull arm 2 sufficiently many times, and setting parameters appropriately yields the lower bound.

More formally, fix $K = 2$ and $T \in \mathbb{N}$. For $\gamma \in [0, 1]$, $\tau \in \{1, \ldots, T\}$ and $p \in [T]$, define the sets

$$E_\gamma := \left\{ i_{1:T} : \sum_{t=1}^{T} \mathbb{I}\{i_t = 2\} \ge \gamma T \right\}$$

and

$$E_{\gamma,\tau}^p := \left\{ i_{1:T} : \sum_{s=\tau+1}^{\tau+\frac{p}{\gamma}} \mathbb{I}\{i_s = 2\} \le p \right\}.$$

Consider the sequence of loss functions $\ell_1, \ldots, \ell_T$, such that $\ell_{1:T}(2) = 1$ and $\ell_{1:T}(1) = \frac{1}{2}$. Our assumptions on the bandit algorithms are with respect to their behavior on $\ell_1, \ldots, \ell_T$. In particular, we will consider bandit algorithms $\mathcal{A}$ for which there exists $\gamma \in [0, 1]$, $\tau < \frac{T}{2}$ and $\gamma\tau \le p \le \gamma(T - \tau)$ such that:

(1) $\mathbb{P}(I_1, \ldots, I_T \in E_\gamma) \ge \frac{1}{2}$ and

(2) $\mathbb{P}(I_1, \ldots, I_T \in E_{\gamma,\tau}^p) \ge \frac{1}{2}$,

where $I_{1:T}$ are the random variables denoting the actions played by $\mathcal{A}$ when run on the sequence of loss functions

$\ell_{1:T}$. The first condition simply lower bounds the probability that $\mathcal{A}$ plays action 2 by $\gamma$, when $\mathcal{A}$ is run on $\ell_{1:T}$. The second condition states that $\mathcal{A}$ drops, and subsequently maintains, the probability of playing action 2 to $\gamma$ in roughly $\tau$ rounds. Accordingly, when $\tau$ is small, condition (2) states that $\mathcal{A}$ drops the probability of playing action 2 down to $\gamma$ relatively quickly. One should really think of $\gamma$ as being $O\left(\frac{R_\mathcal{A}(\ell_{1:T})}{T}\right)$, where $R_\mathcal{A}(\ell_{1:T})$ denotes the expected regret of $\mathcal{A}$ when run on $\ell_{1:T}$. Then, condition (1) is trivially satisfied, while condition (2) states that $\mathcal{A}$ roughly drops and keeps the probability of playing action 2 around $O\left(\frac{R_\mathcal{A}(\ell_{1:T})}{T}\right)$ by round $\tau$, and keeps it there. In other words, after round $\tau$, $\mathcal{A}$ plays action 2 on average $p$ times in $\frac{p}{\gamma} \approx O(\frac{pT}{R_\mathcal{A}(T)})$ rounds. The latter property is reasonable for bandit algorithms given that $\ell_t(2) - \ell_t(1) = \frac{1}{2}$ for all $t \in [T]$, and thus any low-regret bandit algorithm should not be playing action 2 often. Lemma 5.1 provides a lower bound on the expected regret of private bandit algorithms that satisfy these two conditions.

**Lemma 5.1.** *Let $\varepsilon \le 1$ and $\mathcal{A}$ be any $\varepsilon$-differentially private algorithm. If $\mathcal{A}$ satisfies conditions (1) and (2) with parameters $\gamma \in [0, \frac{1}{2}]$, $\tau < \frac{T}{2}$ and $2\gamma\tau \le p \le \gamma(T - \tau)$, then the worst-case expected regret of $\mathcal{A}$ is at least*

$$\max\left\{ \frac{\gamma T}{4}, \left(1 - \frac{1}{2} e^{\varepsilon p}\right) \frac{p}{4\gamma} - \frac{\tau}{2} \right\} \ge$$

$$\sqrt{\frac{pT(1 - \frac{1}{2} e^{\varepsilon p})}{16}} - \frac{\tau}{2}.$$

*In particular, if $\mathcal{A}$ satisfies conditions (1) and (2) with parameters $\gamma \in [0, \frac{1}{2}]$, $\tau \in o\left(\sqrt{\frac{T}{\varepsilon}}\right)$, and $p = O(\frac{1}{\varepsilon})$, then the worst-case expected regret of $\mathcal{A}$ is $\Omega\left(\sqrt{\frac{T}{\varepsilon}}\right)$.*

Lemma 5.1 shows that if one wants to design an $\varepsilon$-differentially private algorithm (for $\varepsilon \le 1$) whose upper bound enjoys an additive separation between $T$ and $\varepsilon$, then there cannot exist a $\gamma \in [0, \frac{1}{2}]$ such that it satisfies conditions (1) and (2) with $\tau \in o\left(\sqrt{\frac{T}{\varepsilon}}\right)$, and $p \le \gamma(T - \tau)$. Unfortunately, an implication of Lemma 5.1 is that such an additive separation is not possible for EXP3 and its batched variants. Indeed, batched EXP3 on the loss sequence $\ell_{1:T}$,

where $\ell_t(1) = 1/2$ and $\ell_t(2) = 1$, has regret at least $\gamma T + \kappa/\eta$, where $\kappa$ is the batch size. Assume, towards a contradiction that for some setting of parameters, this was $o(\sqrt{T/\varepsilon})$. This implies that $\gamma$ is $o(1/\sqrt{\varepsilon T})$, and that $\kappa/\eta$ is $o(\sqrt{T/\varepsilon})$. EXP3 drops the probability of a bad arm to $O(\gamma)$ in $\log(1/\gamma)/\eta$ steps, and the batched version will do so in $\kappa \log(1/\gamma)/\eta$ steps. Thus condition (2) is satisfied with $\tau$ being $o(\sqrt{T/\varepsilon})$ and $p = \frac{1}{\varepsilon}$ (ignoring logarithmic factors). This then yields a lower bound of $\omega(\sqrt{T/\varepsilon})$ using Lemma 5.1, which contradicts the assumption. Thus the expected regret has to be $\Omega(\sqrt{T/\varepsilon})$ up to logarithmic factors. We provide the proof of Lemma 5.1 in Appendix G.

## 6. Future work

There are several important directions of future work. First, it is important to understand whether an additive separation between $\varepsilon$ and $T$ is possible under bandit feedback. Note that this is the case in the full-information setting under an oblivious adversary (Asi et al., 2023). Another important direction is improving our upper bounds for private bandits with expert advice. In particular, it would be interesting to see whether sublinear regret is possible for all $N > 1$ and $\epsilon \in \omega(\frac{1}{T})$.

## Impact Statement

This paper presents work whose goal is to advance the field of Machine Learning. There are many potential societal consequences of our work, none which we feel must be specifically highlighted here.

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

# A. Privacy properties and privacy-preserving mechanisms

**Definition A.1** ($\varepsilon$-indistinguishability). Let $X$ and $Y$ be random variables with support $\mathcal{X}$. Let

$$D_\infty(X||Y) := \max_{S \subseteq \mathcal{X}} \left[ \ln\left( \frac{\mathbb{P}(X \in S)}{\mathbb{P}(Y \in S)} \right) \right]$$

be the max divergence. Then $X$ and $Y$ are $\varepsilon$-indistinguishable if and only if

$$\max\{D_\infty(X||Y), D_\infty(Y||X)\} \leq \varepsilon.$$

**Definition A.2** (($\varepsilon, \delta$)-indistinguishability). Let $X$ and $Y$ be random variables with support $\mathcal{X}$. Let

$$D_\infty^\delta(X||Y) := \max_{S \subseteq \mathcal{X}, \mathbb{P}(X \in S) \geq \delta} \left[ \ln\left( \frac{\mathbb{P}(X \in S) - \delta}{\mathbb{P}(Y \in S)} \right) \right]$$

be the $\delta$-approximate max divergence. Then $X$ and $Y$ are $(\varepsilon, \delta)$-indistinguishable if and only if

$$\max\{D_\infty^\delta(X||Y), D_\infty^\delta(Y||X)\} \leq \varepsilon.$$

The follow lemma relates the two notions of indistinguishability to differential privacy.

**Lemma A.3** (Differential privacy $\equiv$ Indistinguishability (Remark 3.2 in (Dwork et al., 2014))). *Let $\mathcal{X}$ and $\mathcal{Y}$ be arbitrary sets. Let $\mathcal{A}$ be a randomized algorithm such that $\mathcal{A} : \mathcal{X}^n \to \mathcal{Y}$. Then, $\mathcal{A}$ is $\varepsilon$-differentially private if and only if for every pair of neighboring datasets $x_{1:n}$ and $x'_{1:n}$, we have that the random variables $\mathcal{A}(x_{1:n})$ and $\mathcal{A}(x'_{1:n})$ are $\varepsilon$-indistinguishable. Likewise, $\mathcal{A}$ is $(\varepsilon, \delta)$-differentially private if and only if for every pair of neighboring datasets $x_{1:n}$ and $x'_{1:n}$, we have that the random variables $\mathcal{A}(x_{1:n})$ and $\mathcal{A}(x'_{1:n})$ are $(\varepsilon, \delta)$-indistinguishable.*

Next, we cover composition.

**Lemma A.4** (Basic Composition (Corollary 3.15 in (Dwork et al., 2014))). *Let $\mathcal{X}, \mathcal{Y}_1, \mathcal{Y}_2, \ldots, \mathcal{Y}_T$ be arbitrary sets and $n \in \mathbb{N}$. Let $\mathcal{A}_1, \mathcal{A}_2, \ldots, \mathcal{A}_T$ be a sequence of randomized algorithms where $\mathcal{A}_1 : \mathcal{X}^n \to \mathcal{Y}_1$ and $\mathcal{A}_t : \mathcal{Y}_1, \ldots, \mathcal{Y}_{t-1}, \mathcal{X}^n \to \mathcal{Y}_t$ for all $t = 2, 3, \ldots, T$. If for every $t \in [T]$ and every $y_{1:t-1} \in \mathcal{Y}_1 \times \mathcal{Y}_2 \times \cdots \times \mathcal{Y}_{t-1}$, we have that $\mathcal{A}_t(y_{1:t-1}, \cdot)$ is $\varepsilon_t$-differentially private, then the overall algorithm $\mathcal{A} : \mathcal{X}^n \to \mathcal{Y}_1 \times \mathcal{Y}_2 \times \cdots \times \mathcal{Y}_T$, defined as*

$$\mathcal{A}(x_{1:n}) = \Big( \mathcal{A}_1(x_{1:n}), \mathcal{A}_2(\mathcal{A}_1(x_{1:n}), x_{1:n}), \ldots, \mathcal{A}_T(\mathcal{A}_1(x_{1:n}), \mathcal{A}_2(\mathcal{A}_1(x_{1:n}), x_{1:n}), \ldots, x_{1:n}) \Big),$$

*satisfies $\varepsilon T$-differential privacy.*

**Lemma A.5** (Basic Composition (Corollary 3.15 in (Dwork et al., 2014))). *Let $\mathcal{X}, \mathcal{Y}_1, \mathcal{Y}_2, \ldots, \mathcal{Y}_T$ be arbitrary sets and $n \in \mathbb{N}$. Let $\mathcal{A}_1, \mathcal{A}_2, \ldots, \mathcal{A}_T$ be a sequence of randomized algorithms where $\mathcal{A}_1 : \mathcal{X}^n \to \mathcal{Y}_1$ and $\mathcal{A}_t : \mathcal{Y}_1, \ldots, \mathcal{Y}_{t-1}, \mathcal{X}^n \to \mathcal{Y}_t$ for all $t = 2, 3, \ldots, T$. If for every $t \in [T]$ and every $y_{1:t-1} \in \mathcal{Y}_1 \times \mathcal{Y}_2 \times \cdots \times \mathcal{Y}_{t-1}$, we have that $\mathcal{A}_t(y_{1:t-1}, \cdot)$ is $\varepsilon_t$-differentially private, then the overall algorithm $\mathcal{A} : \mathcal{X}^n \to \mathcal{Y}_1 \times \mathcal{Y}_2 \times \cdots \times \mathcal{Y}_T$, defined as*

$$\mathcal{A}(x_{1:n}) = \Big( \mathcal{A}_1(x_{1:n}), \mathcal{A}_2(\mathcal{A}_1(x_{1:n}), x_{1:n}), \ldots, \mathcal{A}_T(\mathcal{A}_1(x_{1:n}), \mathcal{A}_2(\mathcal{A}_1(x_{1:n}), x_{1:n}), \ldots, x_{1:n}) \Big),$$

*satisfies $\varepsilon T$-differential privacy.*

**Lemma A.6** (Advanced Composition (Dwork et al., 2010b; Kairouz et al., 2015)). *Let $\mathcal{X}, \mathcal{Y}_1, \mathcal{Y}_2, \ldots, \mathcal{Y}_T$ be arbitrary sets and $n \in \mathbb{N}$. Let $\mathcal{A}_1, \mathcal{A}_2, \ldots, \mathcal{A}_T$ be a sequence of randomized algorithms where $\mathcal{A}_1 : \mathcal{X}^n \to \mathcal{Y}_1$ and $\mathcal{A}_t : \mathcal{Y}_1, \ldots, \mathcal{Y}_{t-1}, \mathcal{X}^n \to \mathcal{Y}_t$ for all $t = 2, 3, \ldots, T$. If for every $t \in [T]$ and every $y_{1:t-1} \in \mathcal{Y}_1 \times \mathcal{Y}_2 \times \cdots \times \mathcal{Y}_{t-1}$, we have that $\mathcal{A}_t(y_{1:t-1}, \cdot)$ is $\varepsilon_t$-differentially private, then for every $\delta' > 0$, the overall algorithm $\mathcal{A} : \mathcal{X}^n \to \mathcal{Y}_1 \times \mathcal{Y}_2 \times \cdots \times \mathcal{Y}_T$, defined as*

$$\mathcal{A}(x_{1:n}) = \Big( \mathcal{A}_1(x_{1:n}), \mathcal{A}_2(\mathcal{A}_1(x_{1:n}), x_{1:n}), \ldots, \mathcal{A}_T(\mathcal{A}_1(x_{1:n}), \mathcal{A}_2(\mathcal{A}_1(x_{1:n}), x_{1:n}), \ldots, x_{1:n}) \Big),$$

*satisfies $(\varepsilon', \delta')$-differential privacy, where*

$$\varepsilon' \leq \frac{3}{2} \sum_{t=1}^{T} \varepsilon_t^2 + \sqrt{6 \sum_{t=1}^{T} \varepsilon_t^2 \log\left( \frac{1}{\delta'} \right)}.$$

Post-processing and group privacy will also be useful.

**Lemma A.7** (Post Processing (Proposition 2.1 in (Dwork et al., 2014))). *Let $\mathcal{X}, \mathcal{Y}, \mathcal{Z}$ be arbitrary sets and $n \in \mathbb{N}$. Let $\mathcal{A} : \mathcal{X}^n \to \mathcal{Y}$ and $\mathcal{B} : \mathcal{Y} \to \mathcal{Z}$ be randomized algorithms. If $\mathcal{A}$ is $(\varepsilon, \delta)$-differentially private then the composed algorithm $\mathcal{B} \circ \mathcal{A} : \mathcal{X}^n \to \mathcal{Z}$ is also $(\varepsilon, \delta)$-differentially private.*

For our lower bounds in Section 5, the notion of group privacy will be useful.

**Lemma A.8** (Group Privacy (Theorem 2.2 in (Dwork et al., 2014))). *Let $\mathcal{X}$ and $\mathcal{Y}$ be arbitrary sets and let $n \in \mathbb{N}$. Suppose $\mathcal{A} : \mathcal{X}^n \to \mathcal{Y}$ is an $\varepsilon$-differentially private algorithm. Then, for every pair of datasets $x_{1:n}, x'_{1:n}$ that differ in $1 \leq k \leq n$ positions and every event $E \subseteq \mathcal{Y}$, we have that*

$$\mathbb{P}\left[\mathcal{A}(x_{1:n}) \in E\right] \leq e^{k\varepsilon}\mathbb{P}\left[\mathcal{A}(x'_{1:n}) \in E\right].$$

For designing algorithms, the following primitive will be useful.

**Definition A.9** (Laplace Mechanism (Definition 3.3 in (Dwork et al., 2014))). *Let $\mathcal{X}$ be an arbitrary set and $n \in \mathbb{N}$. Suppose $f : \mathcal{X}^n \to \mathbb{R}$ is a query with sensitivity $\Delta$ (i.e. for all pairs of datasets $x_{1:n}, x'_{1:n} \in \mathcal{X}^n$ that differ in exactly one index, we have that $|f(x_{1:n}) - f(x'_{1:n})| \leq \Delta$). Then, for every $\varepsilon$, the mechanism $\mathcal{M} : \mathcal{X}^n \to \mathbb{R}$ defined as $\mathcal{M}(x_{1:n}) = f(x_{1:n}) + Z$, where $Z \sim \mathrm{Lap}(\frac{\Delta}{\varepsilon})$, is $\varepsilon$-differentially private.*

Lastly, as we make comparisons to local differential privacy, we define it below for the sake of completeness.

**Definition A.10** (Local differential privacy (Duchi et al., 2013)). *Let $\mathcal{X}$ and $\mathcal{Y}$ be arbitrary sets. A randomized mechanism $M : \mathcal{X} \to \mathcal{Y}$ is $(\varepsilon, \delta)$-LDP, if for every $x \neq x' \in \mathcal{X}$ and any measurable subset $Y \subset \mathcal{Y}$, we have that*

$$\mathbb{P}\left[M(x) \in Y\right] \leq e^{\varepsilon}\mathbb{P}\left[M(x') \in Y\right] + \delta.$$

When $\delta = 0$, we say that $M$ is $\varepsilon$ local differentially private.

# B. Helper Lemmas

**Lemma B.1** (Hazard Rate of Laplace distribution). *Let $\mathcal{D}$ denote the Laplace distribution $Lap(0, \lambda)$, $f$ and $F$ denote its probability and cumulative density functions respectively. Define*

$$h_{\mathcal{D}}(z) := \frac{f(z)}{1 - F(z)}$$

*to be the hazard rate function of $Lap(0, \lambda)$. Then*

$$\sup_{z \in \mathbb{R}} h_{\mathcal{D}}(z) \leq \frac{1}{\lambda}.$$

*Moreover, $h_{\mathcal{D}}(z)$ is non-decreasing in $z$.*

*Proof.* Recall that for $\lambda > 0$, we have

$$f(z) = \frac{1}{2\lambda}\exp\{-\frac{|x|}{\lambda}\}$$

and

$$F(z) = \begin{cases} \frac{1}{2}\exp\{\frac{z}{\lambda}\}, & \text{if } z \leq 0 \\ 1 - \frac{1}{2}\exp\{-\frac{z}{\lambda}\}, & \text{if } z > 0 \end{cases}.$$

Fix $x \in \mathbb{R}$. If $x \leq 0$, then

$$\frac{f(x)}{1 - F(x)} = \frac{\frac{1}{2\lambda}\exp\{\frac{x}{\lambda}\}}{1 - \frac{1}{2}\exp\{\frac{x}{\lambda}\}} \leq \frac{1}{\lambda}$$

Otherwise, note that when $x \geq 0$, we have

$$\frac{f(x)}{1 - F(x)} = \frac{\frac{1}{2\lambda}\exp\{\frac{-x}{\lambda}\}}{\frac{1}{2}\exp\{\frac{-x}{\lambda}\}} = \frac{1}{\lambda}.$$

This shows that $\sup_{x \in \mathbb{R}} h_{\mathcal{D}}(x) \leq \frac{1}{\lambda}$. To see that $h_{\mathcal{D}}(x)$ is non-decreasing, note that when $x \leq 0$, we have that $h_{\mathcal{D}}(x) = \frac{\frac{1}{2\lambda}\exp\{\frac{x}{\lambda}\}}{1 - \frac{1}{2}\exp\{\frac{x}{\lambda}\}}$ is increasing in $x$ and when $x \geq 0$, $h_{\mathcal{D}}(x)$ is constant. ∎

**Lemma B.2** (Truncated Non-negativity of Noisy Losses)**.** *Let $Z \sim Lap(\lambda)$ and $\ell \in [0, 1]$. Then, for any $M \geq 0$, we have that*

$$\mathbb{E}\left[(Z + \ell)\mathbb{I}\{|Z + \ell| > M\}\right] \geq 0.$$

*Proof.* Let $M \geq 0$ and $\ell \in [0, 1]$. Then, we can write

$$\mathbb{E}\left[(Z + \ell)\mathbb{I}\{|Z + \ell| > M\}\right] = \ell \cdot \mathbb{E}\left[\mathbb{I}\{|Z + \ell| > M\}\right] + \mathbb{E}\left[Z\mathbb{I}\{|Z + \ell| > M\}\right].$$

Since $\ell \geq 0$, it suffices to show that $\mathbb{E}\left[Z\mathbb{I}\{|Z + \ell| > M\}\right] \geq 0$. To that end, note that

$$\mathbb{E}\left[Z\mathbb{I}\{|Z + \ell| > M\}\right] = \mathbb{E}\left[Z\mathbb{I}\{Z > M - \ell\}\right] + \mathbb{E}\left[Z\mathbb{I}\{Z < -M - \ell\}\right].$$

Suppose that $M - \ell \geq 0$. Then, since $Z$ is symmetric random variable (around the origin), $\mathbb{E}\left[Z\mathbb{I}\{Z < -M - \ell\}\right] = -\mathbb{E}\left[Z\mathbb{I}\{Z > M + \ell\}\right]$. Since $M - \ell < M + \ell$, we have that

$$\mathbb{E}\left[Z\mathbb{I}\{|Z + \ell| > M\}\right] = \mathbb{E}\left[Z\mathbb{I}\{Z > M - \ell\}\right] - \mathbb{E}\left[Z\mathbb{I}\{Z > M + \ell\}\right] \geq 0.$$

Finally, suppose that $M - \ell < 0$. Then,

$$\mathbb{E}\left[Z\mathbb{I}\{Z > M - \ell\}\right] = \mathbb{E}\left[Z\mathbb{I}\{0 \geq Z > M - \ell\}\right] + \mathbb{E}\left[Z\mathbb{I}\{Z \geq 0\}\right].$$

Using again the fact that $Z$ is symmetric, we have that

$$\mathbb{E}\left[Z\mathbb{I}\{0 \geq Z > M - \ell\}\right] = -\mathbb{E}\left[Z\mathbb{I}\{0 \leq Z < \ell - M\}\right].$$

Finally, since $\ell - M \leq M + \ell$, we have that

$$\mathbb{E}\left[Z\mathbb{I}\{|Z + \ell| > M\}\right] = \mathbb{E}\left[Z\mathbb{I}\{Z \geq 0\}\right] - \mathbb{E}\left[Z\mathbb{I}\{0 \leq Z < \ell - M\}\right] - \mathbb{E}\left[Z\mathbb{I}\{Z > M + \ell\}\right] \geq 0,$$

completing the proof. ∎

**Lemma B.3** (Norms of Laplace Vectors (Fact C.1 in (Agarwal & Singh, 2017)))**.** *If $Z_1, \ldots, Z_T \sim (\mathrm{Lap}(\lambda))^N$, then*

$$\mathbb{P}(\exists t \in [T] : ||Z_t||_\infty^2 \geq 10\lambda^2 \log^2(NT)) \leq \frac{1}{T}$$

## C. Proof of Lemmas 3.3 and 3.4

### C.1. Proof of Lemma 3.3

Note that the sequence of actions played by Algorithm 1 are completely determined by $I_1, \ldots, I_{\lfloor \frac{T}{\tau} \rfloor}$ in a dataset-independent way. Thus, by post-processing it suffices to show that the actions $I_1, \ldots, I_{\lfloor \frac{T}{\tau} \rfloor}$ are output in a $\varepsilon$-differentially private manner. Note that the distribution over the action $I_1$ is independent of the dataset $\ell_1, \ldots, \ell_T$. Thus, it suffices to only prove privacy with respect to the actions $I_2, \ldots, I_{\lfloor \frac{T}{\tau} \rfloor}$. Consider the sequence of mechanisms $M_2, \ldots, M_{\lfloor \frac{T}{\tau} \rfloor}$, where $M_2 : [K] \times \ell_{1:T} \to \mathbb{R} \times [K]$ is defined as

$$M_2(i_1, \ell_{1:T}) = \left(\hat{\ell}_1(i_1) + Z_1, \mathcal{B}((i_1, \hat{\ell}_1(i_1) + Z_1))\right),$$

for $Z_1 \sim \mathrm{Lap}(\frac{1}{\tau\varepsilon})$ and $M_j : ([K] \times \mathbb{R})^{j-2} \times [K] \times \ell_{1:T} \to \mathbb{R} \times [K]$ is defined as

$$M_j((i_s, r_s)_{s=1}^{j-2}, i_{j-1}, \ell_{1:T}) = \left(\hat{\ell}_{j-1}(i_{j-1}) + Z_{j-1}, \mathcal{B}((i_s, r_s)_{s=1}^{j-2} \circ (i_{j-1}, \hat{\ell}_{j-1}(i_{j-1}) + Z_{j-1}))\right),$$

for $Z_{j-1} \sim \text{Lap}(\frac{1}{\tau\varepsilon})$. Observe that Algorithm 1 is precisely the mechanism $M : \ell_{1:T} \to ([K] \times \mathbb{R})^T$ that adaptively composes $M_2, \ldots, M_{\lfloor \frac{T}{\tau} \rfloor}$. We will now show that $M$ is $\varepsilon$-differentially private by showing that $M(\ell_{1:t})$ and $M(\ell'_{1:t})$ are $\varepsilon$-indistinguishable for arbitrary neighboring datasets $\ell_{1:T}$ and $\ell'_{1:T}$.

Consider two datasets $\ell_{1:T}$ and $\ell'_{1:T}$ that differ in exactly one position. Let $t' \in [T]$ be the index where the two datasets differ. Let $j' \in \{1, \ldots, \lfloor \frac{T}{\tau} \rfloor\}$ be the batch in where the $t'$ lies. That is, let $j' \in \{1, \ldots, \lfloor \frac{T}{\tau} \rfloor\}$ such that $t' \in \{(j'-1)\tau+1, \ldots, j'\tau\}$. Fix outcomes $(i_s, r_s)_{s=1}^{\lfloor \frac{T}{\tau} \rfloor - 2} \in ([K] \times \mathbb{R})^{\lfloor \frac{T}{\tau} \rfloor - 2}$ and $i_{\lfloor \frac{T}{\tau} \rfloor - 1} \in [K]$.

For all $j \leq j'$, we have that the random variables $M_j((i_s, r_s)_{s=1}^{j-2}, i_{j-1}, \ell_{1:T})$ and $M_j((i_s, r_s)_{s=1}^{j-2}, i_{j-1}, \ell'_{1:T})$ are 0-indistinguishable. We now show that the random variables $M_{j'+1}((i_s, r_s)_{s=1}^{j'-1}, i_{j'}, \ell_{1:T})$ and $M_{j'+1}((i_s, r_s)_{s=1}^{j'-1}, i_{j'}, \ell'_{1:T})$ are $\varepsilon$-indistinguishable.

Recall that

$$M_{j'+1}((i_s, r_s)_{s=1}^{j'-1}, i_{j'}, \ell_{1:T}) = \left( \hat{\ell}_{j'}(i_{j'}) + Z_{j'}, \mathcal{B}((i_s, r_s)_{s=1}^{j'-1} \circ (i_{j'}, \hat{\ell}_{j'}(i_{j'}) + Z_{j'})) \right).$$

Note that the query $\hat{\ell}_{j'}(i_{j'})$ has sensitivity at most $\frac{1}{\tau}$. Indeed, we have that

$$\left| \hat{\ell}_{j'}(i_{j'}) - \hat{\ell}'_{j'}(i_{j'}) \right| = \left| \frac{1}{\tau} \sum_{s=(j'-1)\tau+1}^{j'\tau} \ell_s(i_{j'}) - \ell'_s(i_{j'}) \right| = \frac{1}{\tau} \left| \ell_{t'}(i_{j'}) - \ell'_{t'}(i_{j'}) \right| \leq \frac{1}{\tau}.$$

Thus, by Definition A.9 and post-processing, we have that $M_{j'+1}((i_s, r_s)_{s=1}^{j-2}, i_{j-1}, \ell_{1:T})$ and $M_{j'+1}((i_s, r_s)_{s=1}^{j-2}, i_{j-1}, \ell'_{1:T})$ are $\varepsilon$-indistinguishable for all inputs.

To complete the proof, we now show that for all $j > j' + 1$, $M_j((i_s, r_s)_{s=1}^{j-2}, i_{j-1}, \ell_{1:T})$ and $M_j((i_s, r_s)_{s=1}^{j-2}, i_{j-1}, \ell'_{1:T})$ are 0-indistinguishable. Fix some $j > j' + 1$. Recall, that

$$M_j((i_s, r_s)_{s=1}^{j-2}, i_{j-1}, \ell_{1:T}) = \left( \hat{\ell}_{j-1}(i_{j-1}) + Z_{j-1}, \mathcal{B}((i_s, r_s)_{s=1}^{j-2} \circ (i_{j-1}, \hat{\ell}_{j-1}(i_{j-1}) + Z_{j-1})) \right).$$

Since for every $s \in \{(j-1)\tau + 1, \ldots, j\tau\}$ we have that $\ell_s = \ell'_s$, we get that $\hat{\ell}_{j-1}(i_{j-1}) + Z_{j-1}$ and $\hat{\ell}'_{j-1}(i_{j-1}) + Z_{j-1}$ are same in distribution. The same can be said about $\mathcal{B}((i_s, r_s)_{s=1}^{j-2} \circ (i_{j-1}, \hat{\ell}_{j-1}(i_{j-1}) + Z_{j-1}))$ and $\mathcal{B}((i_s, r_s)_{s=1}^{j-2} \circ (i_{j-1}, \hat{\ell}'_{j-1}(i_{j-1}) + Z_{j-1}))$. Accordingly, $M_j((i_s, r_s)_{s=1}^{j-2}, i_{j-1}, \ell_{1:T})$ and $M_j((i_s, r_s)_{s=1}^{j-2}, i_{j-1}, \ell'_{1:T})$ are 0-indistinguishable for all inputs. Since $M$ is the composition of $M_2, \ldots, M_{\lfloor \frac{T}{\tau} \rfloor}$, by basic composition, we have that $M(\ell_{1:T})$ and $M(\ell'_{1:T})$ are $\varepsilon$-indistinguishable, and therefore $M$ is $\varepsilon$-differentially private. This completes the proof.

### C.2. Proof of Lemma 3.4

Let $\ell_1, \ldots, \ell_T$ be any sequence of loss functions. Note that the bandit algorithm $\mathcal{B}$ is evaluated on the loss sequence $\hat{\ell}_1 + Z_1, \ldots, \hat{\ell}_{\lfloor \frac{T}{\tau} \rfloor} + Z_{\lfloor \frac{T}{\tau} \rfloor}$ where $\hat{\ell}_j(i) = \frac{1}{\tau} \sum_{s=(j-1)\tau+1}^{j\tau} \ell_s(i)$ and $Z_j \sim \text{Lap}(\frac{1}{\tau\varepsilon})$. Let $I_1, \ldots, I_{\lfloor \frac{T}{\tau} \rfloor}$ be the random variables denoting the predictions of $\mathcal{B}$ as indicated in Line 4 in Algorithm 1. By definition of $\tilde{R}_{\mathcal{B}}\left(\lfloor \frac{T}{\tau} \rfloor, K, \frac{1}{\tau\varepsilon}\right)$ we get that

$$\mathbb{E}\left[ \sum_{j=1}^{\lfloor \frac{T}{\tau} \rfloor} \hat{\ell}_j(I_j) \right] - \inf_{i \in [K]} \sum_{j=1}^{\lfloor \frac{T}{\tau} \rfloor} \hat{\ell}_j(i) \leq \tilde{R}_{\mathcal{B}}\left(\left\lfloor \frac{T}{\tau} \right\rfloor, K, \frac{1}{\tau\varepsilon}\right).$$

By definition of $\hat{\ell}_s$, we have that

$$\mathbb{E}\left[ \sum_{j=1}^{\lfloor \frac{T}{\tau} \rfloor} \sum_{s=(j-1)\tau+1}^{j\tau} \ell_s(I_j) \right] - \inf_{i \in [K]} \sum_{j=1}^{\lfloor \frac{T}{\tau} \rfloor} \sum_{s=(j-1)\tau+1}^{j\tau} \ell_s(i) \leq \tau \tilde{R}_{\mathcal{B}}\left(\left\lfloor \frac{T}{\tau} \right\rfloor, K, \frac{1}{\tau\varepsilon}\right).$$

Next, note that by construction, we have that for every $j \in \{1, \ldots, \lfloor \frac{T}{\tau} \rfloor\}$ and $s \in \{(j-1)\tau + 1, \ldots, j\tau\}$, we have that $I_s = I_j$. Thus, we can write

$$\mathbb{E}\left[\sum_{j=1}^{\lfloor \frac{T}{\tau} \rfloor} \sum_{s=(j-1)\tau+1}^{j\tau} \ell_s(I_s)\right] - \inf_{i\in[K]} \sum_{j=1}^{\lfloor \frac{T}{\tau} \rfloor} \sum_{s=(j-1)\tau+1}^{j\tau} \ell_s(i) \leq \tau \tilde{\mathrm{R}}_{\mathcal{B}}\left(\left\lfloor \frac{T}{\tau} \right\rfloor, K, \frac{1}{\tau\varepsilon}\right)$$

which further gives

$$\mathbb{E}\left[\sum_{t=1}^{\tau\lfloor \frac{T}{\tau} \rfloor} \ell_t(I_t)\right] - \inf_{i\in[K]} \sum_{t=1}^{\tau\lfloor \frac{T}{\tau} \rfloor} \ell_t(i) \leq \tau \tilde{\mathrm{R}}_{\mathcal{B}}\left(\left\lfloor \frac{T}{\tau} \right\rfloor, K, \frac{1}{\tau\varepsilon}\right).$$

Finally, the expected regret for rounds $\tau \left\lfloor \frac{T}{\tau} \right\rfloor + 1, \ldots, T$ can be bounded above by $\tau$. Thus, overall, we have that

$$\mathbb{E}\left[\sum_{t=1}^{T} \ell_t(I_t)\right] - \inf_{i\in[K]} \sum_{t=1}^{T} \ell_t(i) \leq \tau \tilde{\mathrm{R}}_{\mathcal{B}}\left(\left\lfloor \frac{T}{\tau} \right\rfloor, K, \frac{1}{\tau\varepsilon}\right) + \tau \leq \tau \tilde{\mathrm{R}}_{\mathcal{B}}\left(\frac{T}{\tau}, K, \frac{1}{\tau\varepsilon}\right) + \tau.$$

Noting that $\ell_1, \ldots, \ell_T$ was arbitrary completes the proof.

## D. Proof of Corollary 3.2

The following Theorem from (Huang et al., 2022) will be useful.

**Theorem D.1** (Theorem 4.1 in (Huang et al., 2022)). *Let $\tilde{\ell}_1, \ldots, \tilde{\ell}_T$ be any sequence of random loss functions that satisfy the following two properties: (1) for every $i \in [K]$ and $t \in [T]$, the random variable $\tilde{\ell}_t(i)$ is truncated non-negative and (2) for every $i \in [K]$ and $t \in [T]$, the random variable $\tilde{\ell}_t(i)$ is heavy-tailed with parameters $\alpha \in (1, 2]$ and $\sigma > 0$. Then, the expected regret of* HTINF *(Algorithm 1 in (Huang et al., 2022)) when run on $\tilde{\ell}_1, \ldots, \tilde{\ell}_T$ is at most $30\sigma K^{1-\frac{1}{\alpha}}(T+1)^{\frac{1}{\alpha}}$.*

We now make precise the definition of truncated non-negativity and heavy-tails.

**Definition D.2** (Truncated Non-negativity). A random variable $X$ is truncated non-negative if for every $M \geq 0$, we have that $\mathbb{E}\left[X \cdot \mathbb{I}\{|X| > M\}\right] \geq 0$.

In Appendix B, we prove that random losses of the form $\tilde{\ell}(i) = \ell(i) + Z_i$ are truncated non-negative when $\ell(i) \in [0, 1]$ and $Z_i \sim \mathrm{Lap}(\lambda)$.

**Definition D.3** (($\alpha, \sigma$)-Heavy-tailed loss). A random loss $\tilde{\ell}(i)$ is $(\alpha, \sigma)$-heavy tailed if $\mathbb{E}\left[|\tilde{\ell}(i)|^\alpha\right] \leq \sigma^\alpha$.

In addition, if $\tilde{\ell}(i) = \ell(i) + Z_i$, where $\ell(i) \in [0, 1]$ and $Z_i \sim \mathrm{Lap}(\lambda)$, then $\tilde{\ell}(i)$ is $(2, \sqrt{2 + 4\lambda^2})$-heavy tailed. We are now ready to prove Corollary 3.2.

*Proof.* (of Corollary 3.2) In order to use Theorem 3.1, we need to upper bound $\tilde{\mathrm{R}}_{\mathrm{HTINF}}(T, \lambda)$. Let $\ell_1, \ldots, \ell_T$ be any sequence of loss functions such that $\ell_t : [K] \to [0, 1]$ and let $\tilde{\ell}_1, \ldots, \tilde{\ell}_T$ be such that $\tilde{\ell}_t(i) = \ell_t(i) + Z_t(i)$ where $Z_t(i) \sim \mathrm{Lap}(\lambda)$. Then, since for every $t \in [T]$ and $i \in [K]$, we have that $\tilde{\ell}_t(i)$ is truncated non-negative and $(2, \sqrt{2 + 4\lambda^2})$-heavy tailed, Theorem D.1 implies that

$$\tilde{\mathrm{R}}_{\mathrm{HTINF}}(T, K, \lambda) \leq 30\sqrt{(2 + 4\lambda^2)K(T+1)}.$$

Finally, to get Corollary 3.2, we just upper bound

$$\frac{2}{\varepsilon}\tilde{\mathrm{R}}_{\mathrm{HTINF}}(\varepsilon T, K, 1) + \frac{2}{\varepsilon} \leq 208\frac{\sqrt{TK}}{\sqrt{\varepsilon}} + \frac{2}{\varepsilon},$$

for $\varepsilon \geq \frac{1}{T}$. ∎

---

**Algorithm 3** EXP3 with Mixing

---

1: **Input:** Action space $[K]$, learning rate $\eta$, mixing parameter $\gamma > 0$
2: **Initialize:** $w_1(i) = 1$ for all $i \in [K]$
3: **for** $t = 1, \ldots, T$ **do**
4:     Set $P_t(i) = (1 - \gamma)\frac{w_t(i)}{\sum_{i \in [K]} w_t(i)} + \frac{\gamma}{K}$
5:     Draw $I_t \sim P_t$
6:     Observe loss $\ell_t(I_t)$ and construct unbiased estimator $\hat{\ell}_t(i) = \frac{\ell_t(i)\mathbb{I}\{I_t = i\}}{P_t(i)}$
7:     Update $w_{t+1}(i) \leftarrow w_t(i) \cdot \exp\{-\eta\hat{\ell}_t(i)\}$ for all $i \in [K]$
8: **end for**

---

# E. Additional Upper Bounds for Private Adversarial Bandits

In this section, we instantiate Theorem 3.1 with other (non-private) bandit algorithms to obtain two other regret upper bounds.

## E.1. EXP3 Conversion

Corollary E.1 follows by letting $\mathcal{B}$ in Theorem 3.1 be the classical EXP3 algorithm (Auer et al., 2002). See Appendix D for the pseudocode of EXP3.

**Corollary E.1** (EXP3 Conversion). *For every $\varepsilon \leq 1$, if $\mathcal{B}$ is EXP3 run with learning rate*

$$\eta = \sqrt{\frac{\log(K)}{22\,\varepsilon KT \log^2(\varepsilon KT)}}$$

*and mixing parameter $\gamma = 4\eta K \log(\varepsilon KT)$, then Algorithm 1, when run with $\mathcal{B}$ and $\tau = \lceil\frac{1}{\varepsilon}\rceil$, is $\varepsilon$-differentially private and suffers worst-case expected regret at most*

$$\frac{36\sqrt{TK\log(K)}\log(KT)}{\sqrt{\varepsilon}} + \frac{4}{\varepsilon}.$$

Algorithm 3 provides the pseudocode for the version of EXP3 that we consider.

The following lemma about EXP3 will be useful when proving Corollary E.1.

**Lemma E.2** ((Auer et al., 2002; Bubeck et al., 2012)). *For any sequence of loss functions $\ell_1, \ldots, \ell_T$, where $\ell_t : [K] \to \mathbb{R}$, if $\eta > 0$ is such that $\eta \max_{i \in [K]} -\hat{\ell}_t(i) \leq 1$ for all $t \in [T]$, then EXP3 when run on $\ell_1, \ldots, \ell_T$ outputs distributions $P_{1:T} \in \Pi([K])^T$ such that*

$$\mathbb{E}\left[\sum_{t=1}^{T}\sum_{i=1}^{K} P_t(i)\ell_t(i)\right] \leq \inf_{i \in [K]} \sum_{t=1}^{T} \ell_t(i) + 2\gamma T + \frac{\log(K)}{\eta} + \eta \sum_{t=1}^{T}\sum_{i=1}^{K} \ell_t(i)^2,$$

*where $\hat{\ell}_t$ is the unbiased estimate of the true loss $\ell_t$ that EXP3 computes in Line 6 of Algorithm 3 and the expectation is taken only with respect to the randomness of EXP3.*

*Proof.* (of Corollary E.1) In order to use Theorem 3.1, we first need to bound $\tilde{R}_{\text{EXP3}}(T, K, \lambda)$. Let $\ell_1, \ldots, \ell_T$ be any sequence of loss functions such that $\ell_t : [K] \to [0,1]$ and let $\tilde{\ell}_1, \ldots, \tilde{\ell}_T$ be such that $\tilde{\ell}_t(i) = \ell_t(i) + Z_t(i)$ where $Z_t(i) \sim \text{Lap}(\lambda)$. Let $E$ be the event that there exists a $t \in [T]$ such that $\max_{i \in [K]} |Z_t(i)|^2 \geq 10\lambda^2 \log^2 KT$. Then, Lemma B.3 shows that $\mathbb{P}[E] \leq \frac{1}{T}$. Moreover, note that $\mathbb{E}[Z_t(i)|E^c] = 0$ for all $i \in [K]$ and $t \in [T]$. Let $\mathcal{A}$ be the random variable denoting the internal randomness of EXP3. We need to bound

$$\tilde{R}_{\text{EXP3}}(T, K, \lambda) = \mathbb{E}_{\mathcal{A}, Z_{1:T}}\left[\sum_{t=1}^{T} \ell_t(\text{EXP3}(\tilde{\mathcal{H}}_t)) - \inf_{i \in [K]} \sum_{t=1}^{T} \ell_t(i)\right].$$

We can write $\tilde{R}_{\text{EXP3}}(T, K, \lambda)$ as

$$\mathop{\mathbb{E}}_{\mathcal{A}, Z_{1:T}}\left[\sum_{t=1}^{T}\ell_t(\text{EXP3}(\tilde{\mathcal{H}}_t)) - \inf_{i\in[K]}\sum_{t=1}^{T}\ell_t(i)\Big| E\right]\mathbb{P}(E) + \mathop{\mathbb{E}}_{\mathcal{A}, Z_{1:T}}\left[\sum_{t=1}^{T}\ell_t(\text{EXP3}(\tilde{\mathcal{H}}_t)) - \inf_{i\in[K]}\sum_{t=1}^{T}\ell_t(i)\Big| E^c\right]\mathbb{P}(E^c)$$

Since $\mathbb{E}_{\mathcal{A}, Z_{1:T}}\left[\sum_{t=1}^{T}\ell_t(\text{EXP3}(\tilde{\mathcal{H}}_t)) - \inf_{i\in[K]}\sum_{t=1}^{T}\ell_t(i)\Big| E\right] \leq T$, we have that

$$\tilde{R}_{\text{EXP3}}(T, K, \lambda) \leq \mathop{\mathbb{E}}_{\mathcal{A}, Z_{1:T}}\left[\sum_{t=1}^{T}\ell_t(\text{EXP3}(\tilde{\mathcal{H}}_t)) - \inf_{i\in[K]}\sum_{t=1}^{T}\ell_t(i)\Big| E^c\right] + 1$$

We now want to use Lemma E.2 to bound $\mathbb{E}_{\mathcal{A}, Z_{1:T}}\left[\sum_{t=1}^{T}\ell_t(\text{EXP3}(\tilde{\mathcal{H}}_t)) - \inf_{i\in[K]}\sum_{t=1}^{T}\ell_t(i)\Big| E^c\right]$. Recall, that EXP3 is actually running on the noisy losses $\tilde{\ell}_1, \ldots, \tilde{\ell}_T$. So, in order to use Lemma E.2, we need to pick $\gamma, \eta > 0$ such that $\eta \max_{i\in[K]} -\hat{\tilde{\ell}}_t(i) \leq 1$, where we use $\hat{\tilde{\ell}}_t$ to denote the unbiased estimate that EXP3 constructs of the true (noisy) loss $\tilde{\ell}_t$. In particular, recall that EXP3 constructs $\hat{\tilde{\ell}}_t(i) = \frac{\tilde{\ell}(i)\mathbb{I}\{I_t=i\}}{P_t(i)}$ where we used $P_t(i)$ to denote the measure that EXP3 uses to select its action $I_t$ on round $t \in [T]$. Moreover, given a mixing parameter $\gamma > 0$, we have that $P_t(i) \geq \frac{\gamma}{K}$. Thus, we need to pick $\gamma$ and $\eta$ such that

$$\eta \max_{i\in[K]} -\hat{\tilde{\ell}}_t(i) \leq \frac{\eta K}{\gamma}\max_{i\in[K]}|Z_t(i)| \leq 1.$$

Conditioned on event $E^c$, we have that $\max_{i\in[K]}|Z_t(i)| \leq 4\lambda\log(KT)$. Thus, it suffices to pick $\gamma = 4\eta\lambda K\log(KT)$. Then, conditioned on the event $E^c$ and the random variables $Z_1, \ldots, Z_T$, we can use Lemma E.2 to get that

$$\mathop{\mathbb{E}}_{\mathcal{A}}\left[\sum_{t=1}^{T}\sum_{i=1}^{K}P_t(i)\tilde{\ell}_t(i)\Big| E^c, Z_{1:T}\right] \leq \inf_{i\in[K]}\sum_{t=1}^{T}\tilde{\ell}_t(i) + 2\gamma T + \frac{\log(K)}{\eta} + \eta\sum_{t=1}^{T}\sum_{i=1}^{K}\tilde{\ell}_t(i)^2.$$

Taking an outer expectation, then gives that

$$\mathop{\mathbb{E}}_{\mathcal{A}, Z_{1:T}}\left[\sum_{t=1}^{T}\sum_{i=1}^{K}P_t(i)\tilde{\ell}_t(i)\Big| E^c\right] \leq \inf_{i\in[K]}\mathop{\mathbb{E}}_{Z_{1:T}}\left[\sum_{t=1}^{T}\tilde{\ell}_t(i)\Big| E^c\right] + 2\gamma T + \frac{\log(K)}{\eta} + \eta\mathop{\mathbb{E}}_{Z_{1:T}}\left[\sum_{t=1}^{T}\sum_{i=1}^{K}\tilde{\ell}_t(i)^2\Big| E^c\right].$$

Since $Z_t(i)$, conditioned on $E^c$, is zero-mean and $Z_t(i)$ conditioned on the history $\tilde{\mathcal{H}}_t$ is independent of $P_t(i)$, we have that

$$\mathop{\mathbb{E}}_{\mathcal{A}, Z_{1:T}}\left[\sum_{t=1}^{T}\sum_{i=1}^{K}P_t(i)\ell_t(i)\Big| E^c\right] \leq \inf_{i\in[K]}\sum_{t=1}^{T}\ell_t(i) + 2\gamma T + \frac{\log(K)}{\eta} + \eta\mathop{\mathbb{E}}_{Z_{1:T}}\left[\sum_{t=1}^{T}\sum_{i=1}^{K}\tilde{\ell}_t(i)^2\Big| E^c\right],$$

which further gives

$$\tilde{R}_{\text{EXP3}}(T, K, \lambda) \leq 2\gamma T + \frac{\log(K)}{\eta} + \eta\mathop{\mathbb{E}}_{Z_{1:T}}\left[\sum_{t=1}^{T}\sum_{i=1}^{K}\tilde{\ell}_t(i)^2\Big| E^c\right] + 1.$$

It just remains to bound $\mathbb{E}_{Z_{1:T}}\left[\sum_{t=1}^{T}\sum_{i=1}^{K}\tilde{\ell}_t(i)^2\Big| E^c\right]$. Note that we can write

$$\eta \mathop{\mathbb{E}}_{Z_{1:T}} \left[ \sum_{t=1}^{T} \sum_{i=1}^{K} \tilde{\ell}_t(i)^2 \Bigg| E^c \right] \leq \eta K \mathop{\mathbb{E}}_{Z_{1:T}} \left[ \sum_{t=1}^{T} \max_{i \in [K]} \tilde{\ell}_t(i)^2 \Bigg| E^c \right]$$

$$\leq \eta K \mathop{\mathbb{E}}_{Z_{1:T}} \left[ \sum_{t=1}^{T} \max_{i \in [K]} (\ell_t(i) + Z_t(i))^2 \Bigg| E^c \right]$$

$$\leq 2\eta K \mathop{\mathbb{E}}_{Z_{1:T}} \left[ \sum_{t=1}^{T} (1 + \max_{i \in [K]} Z_t(i)^2) \Bigg| E^c \right]$$

$$\leq 2\eta K \sum_{t=1}^{T} (1 + 10\lambda^2 \log^2 KT)$$

$$= 2\eta T K (1 + 10\lambda^2 \log^2 KT).$$

Plugging this bound back in gives that

$$\tilde{R}_{\text{EXP3}}(T, K, \lambda) \leq 2\gamma T + \frac{\log(K)}{\eta} + 2\eta T K (1 + 10\lambda^2 \log^2 KT) + 1.$$

Recall that we picked $\gamma = 4\eta \lambda K \log(KT)$. Substituting this selection gives

$$\tilde{R}_{\text{EXP3}}(T, K, \lambda) \leq 8\eta \lambda K T \log(KT) + \frac{\log(K)}{\eta} + 2\eta T K (1 + 10\lambda^2 \log^2 KT) + 1.$$

We can then write

$$\tilde{R}_{\text{EXP3}}(T, K, \lambda) \leq \frac{\log(K)}{\eta} + 2\eta T K (1 + 10 \max\{\lambda^2, \lambda\} \log^2 KT) + 1.$$

Picking $\eta = \sqrt{\frac{\log(K)}{2TK(1 + 10 \max\{\lambda^2, \lambda\} \log^2 KT)}}$, we get overall that

$$\tilde{R}_{\text{EXP3}}(T, K, \lambda) \leq 2\sqrt{2TK \log(K)(1 + 10 \max\{\lambda^2, \lambda\} \log^2 KT)} + 1.$$

Finally, Corollary E.1 follows by the fact that

$$\frac{2}{\varepsilon} \tilde{R}_{\text{EXP3}}(\varepsilon T, K, 1) + \frac{2}{\varepsilon} \leq 36 \frac{\sqrt{TK \log(K)} \log(KT)}{\sqrt{\varepsilon}} + \frac{4}{\varepsilon}.$$

This completes the proof. ∎

## E.2. FTPL Conversion

Corollary E.3 follows by using Follow-the-Perturbed-Leader (FTPL) with Geometric Resampling (Neu & Bartók, 2016). The pseudocode for FTPL with Geometric Resampling is provided in Algorithm 4.

**Corollary E.3** (FTPL Conversion). *For every $\varepsilon \in [\frac{1}{T}, 1]$, if $\mathcal{B}$ is* FTPL *with perturbation distribution* $\text{Lap}\left(\frac{1}{\eta}\right)$ *and Geometric Resampling threshold $M$ (see Algorithm 4), where $M = \sqrt{\varepsilon KT}$ and*

$$\eta = \min \left\{ \sqrt{\frac{\log(K)}{(\varepsilon KT + 10\varepsilon KT \log^2(\varepsilon KT))}}, \frac{1}{M(1 + 4\log(\varepsilon T))} \right\},$$

---

**Algorithm 4** Bandit FTPL with Geometric Resampling (Neu & Bartók, 2016)

1: **Input:** $M, \eta$
2: **Initialize:** $\hat{L}_0(i) = 0$ for all $i \in [K]$
3: **for** $t = 1, \ldots, T$ **do**
4:     Sample $Z_1, \ldots, Z_K$ i.i.d. from $\mathrm{Lap}(0, \frac{1}{\eta})$.
5:     Select action $I_t \in \arg\min_{i \in [K]}(\hat{L}_{t-1}(i) + Z_i)$
6:     Observe loss $\ell_t(I_t)$
7:     Let $M_t = 0$.
8:     **for** $i = 1, 2, \ldots, M$ **do**
9:         Sample $Z_1', \ldots, Z_K'$ i.i.d. from $\mathrm{Lap}(0, \frac{1}{\eta})$.
10:        **if** $I_t \in \arg\max_{i \in [K]}(\hat{L}_{t-1}(i) + Z_i')$ **then**
11:           Set $M_t = i$.
12:           **break**
13:        **end if**
14:     **end for**
15:     Define $\hat{\ell}_t(i) = \ell_t(i) M_t \mathbb{I}\{I_t = i\}$.
16:     Update $\hat{L}_t = \hat{L}_{t-1} + \hat{\ell}_t(i)$.
17: **end for**

---

*Algorithm 1, when run with $\mathcal{B}$ and $\tau = \lceil \frac{1}{\varepsilon} \rceil$, is $\varepsilon$-differentially private and suffers worse-case expected regret at most*

$$32 \frac{\sqrt{KT} \log(K) \log(KT)}{\sqrt{\varepsilon}} + \frac{2}{\varepsilon}.$$

We now prove Corollary E.3. Lemma E.4 first bounds $\tilde{R}_{\mathcal{B}}(T, K, \lambda)$ when $\mathcal{B}$ is Algorithm 4.

**Lemma E.4.** *Let $\mathcal{B}$ denote Algorithm 4. Then, if $M = \sqrt{KT}$ and*

$$\eta = \min\left\{ \sqrt{\frac{\log(K)}{(KT + 10KT\lambda^2 \log^2(KT))}}, \frac{1}{M(1 + 4\lambda \log(T))} \right\},$$

*we have that*

$$\tilde{R}_{\mathcal{B}}(T, K, \lambda) \leq 11\lambda\sqrt{KT} \log(K) \log(KT) + 10\sqrt{KT}$$

*Proof.* Let $\ell_1, \ldots, \ell_T$ be any sequence of loss functions such that $\ell_t : [K] \to [0, 1]$ and let $\tilde{\ell}_1, \ldots, \tilde{\ell}_T$ be such that $\tilde{\ell}_t(i) = \ell_t(i) + G_t(i)$ where $G_t(i) \sim \mathrm{Lap}(\lambda)$. Let $E$ be the event that there exists a $t \in [T]$ such that $\max_{i \in [K]} |G_t(i)|^2 \geq 10\lambda^2 \log^2 KT$. Then, Lemma B.3 shows that $\mathbb{P}[E] \leq \frac{1}{T}$. Moreover, note that $\mathbb{E}[G_t(i)|E^c] = 0$ for all $i \in [K]$ and $t \in [T]$. We need to bound

$$\tilde{R}_{\mathcal{B}}(T, K, \lambda) = \mathop{\mathbb{E}}_{\mathcal{B}, G_{1:T}} \left[ \sum_{t=1}^{T} \ell_t(\mathcal{B}(\tilde{\mathcal{H}}_t)) - \inf_{i \in [K]} \sum_{t=1}^{T} \ell_t(i) \right].$$

We can write $\tilde{R}_{\mathcal{B}}(T, K, \lambda)$ as

$$\mathop{\mathbb{E}}_{\mathcal{B}, G_{1:T}} \left[ \sum_{t=1}^{T} \ell_t(\mathcal{B}(\tilde{\mathcal{H}}_t)) - \inf_{i \in [K]} \sum_{t=1}^{T} \ell_t(i) \bigg| E \right] \mathbb{P}(E) + \mathop{\mathbb{E}}_{\mathcal{B}, G_{1:T}} \left[ \sum_{t=1}^{T} \ell_t(\mathcal{B}(\tilde{\mathcal{H}}_t)) - \inf_{i \in [K]} \sum_{t=1}^{T} \ell_t(i) \bigg| E^c \right] \mathbb{P}(E^c)$$

Since $\mathbb{E}_{\mathcal{B}, G_{1:T}} \left[ \sum_{t=1}^{T} \ell_t(\mathcal{B}(\tilde{\mathcal{H}}_t)) - \inf_{i \in [K]} \sum_{t=1}^{T} \ell_t(i) \big| E \right] \leq T$, we have that

$$\tilde{\mathrm{R}}_{\mathcal{B}}(T, K, \lambda) \leq \mathop{\mathbb{E}}_{\mathcal{B}, G_{1:T}} \left[ \sum_{t=1}^{T} \ell_t(\mathcal{B}(\tilde{\mathcal{H}}_t)) - \inf_{i \in [K]} \sum_{t=1}^{T} \ell_t(i) \middle| E^c \right] + 1$$

$$\leq \mathop{\mathbb{E}}_{\mathcal{B}, G_{1:T}} \left[ \sum_{t=1}^{T} \ell_t(\mathcal{B}(\tilde{\mathcal{H}}_t)) \middle| E^c \right] - \inf_{i \in [K]} \sum_{t=1}^{T} \ell_t(i) + 1$$

Let $i^\star$ be the arm that minimizes $\sum_{t=1}^{T} \ell_t(i)$. Moreover, let $\hat{\tilde{\ell}}_t$ denote the unbiased estimate that Algorithm 4 constructs of the true (noisy) loss $\tilde{\ell}_t$ when run on the noisy losses $\tilde{\ell}_1, \ldots, \tilde{\ell}_T$. We start with the following regret decomposition for FTPL from (**?**)Lemma 3]honda2023follow.

$$\mathop{\mathbb{E}}_{\mathcal{B}, G_{1:T}} \left[ \sum_{t=1}^{T} \hat{\tilde{\ell}}_t(I_t) \middle| E^c \right] - \mathop{\mathbb{E}}_{\mathcal{B}, G_{1:T}} \left[ \sum_{t=1}^{T} \hat{\tilde{\ell}}_t(i^\star) \middle| E^c \right] \leq 2 \mathop{\mathbb{E}}_{Z \sim \mathrm{Lap}(\frac{1}{\eta})^K} \left[ \max_{i \in [K]} |Z_i| \right] + \mathop{\mathbb{E}}_{\mathcal{B}, G_{1:T}} \left[ \sum_{t=1}^{T} \sum_{i=1}^{K} \hat{\tilde{\ell}}_t(i)(P_t(i) - P_{t+1}(i)) \middle| E^c \right],$$

where we define $P_t(i) := \mathbb{P}\left[ I_t = i \middle| \hat{\tilde{\ell}}_1, \ldots, \hat{\tilde{\ell}}_{t-1} \right]$. The first term on the right can be bounded as

$$2 \mathop{\mathbb{E}}_{Z \sim \mathrm{Lap}(\frac{1}{\eta})^K} \left[ \max_{i \in [K]} |Z_i| \right] \leq \frac{6 \log(K)}{\eta}.$$

As for the second term, Lemma 5 from ([Cheng et al.](#)) gives that

$$\exp\{-\eta \|\hat{\tilde{\ell}}_t\|_1\} \leq \frac{P_{t+1}(i)}{P_t(i)} \leq \exp\{\eta \|\hat{\tilde{\ell}}_t\|_1\}.$$

Accordingly, we have that

$$P_t(i)(1 - \exp\{\eta \|\hat{\tilde{\ell}}_t\|_1\}) \leq P_t(i) - P_{t+1}(i) \leq P_t(i)(1 - \exp\{-\eta \|\hat{\tilde{\ell}}_t\|_1\}).$$

Thus, we can bound

$$\hat{\tilde{\ell}}_t(i)(P_t(i) - P_{t+1}(i)) \leq \hat{\tilde{\ell}}_t(i) P_t(i)(\exp\{\eta \|\hat{\tilde{\ell}}_t\|_1\} - 1).$$

For $\eta > 0$ such that $\eta \|\hat{\tilde{\ell}}_t\|_1 \leq 1$, we have that

$$\exp\{\eta \|\hat{\tilde{\ell}}_t\|_1\} \leq 2\eta \|\hat{\tilde{\ell}}_t\|_1 + 1.$$

Since $\|\hat{\tilde{\ell}}_t\|_1 \leq |M_t(\ell_t(I_t) + G_t(I_t))| \leq M(1 + 4\eta \log(T))$, it suffices to pick $\eta \leq \frac{1}{M(1 + 4\lambda \log(T))}$. For this choice of $\eta$, we have that

$$\hat{\tilde{\ell}}_t(i)(P_t(i) - P_{t+1}(i)) \leq 2 P_t(i) \eta \hat{\tilde{\ell}}_t(i) \|\hat{\tilde{\ell}}_t\|_1 \leq 2 P_t(i) \eta (\hat{\tilde{\ell}}_t(i))^2.$$

Plugging this in gives

$$\mathop{\mathbb{E}}_{\mathcal{B}, G_{1:T}} \left[ \sum_{t=1}^{T} \sum_{i=1}^{K} \hat{\tilde{\ell}}_t(i)(P_t(i) - P_{t+1}(i)) \middle| E^c \right] \leq 2\eta \mathop{\mathbb{E}}_{\mathcal{B}, G_{1:T}} \left[ \sum_{t=1}^{T} \sum_{i=1}^{K} P_t(i)(\hat{\tilde{\ell}}_t(i))^2 \middle| E^c \right]$$

and therefore

$$\mathop{\mathbb{E}}_{\mathcal{B},G_{1:T}}\left[\sum_{t=1}^{T}\hat{\tilde{\ell}}_t(I_t) - \hat{\tilde{\ell}}_t(i^\star)\bigg| E^c\right] \le \frac{6\log(K)}{\eta} + 2\eta\mathop{\mathbb{E}}_{\mathcal{B},G_{1:T}}\left[\sum_{t=1}^{T}\sum_{i=1}^{K}P_t(i)(\hat{\tilde{\ell}}_t(i))^2\bigg|E^c\right].$$

To bound the second term on the right hand side, we have that

$$
\begin{aligned}
\mathop{\mathbb{E}}_{\mathcal{B},G_{1:T}}\left[\sum_{t=1}^{T}\sum_{i=1}^{K}P_t(i)(\hat{\tilde{\ell}}_t(i))^2\bigg|E^c\right] &= \mathop{\mathbb{E}}_{\mathcal{B},G_{1:T}}\left[\sum_{t=1}^{T}\sum_{i=1}^{K}P_t(i)(\tilde{\ell}_t(i))^2\mathbb{I}\{I_t=i\}(M_t)^2\bigg|E^c\right] \\
&\le 2\mathop{\mathbb{E}}_{\mathcal{B},G_{1:T}}\left[\sum_{t=1}^{T}\sum_{i=1}^{K}P_t(i)(\tilde{\ell}_t(i))^2\mathbb{I}\{I_t=i\}\frac{1}{(P_t(i))^2}\bigg|E^c\right] \\
&= 2\mathop{\mathbb{E}}_{\mathcal{B},G_{1:T}}\left[\sum_{t=1}^{T}\sum_{i=1}^{K}(\tilde{\ell}_t(i))^2\mathbb{I}\{I_t=i\}\frac{1}{P_t(i)}\bigg|E^c\right] \\
&= 2\mathop{\mathbb{E}}_{G_{1:T}}\left[\sum_{t=1}^{T}\sum_{i=1}^{K}(\ell_t(i)+G_t(i))^2\bigg|E^c\right] \\
&\le 2\mathop{\mathbb{E}}_{G_{1:T}}\left[\sum_{t=1}^{T}\sum_{i=1}^{K}(1+G_t(i)^2)\bigg|E^c\right] \\
&= 2KT + 20KT\lambda^2\log^2(KT),
\end{aligned}
$$

where the first inequality follows from Lemma 12 in (Cheng et al.). Thus,

$$\mathop{\mathbb{E}}_{\mathcal{B},G_{1:T}}\left[\sum_{t=1}^{T}\hat{\tilde{\ell}}_t(I_t) - \hat{\tilde{\ell}}_t(i^\star)\bigg|E^c\right] \le \frac{6\log(K)}{\eta} + 4\eta KT + 40\eta KT\lambda^2\log^2(KT).$$

Next, note that

$$
\begin{aligned}
\mathop{\mathbb{E}}_{\mathcal{B},G_{1:T}}\left[\sum_{t=1}^{T}\tilde{\ell}_t(I_t) - \tilde{\ell}_t(i^\star)\bigg|E^c\right] &= \mathop{\mathbb{E}}_{\mathcal{B},G_{1:T}}\left[\sum_{t=1}^{T}\hat{\tilde{\ell}}_t(I_t) - \hat{\tilde{\ell}}_t(i^\star)\bigg|E^c\right] + \mathop{\mathbb{E}}_{\mathcal{B},G_{1:T}}\left[\sum_{t=1}^{T}\tilde{\ell}_t(I_t) - \hat{\tilde{\ell}}_t(I_t)\bigg|E^c\right] \\
&\quad + \mathop{\mathbb{E}}_{\mathcal{B},G_{1:T}}\left[\sum_{t=1}^{T}\hat{\tilde{\ell}}_t(i^\star) - \tilde{\ell}_t(i^\star)\bigg|E^c\right].
\end{aligned}
$$

Thus, it suffices to upper bound the latter two terms. Starting with the third term, we have that

$$
\begin{aligned}
\mathop{\mathbb{E}}_{\mathcal{B},G_{1:T}}\left[\sum_{t=1}^{T}\hat{\tilde{\ell}}_t(i^\star)-\tilde{\ell}_t(i^\star)\,\Big|\,E^c\right] &= \mathop{\mathbb{E}}_{\mathcal{B},G_{1:T}}\left[\sum_{t=1}^{T}\tilde{\ell}_t(i^\star)(1-(1-P_t(i^\star))^M)-\tilde{\ell}_t(i^\star)\,\Big|\,E^c\right]\\
&= \mathop{\mathbb{E}}_{\mathcal{B},G_{1:T}}\left[\sum_{t=1}^{T}\tilde{\ell}_t(i^\star)-\tilde{\ell}_t(i^\star)(1-P_t(i^\star))^M-\tilde{\ell}_t(i^\star)\,\Big|\,E^c\right]\\
&= \mathop{\mathbb{E}}_{\mathcal{B},G_{1:T}}\left[\sum_{t=1}^{T}-\tilde{\ell}_t(i^\star)(1-P_t(i^\star))^M\,\Big|\,E^c\right]\\
&= \mathop{\mathbb{E}}_{\mathcal{B},G_{1:T}}\left[\sum_{t=1}^{T}-(\ell_t(i^\star)+G_t(i))(1-P_t(i^\star))^M\,\Big|\,E^c\right]\\
&= \mathop{\mathbb{E}}_{\mathcal{B},G_{1:T}}\left[\sum_{t=1}^{T}-\ell_t(i^\star)(1-P_t(i^\star))^M\,\Big|\,E^c\right]\\
&\leq 0,
\end{aligned}
$$

where the second equality follows by Lemma 4 from (Neu & Bartók, 2016). Now, for the second term, by Lemma 5 from (Neu & Bartók, 2016) we have that

$$
\mathop{\mathbb{E}}_{\mathcal{B},G_{1:T}}\left[\sum_{t=1}^{T}\tilde{\ell}_t(I_t)-\hat{\tilde{\ell}}_t(I_t)\,\Big|\,E^c\right]\leq\frac{KT}{eM}.
$$

Combining all our bounds gives

$$
\mathop{\mathbb{E}}_{\mathcal{B},G_{1:T}}\left[\sum_{t=1}^{T}\tilde{\ell}_t(I_t)-\tilde{\ell}_t(i^\star)\,\Big|\,E^c\right]\leq\frac{6\log(K)}{\eta}+4\eta(KT+10KT\lambda^2\log^2(KT))+\frac{KT}{eM}.
$$

For $M=\sqrt{KT}$ and $\eta=\min\left\{\sqrt{\frac{\log(K)}{(KT+10KT\lambda^2\log^2(KT))}},\frac{1}{\sqrt{KT}(1+4\lambda\log(T))}\right\}$, we get that

$$
\mathop{\mathbb{E}}_{\mathcal{B},G_{1:T}}\left[\sum_{t=1}^{T}\tilde{\ell}_t(I_t)-\tilde{\ell}_t(i^\star)\,\Big|\,E^c\right]\leq 10\lambda\sqrt{KT}\log(K)\log(KT)+10\sqrt{KT}.
$$

Since $\mathbb{E}_{\mathcal{B},G_{1:T}}\left[\sum_{t=1}^{T}\tilde{\ell}_t(I_t)-\tilde{\ell}_t(i^\star)\,\Big|\,E^c\right]=\mathbb{E}_{\mathcal{B},G_{1:T}}\left[\sum_{t=1}^{T}\ell_t(I_t)-\ell_t(i^\star)\,\Big|\,E^c\right]$, we have that

$$
\tilde{\mathrm{R}}_{\mathcal{B}}(T,K,\lambda)\leq 10\lambda\sqrt{KT}\log(K)\log(KT)+10\sqrt{KT}+1,
$$

which completes the proof. ∎

Equipped with Lemma E.4, we are now ready to prove Corollary E.3.

*Proof.* (of Corollary E.3) Let $\mathcal{B}$ be Algorithm 4 with the hyperparameters selected according to Lemma E.4. Then, we know that

$$
\tilde{\mathrm{R}}_{\mathcal{B}}(T,K,\lambda)\leq 11\lambda\sqrt{KT}\log(K)\log(KT)+10\sqrt{KT}.
$$

By Theorem 3.1, we can convert $\mathcal{B}$ into an $\varepsilon$-differentially private algorithm $\mathcal{A}$ such that

---

**Algorithm 5** Bandit to Bandit with Expert Advice

---

1: **Input:** Bandit algorithm $\mathcal{B}$, Number of experts $N$, Action space $[K]$
2: **Initialize:** $\mathcal{B}$ with action space $[N]$
3: **for** $t = 1, \ldots, T$ **do**
4:      Receive expert predictions $\mu_t^1, \ldots, \mu_t^N \in \Pi([K])^N$
5:      Sample $I_t^i \sim \mu_t^j$ for all $j \in [N]$
6:      Define $\tilde{\ell}_t(j) := \ell_t(I_t^j)$ for all $j \in [N]$
7:      Receive expert $J_t \in [N]$ from $\mathcal{B}$
8:      Play action $I_t^{J_t} \in [K]$ and observe loss $\ell_t(I_t^{J_t})$
9:      Pass $\tilde{\ell}_t(J_t)$ to $\mathcal{B}$
10: **end for**

---

$$
\begin{aligned}
\mathrm{R}_{\mathcal{A}}(T, K) &\leq \frac{2}{\varepsilon} \tilde{\mathrm{R}}_{\mathcal{B}}(\varepsilon T, K, 1) + \frac{2}{\varepsilon} \\
&\leq \frac{22}{\varepsilon} \sqrt{K \varepsilon T} \log(K) \log(KT) + 10\sqrt{KT} + \frac{2}{\varepsilon} \\
&\leq \frac{32\sqrt{KT} \log(K) \log(KT)}{\sqrt{\varepsilon}} + \frac{2}{\varepsilon},
\end{aligned}
$$

completing the proof. ∎

## F. Proofs for Bandits with Expert Advice

The following guarantee about Multiplicative Weights (MW) will be useful when proving utility guarantees.

**Lemma F.1** ((Cesa-Bianchi & Lugosi, 2006; Littlestone & Warmuth, 1994)). *For any sequence of loss functions $\ell_1, \ldots, \ell_T$, where $\ell_t : [N] \to \mathbb{R}$, if $\eta > 0$ is such that $\eta \max_{j \in [N]} -\ell_t(j) \leq 1$ for all $t \in [T]$, then MW when run on $\ell_1, \ldots, \ell_T$ outputs distributions $P_{1:T} \in \Pi([N])^T$ such that*

$$
\sum_{t=1}^{T} \sum_{j=1}^{N} P_t(j)\ell_t(j) \leq \inf_{j \in [N]} \sum_{t=1}^{T} \ell_t(j) + \frac{\log(N)}{\eta} + \eta \sum_{t=1}^{T} \sum_{j=1}^{N} P_t(j)\ell_t(j)^2.
$$

### F.1. Proof of Theorem 4.1

*Proof.* (of Theorem 4.1) Consider a loss sequence $\ell_1, \ldots, \ell_T$ and a sequence of expert predictions $\mu_{1:T}^{1:N}$. Let $j^\star \in \arg\min_{j \in [N]} \sum_{t=1}^{T} \sum_{i=1}^{K} \mu_t^j(i)\ell_t(i)$ denote an optimal expert in hindsight. By definition of the bandit algorithm $\mathcal{B}$, pointwise for every $I_{1:T}^{1:N}$, we have that

$$
\mathbb{E}\left[\sum_{t=1}^{T} \tilde{\ell}_t(J_t)\right] \leq \sum_{t=1}^{T} \tilde{\ell}_t(j^\star) + \mathrm{R}_{\mathcal{B}}(T, N).
$$

By definition of $\tilde{\ell}_t$, we then have that

$$
\mathbb{E}\left[\sum_{t=1}^{T} \ell_t(I_t^{J_t})\right] \leq \sum_{t=1}^{T} \ell_t(I_t^{j^\star}) + \mathrm{R}_{\mathcal{B}}(T, N).
$$

Taking an outer expectation with respect to the randomness of $I_{1:T}^{1:N}$, we have,

$$
\mathbb{E}\left[\sum_{t=1}^{T} \ell_t(I_t^{j^\star})\right] = \sum_{t=1}^{T} \sum_{i=1}^{K} \mu_t^{j^\star}(i) \cdot \ell_t(i)
$$

---

**Algorithm 6** Local-DP EXP4

1: **Input:** Action space $[K]$, Number of experts $N$, privacy parameters $\varepsilon > 0, \eta, \gamma > 0$
2: **Initialize:** $w_1(j) = 1$ for all $j \in [N]$
3: **for** $t = 1, \ldots, T$ **do**
4:     Receive expert advice $\mu_t^1, \ldots, \mu_t^N$
5:     Set $P_t(j) = \frac{w_t(j)}{\sum_{j \in [N]} w_t(j)}$
6:     Set $Q_t(i) = (1 - \gamma) \sum_{j=1}^{N} P_t(j) \mu_t^j(i) + \frac{\gamma}{K}$.
7:     Predict $I_t \sim Q_t$
8:     Observe loss $\ell_t(I_t)$ and define $\ell_t'(i) := \ell_t(i) + Z_t^i$, where $Z_t^i \sim \text{Lap}(0, \frac{1}{\varepsilon})$
9:     Construct unbiased estimator $\hat{\ell}_t'(i) = \frac{\ell_t'(i) \mathbb{I}\{I_t = i\}}{Q_t(i)}$
10:    Define $\tilde{\ell}_t'(j) := \mu_t^j \cdot \hat{\ell}_t'$ for all $j \in [N]$
11:    Update $w_{t+1}(j) \leftarrow w_t(j) \cdot \exp\{-\eta \tilde{\ell}_t'(j)\}$
12: **end for**

---

which completes the proof. ∎

## F.2. Proof of Theorem 4.2

Let $\mathcal{B}$ be any bandit algorithm. Then, for every $\tau \geq 1$. We need to show that there exists a $\varepsilon$-differentially private bandit with expert advice algorithm $\mathcal{A}_\tau$ such that

$$\text{R}_{\mathcal{A}_\tau}(T, K, N) \leq \tau \tilde{\text{R}}_{\mathcal{B}}(\frac{T}{\tau}, N, \frac{1}{\varepsilon \tau}) + \tau.$$

*Proof.* (of Utility in Theorem 4.2). Fix $\varepsilon \leq 1$ and $\tau \geq 1$. By Theorem 3.1, we can convert $\mathcal{B}$ into an $\varepsilon$-differentially private bandit algorithm $\mathcal{B}_\tau$ such that

$$\text{R}_{\mathcal{B}_\tau}(T, K) \leq \tau \tilde{\text{R}}_{\mathcal{B}}(\frac{T}{\tau}, K, \frac{1}{\varepsilon \tau}) + \tau.$$

Then, using Theorem 4.1, we can convert $\mathcal{B}_\tau$ into a bandit with expert advice algorithm $\mathcal{A}_\tau$ such that

$$\text{R}_{\mathcal{A}_\tau}(T, K, N) \leq \text{R}_{\mathcal{B}_\tau}(T, N) \leq \tau \tilde{\text{R}}_{\mathcal{B}}(\frac{T}{\tau}, N, \frac{1}{\varepsilon \tau}) + \tau,$$

completing the proof. ∎

*Proof.* (of Privacy in Theorem 4.2) Consider the same algorithm as in the proof of the utility guarantee. That is, let $\mathcal{A}_\tau$ be the result of using Theorem 1 to convert $\mathcal{B}$ to $\mathcal{B}_\tau$ and Theorem 4.1 to convert $\mathcal{B}_\tau$ to $\mathcal{A}_\tau$. By Theorem 3.1, we know that $\mathcal{B}_\tau$ is $\varepsilon$-differentially private. It suffices to show that Algorithm 5, when given $\mathcal{E}_\tau$ as input is also $\varepsilon$-differentially private. To that end, let $\ell_{1:T}$ and $\ell_{1:T}'$ be two sequences that differ at exactly one timepoint. Let $\mu_{1:T}^{1:N}$ be any sequence of expert advice and fix $I_t^i \sim \mu_t^i$ for all $t \in [T]$ and $i \in [N]$. Observe that Algorithm 5 instantiates $\mathcal{B}_\tau$ on the action space $[N]$ and simulates $\mathcal{B}_\tau$ on the sequence of losses $\tilde{\ell}_t(j) := \ell_t(I_t^j)$. Let $\tilde{\ell}_{1:T}$ and $\tilde{\ell}_{1:T}'$ denote the two sequences of losses that Algorithm 5 simulates $\mathcal{B}_\tau$ on when run on $\ell_{1:T}$ and $\ell_{1:T}'$ respectively. Note that $\tilde{\ell}_{1:T}$ and $\tilde{\ell}_{1:T}'$ differ at exactly one timepoint. Thus, $\mathcal{B}_\tau$ outputs actions $J_1, \ldots, J_T$ in an $\varepsilon$-differentially private manner. Finally, by post-processing it follows that the sequence of actions $I_t^{J_t}$ output by Algorithm 5 is also $\varepsilon$-differentially private. ∎

## F.3. Proof of Theorem 4.4

*Proof.* (of Utility in Theorem 4.4) Fix $\varepsilon \leq 1$. Let $\lambda = \frac{1}{\varepsilon}$. Let $\ell_1, \ldots, \ell_T$ be any sequence of loss functions and $\mu_{1:T}^{1:N}$ be any sequence of advice vectors. Let $E$ be the event that there exists a $t \in [T]$ such that $\max_{i \in [K]} |Z_t^i|^2 \geq 10\lambda^2 \log^2(KT)$. Then, Lemma B.3 shows that $\mathbb{P}[E] \leq \frac{1}{T}$. Moreover, note that $\mathbb{E}[Z_t^i | E^c] = 0$ for all $i \in [K]$ and $t \in [T]$. Let $\mathcal{A}$ be the random variable denoting the internal randomness of Algorithm 6 when sampling actions $I_t$. We need to bound

$$\mathrm{R}(T, K, N) := \mathop{\mathbb{E}}_{\mathcal{A}, Z_{1:T}} \left[ \sum_{t=1}^{T} \ell_t(I_t) - \inf_{j \in [N]} \sum_{t=1}^{T} \mu_t^j \cdot \ell_t \right].$$

We can write $\mathrm{R}(T, K, N)$ as

$$\mathop{\mathbb{E}}_{\mathcal{A}, Z_{1:T}} \left[ \sum_{t=1}^{T} \ell_t(I_t) - \inf_{j \in [N]} \sum_{t=1}^{T} \mu_t^j \cdot \ell_t \middle| E \right] \mathbb{P}(E) + \mathop{\mathbb{E}}_{\mathcal{A}, Z_{1:T}} \left[ \sum_{t=1}^{T} \ell_t(I_t) - \inf_{j \in [N]} \sum_{t=1}^{T} \mu_t^j \cdot \ell_t \middle| E^c \right] \mathbb{P}(E^c)$$

Since $\mathbb{E}_{\mathcal{A}, Z_{1:T}} \left[ \sum_{t=1}^{T} \ell_t(I_t) - \inf_{j \in [N]} \sum_{t=1}^{T} \mu_t^j \cdot \ell_t \middle| E \right] \leq T$, we have that

$$\mathrm{R}(T, K, N) \leq \mathop{\mathbb{E}}_{\mathcal{A}, Z_{1:T}} \left[ \sum_{t=1}^{T} \ell_t(I_t) - \inf_{j \in [N]} \sum_{t=1}^{T} \mu_t^j \cdot \ell_t \middle| E^c \right] + 1.$$

Accordingly, for the remainder of the proof, we will assume that event $E^c$ has occurred, which further implies that $\max_{t \in [T]} \max_{i \in [K]} |Z_t^i| \leq 4\lambda \log(KT)$.

Algorithm 6 runs Multiplicative Weights using the noisy losses $\tilde{\ell}_1', \ldots, \tilde{\ell}_T'$. For $\gamma = 4\eta K \lambda \log(KT)$, we have that

$$\eta \max_{t \in [T]} \max_{j \in [N]} -\tilde{\ell}_t'(j) = \eta \max_{t \in [T]} \max_{j \in [N]} -\mu_t^j \cdot \tilde{\ell}_t' = \eta \max_{t \in [T]} \max_{j \in [N]} -\mu_t^j(I_t) \frac{(\ell_t(I_t) + Z_t^{I_t})}{Q_t(I_t)} \leq \frac{\eta K}{\gamma} (4\lambda \log(KT)) \leq 1.$$

Accordingly, for this choice of $\gamma$, Lemma F.1 implies that

$$\sum_{t=1}^{T} \sum_{j=1}^{N} P_t(j) \tilde{\ell}_t'(j) \leq \inf_{j \in [N]} \sum_{t=1}^{T} \tilde{\ell}_t'(j) + \frac{\log(N)}{\eta} + \eta \sum_{t=1}^{T} \sum_{j=1}^{N} P_t(j) \tilde{\ell}_t'(j)^2.$$

Taking expectation of both sides, we have that

$$\mathop{\mathbb{E}}_{\mathcal{A}, Z_{1:T}} \left[ \sum_{t=1}^{T} \sum_{j=1}^{N} P_t(j) \tilde{\ell}_t'(j) \middle| E^c \right] \leq \inf_{j \in [N]} \mathop{\mathbb{E}}_{Z_{1:T}} \left[ \sum_{t=1}^{T} \tilde{\ell}_t'(j) \middle| E^c \right] + \frac{\log(N)}{\eta} + \eta \mathop{\mathbb{E}}_{\mathcal{A}, Z_{1:T}} \left[ \sum_{t=1}^{T} \sum_{j=1}^{N} P_t(j) \tilde{\ell}_t'(j)^2 \middle| E^c \right].$$

We now analyze each of the three terms with expectations separately. First,

$$\mathop{\mathbb{E}}_{\mathcal{A}, Z_{1:T}} \left[ \sum_{t=1}^{T} \sum_{j=1}^{N} P_r(j) \tilde{\ell}_t'(j) \middle| E^c \right] = \mathop{\mathbb{E}}_{\mathcal{A}, Z_{1:T}} \left[ \sum_{t=1}^{T} \sum_{j=1}^{N} P_t(j) \sum_{i=1}^{K} \hat{\ell}_t'(i) \mu_t^j(i) \middle| E^c \right]$$

$$= \mathop{\mathbb{E}}_{\mathcal{A}, Z_{1:T}} \left[ \sum_{t=1}^{T} \sum_{i=1}^{K} \left( \sum_{j=1}^{N} P_t(j) \mu_t^j(i) \right) \hat{\ell}_t'(i) \middle| E^c \right]$$

$$= \mathop{\mathbb{E}}_{\mathcal{A}, Z_{1:T}} \left[ \sum_{t=1}^{T} \sum_{i=1}^{K} \left( \frac{Q_t(i) - \frac{\gamma}{K}}{1 - \gamma} \right) \hat{\ell}_t'(i) \middle| E^c \right]$$

$$= \frac{1}{(1 - \gamma)} \mathop{\mathbb{E}}_{\mathcal{A}, Z_{1:T}} \left[ \sum_{t=1}^{T} \sum_{i=1}^{K} Q_t(i) \hat{\ell}_t'(i) \middle| E^c \right] - \frac{\gamma}{K(1 - \gamma)} \mathop{\mathbb{E}}_{\mathcal{A}, Z_{1:T}} \left[ \sum_{t=1}^{T} \sum_{i=1}^{K} \hat{\ell}_t'(i) \middle| E^c \right]$$

Next,

$$
\mathop{\mathbb{E}}_{\mathcal{A},Z_{1:T}}\left[\sum_{t=1}^{T}\sum_{j=1}^{N}P_t(j)\tilde{\ell}'_t(j)^2\bigg|E^c\right] = \mathop{\mathbb{E}}_{\mathcal{A},Z_{1:T}}\left[\sum_{t=1}^{T}\sum_{j=1}^{N}P_t(j)(\mu_t^j\cdot\hat{\ell}'_t)^2\bigg|E^c\right]
$$

$$
\leq \mathop{\mathbb{E}}_{\mathcal{A},Z_{1:T}}\left[\sum_{t=1}^{T}\sum_{j=1}^{N}P_t(j)\sum_{i=1}^{K}\hat{\ell}'_t(i)^2\mu_t^j(i)\bigg|E^c\right]
$$

$$
= \mathop{\mathbb{E}}_{\mathcal{A},Z_{1:T}}\left[\sum_{t=1}^{T}\sum_{i=1}^{K}\left(\sum_{j=1}^{N}P_t(j)\mu_t^j(i)\right)\hat{\ell}'_t(i)^2\bigg|E^c\right]
$$

$$
= \mathop{\mathbb{E}}_{\mathcal{A},Z_{1:T}}\left[\sum_{t=1}^{T}\sum_{i=1}^{K}\left(\frac{Q_t(i)-\frac{\gamma}{K}}{1-\gamma}\right)\hat{\ell}'_t(i)^2\bigg|E^c\right]
$$

$$
\leq \frac{1}{(1-\gamma)}\mathop{\mathbb{E}}_{\mathcal{A},Z_{1:T}}\left[\sum_{t=1}^{T}\sum_{i=1}^{K}Q_t(i)\hat{\ell}'_t(i)^2\bigg|E^c\right].
$$

Finally,

$$
\inf_{j\in[N]}\mathop{\mathbb{E}}_{Z_{1:T}}\left[\sum_{t=1}^{T}\tilde{\ell}'_t(j)\bigg|E^c\right] = \inf_{j\in[N]}\mathop{\mathbb{E}}_{Z_{1:T}}\left[\sum_{t=1}^{T}\hat{\ell}'_t\cdot\mu_t^j\bigg|E^c\right]
$$

$$
= \inf_{j\in[N]}\mathop{\mathbb{E}}_{Z_{1:T}}\left[\sum_{t=1}^{T}\ell'_t\cdot\mu_t^j\bigg|E^c\right]
$$

$$
= \inf_{j\in[N]}\sum_{t=1}^{T}\ell_t\cdot\mu_t^j,
$$

where the second equality follows by the unbiasedness of $\hat{\ell}'_t$ and the last by the fact that $Z_t^i$ is zero-mean (conditioned on $E^c$). Putting all the bounds together, we get that

$$
\frac{1}{(1-\gamma)}\mathop{\mathbb{E}}_{\mathcal{A},Z_{1:T}}\left[\sum_{t=1}^{T}\sum_{i=1}^{K}Q_t(i)\hat{\ell}'_t(i)\bigg|E^c\right]
$$

is at most

$$
\inf_{j\in[N]}\sum_{t=1}^{T}\ell_t\cdot\mu_t^j + \frac{\log(N)}{\eta} + \frac{\gamma}{K(1-\gamma)}\mathop{\mathbb{E}}_{Z_{1:T}}\left[\sum_{t=1}^{T}\sum_{i=1}^{K}\hat{\ell}'_t(i)\bigg|E^c\right] + \frac{\eta}{(1-\gamma)}\mathop{\mathbb{E}}_{\mathcal{A},Z_{1:T}}\left[\sum_{t=1}^{T}\sum_{i=1}^{K}Q_t(i)\hat{\ell}'_t(i)^2\bigg|E^c\right].
$$

Multiplying both sides by $(1-\gamma)$, we have that $\mathbb{E}_{\mathcal{A},Z_{1:T}}\left[\sum_{t=1}^{T}\sum_{i=1}^{K}Q_t(i)\hat{\ell}'_t(i)\big|E^c\right]$ is at most

$$
(1-\gamma)\inf_{j\in[N]}\sum_{t=1}^{T}\ell_t\cdot\mu_t^j + \frac{(1-\gamma)\log(N)}{\eta} + \frac{\gamma}{K}\mathop{\mathbb{E}}_{Z_{1:T}}\left[\sum_{t=1}^{T}\sum_{i=1}^{K}\hat{\ell}'_t(i)\bigg|E^c\right] + \eta\mathop{\mathbb{E}}_{\mathcal{A},Z_{1:T}}\left[\sum_{t=1}^{T}\sum_{i=1}^{K}Q_t(i)\hat{\ell}'_t(i)^2\bigg|E^c\right]
$$

which implies that

$$
\mathop{\mathbb{E}}_{\mathcal{A},Z_{1:T}}\left[\sum_{t=1}^{T}\sum_{i=1}^{K}Q_t(i)\hat{\ell}'_t(i)\bigg|E^c\right] \leq \inf_{j\in[N]}\sum_{t=1}^{T}\ell_t\cdot\mu_t^j + \frac{\log(N)}{\eta} + \gamma T + \eta\mathop{\mathbb{E}}_{\mathcal{A},Z_{1:T}}\left[\sum_{t=1}^{T}\sum_{i=1}^{K}Q_t(i)\hat{\ell}'_t(i)^2\bigg|E^c\right].
$$

Using the fact that $\hat{\ell}'_t$ is an unbiased estimator of $\ell'_t$ gives that

$$\underset{\mathcal{A}, Z_{1:T}}{\mathbb{E}}\left[\sum_{t=1}^{T}\sum_{i=1}^{K} Q_t(i)\ell_t'(i)\Bigg| E^c\right] \le \inf_{j\in[N]}\sum_{t=1}^{T}\ell_t \cdot \mu_t^j + \frac{\log(N)}{\eta} + \gamma T + \eta\underset{Z_{1:T}}{\mathbb{E}}\left[\sum_{t=1}^{T}\sum_{i=1}^{K}\ell_t'(i)^2\Bigg| E^c\right]$$

Since $Z_t^i$ is zero-mean (conditioned on $E^c$) and independent of $Q_t(i)$, we get that,

$$\underset{\mathcal{A}, Z_{1:T}}{\mathbb{E}}\left[\sum_{t=1}^{T}\sum_{i=1}^{K} Q_t(i)\ell_t(i)\Bigg| E^c\right] \le \inf_{j\in[N]}\sum_{t=1}^{T}\ell_t \cdot \mu_t^j + \frac{\log(N)}{\eta} + \gamma T + \eta\underset{Z_{1:T}}{\mathbb{E}}\left[\sum_{t=1}^{T}\sum_{i=1}^{K}\ell_t'(i)^2\Bigg| E^c\right].$$

It suffices to bound the expectation on the right-hand side. To that end, observe that

$$\begin{aligned}
\underset{Z_{1:T}}{\mathbb{E}}\left[\sum_{t=1}^{T}\sum_{i=1}^{K}\ell_t'(i)^2\Bigg| E^c\right] &= \underset{Z_{1:T}}{\mathbb{E}}\left[\sum_{t=1}^{T}\sum_{i=1}^{K}(\ell_t(i) + Z_t^i)^2\Bigg| E^c\right] \\
&\le 2\underset{Z_{1:T}}{\mathbb{E}}\left[\sum_{t=1}^{T}\sum_{i=1}^{K}(\ell_t(i)^2 + (Z_t^i)^2)\Bigg| E^c\right] \\
&\le 2\underset{Z_{1:T}}{\mathbb{E}}\left[\sum_{t=1}^{T}\sum_{i=1}^{K}(1 + (Z_t^i)^2)\Bigg| E^c\right] \\
&\le 2KT(1 + 10\lambda^2\log^2 KT)
\end{aligned}$$

Thus, overall we have that

$$\underset{\mathcal{A}, Z_{1:T}}{\mathbb{E}}\left[\sum_{t=1}^{T}\sum_{i=1}^{K} Q_t(i)\ell_t(i)\Bigg| E^c\right] \le \inf_{j\in[N]}\sum_{t=1}^{T}\ell_t \cdot \mu_t^j + \frac{\log(N)}{\eta} + \gamma T + 2\eta KT(1 + 10\lambda^2\log^2 KT).$$

Plugging in our choice of $\gamma = 4\eta K\lambda\log(KT)$,

$$\underset{\mathcal{A}, Z_{1:T}}{\mathbb{E}}\left[\sum_{t=1}^{T}\sum_{i=1}^{K} Q_t(i)\ell_t(i)\Bigg| E^c\right] \le \inf_{j\in[N]}\sum_{t=1}^{T}\ell_t \cdot \mu_t^j + \frac{\log(N)}{\eta} + 4\eta KT\lambda\log(KT) + 2\eta KT(1 + 10\lambda^2\log^2 KT).$$

which for $\lambda \ge 1$ gives

$$\underset{\mathcal{A}, Z_{1:T}}{\mathbb{E}}\left[\sum_{t=1}^{T}\sum_{i=1}^{K} Q_t(i)\ell_t(i)\Bigg| E^c\right] \le \inf_{j\in[N]}\sum_{t=1}^{T}\ell_t \cdot \mu_t^j + \frac{\log(N)}{\eta} + 3\eta KT(1 + 10\lambda^2\log^2 KT).$$

Picking $\eta = \sqrt{\frac{\log(N)}{3TK(1+10\lambda^2\log^2 KT)}}$, we have

$$\underset{\mathcal{A}, Z_{1:T}}{\mathbb{E}}\left[\sum_{t=1}^{T}\sum_{i=1}^{K} Q_t(i)\ell_t(i)\Bigg| E^c\right] \le \inf_{j\in[N]}\sum_{t=1}^{T}\ell_t \cdot \mu_t^j + 16\sqrt{TK\log(N)}\lambda\log(KT).$$

For our choice $\lambda = \frac{1}{\varepsilon}$, we get

$$\underset{\mathcal{A}, Z_{1:T}}{\mathbb{E}}\left[\sum_{t=1}^{T}\sum_{i=1}^{K} Q_t(i)\ell_t(i)\Bigg| E^c\right] \le \inf_{j\in[N]}\sum_{t=1}^{T}\ell_t \cdot \mu_t^j + \frac{16\sqrt{TK\log(N)}\log(KT)}{\varepsilon}.$$

Finally, noting that

$$\mathrm{R}(T, K, N) \leq \underset{\mathcal{A}, Z_{1:T}}{\mathbb{E}} \left[ \sum_{t=1}^{T} \sum_{i=1}^{K} Q_t(i) \ell_t(i) \middle| E^c \right] - \inf_{j \in [N]} \sum_{t=1}^{T} \mu_t^j \cdot \ell_t + 1$$

completes the proof. ∎

The proof of privacy in Theorem 4.4 is identical to the proof of Lemma 3.3 after taking batch size $\tau = 1$, so we omit the details here.

### F.4. Proof of Theorem 4.5

*Proof.* (of Utility in Theorem 4.5) Fix $\varepsilon, \delta \in (0, 1]$ and batch size $\tau$. Let $\lambda = \frac{3K\sqrt{N \log(\frac{1}{\delta})}}{\gamma \tau \varepsilon}$. Let $\ell_1, \dots, \ell_T$ be any sequence of loss functions and $\mu_{1:T}^{1:N}$ be any sequence of advice vectors. Let $E$ be the event that there exists a $r \in \{1, \dots, \lfloor \frac{T}{\tau} \rfloor\}$ such that $\max_{j \in [N]} |Z_r^j|^2 \geq 10\lambda^2 \log^2(N \lfloor \frac{T}{\tau} \rfloor)$. Then, Lemma B.3 shows that $\mathbb{P}[E] \leq \frac{\tau}{T}$. Moreover, note that $\mathbb{E}[Z_r^j | E^c] = 0$ for all $j \in [N]$ and $r \in [\lfloor \frac{T}{\tau} \rfloor]$. Let $\mathcal{A}$ be the random variable denoting the internal randomness of Algorithm 6 when sampling actions $I_t$. We need to bound

$$\mathrm{R}(T, K, N) := \underset{\mathcal{A}, Z_{1:T}}{\mathbb{E}} \left[ \sum_{t=1}^{T} \ell_t(I_t) - \inf_{j \in [N]} \sum_{t=1}^{T} \mu_t^j \cdot \ell_t \right].$$

We can write $\mathrm{R}(T, K, N)$ as

$$\underset{\mathcal{A}, Z_{1:T}}{\mathbb{E}} \left[ \sum_{t=1}^{T} \ell_t(I_t) - \inf_{j \in [N]} \sum_{t=1}^{T} \mu_t^j \cdot \ell_t \middle| E \right] \mathbb{P}(E) + \underset{\mathcal{A}, Z_{1:T}}{\mathbb{E}} \left[ \sum_{t=1}^{T} \ell_t(I_t) - \inf_{j \in [N]} \sum_{t=1}^{T} \mu_t^j \cdot \ell_t \middle| E^c \right] \mathbb{P}(E^c)$$

Since $\mathbb{E}_{\mathcal{A}, Z_{1:T}} \left[ \sum_{t=1}^{T} \ell_t(I_t) - \inf_{j \in [N]} \sum_{t=1}^{T} \mu_t^j \cdot \ell_t \middle| E \right] \leq T$, we have that

$$\mathrm{R}(T, K, N) \leq \underset{\mathcal{A}, Z_{1:T}}{\mathbb{E}} \left[ \sum_{t=1}^{T} \ell_t(I_t) - \inf_{j \in [N]} \sum_{t=1}^{T} \mu_t^j \cdot \ell_t \middle| E^c \right] + \tau \tag{1}$$

Accordingly, for the remainder of the proof, we will assume that event $E^c$ has occurred, which further implies that $\max_{r \in [\lfloor \frac{T}{\tau} \rfloor]} \max_{j \in [N]} |Z_r^j| \leq 4\lambda \log(N \lfloor \frac{T}{\tau} \rfloor)$.

Algorithm 2 runs Multiplicative Weights using the noisy, batched losses $\tilde{\ell}_1', \dots, \tilde{\ell}_{\lfloor \frac{T}{\tau} \rfloor}'$. For $\gamma \geq \frac{12\eta K \sqrt{N \log(\frac{1}{\delta})} \log(NT)}{\varepsilon \tau}$, we have that

$$\max_{r \in [\lfloor \frac{T}{\tau} \rfloor]} \max_{j \in [N]} -\eta(\tilde{\ell}_r'(j)) \leq \max_{r \in [\lfloor \frac{T}{\tau} \rfloor]} \max_{j \in [N]} -\eta(\tilde{\ell}_r(j) + Z_r^j) \leq \max_{r \in [\lfloor \frac{T}{\tau} \rfloor]} \max_{j \in [N]} -\eta Z_r^j \leq \eta \frac{12K \sqrt{N \log\left(\frac{1}{\delta}\right)} \log(NT)}{\varepsilon \tau \gamma} \leq 1.$$

Accordingly, for any choice $\gamma \geq \frac{12\eta K \sqrt{N \log(\frac{1}{\delta})} \log(NT)}{\varepsilon \tau}$, Lemma F.1 implies that

$$\sum_{r=1}^{\lfloor \frac{T}{\tau} \rfloor} \sum_{j=1}^{N} P_r(j) \tilde{\ell}_r'(j) \leq \inf_{j \in [N]} \sum_{r=1}^{\lfloor \frac{T}{\tau} \rfloor} \tilde{\ell}_r'(j) + \frac{\log(N)}{\eta} + \eta \sum_{r=1}^{\lfloor \frac{T}{\tau} \rfloor} \sum_{j=1}^{N} P_r(j) \tilde{\ell}_r'(j)^2.$$

Taking expectation of both sides, we have that

$$\mathop{\mathbb{E}}_{\mathcal{A},Z_{1:T}}\left[\left|\sum_{r=1}^{\lfloor\frac{T}{\tau}\rfloor}\sum_{j=1}^{N}P_r(j)\tilde{\ell}_r'(j)\right|\,\middle|\,E^c\right] \le \inf_{j\in[N]}\mathop{\mathbb{E}}_{Z_{1:T}}\left[\left|\sum_{r=1}^{\lfloor\frac{T}{\tau}\rfloor}\tilde{\ell}_r'(j)\right|\,\middle|\,E^c\right] + \frac{\log(N)}{\eta} + \eta\mathop{\mathbb{E}}_{\mathcal{A},Z_{1:T}}\left[\left|\sum_{r=1}^{\lfloor\frac{T}{\tau}\rfloor}\sum_{j=1}^{N}P_r(j)\tilde{\ell}_r'(j)^2\right|\,\middle|\,E^c\right].$$

Using the fact that $Z_r^j$ is zero-mean and conditionally independent of $P_r$ given the history of the game up to and including time point $(r-1)\tau$, we have that

$$\mathop{\mathbb{E}}_{\mathcal{A},Z_{1:T}}\left[\left|\sum_{r=1}^{\lfloor\frac{T}{\tau}\rfloor}\sum_{j=1}^{N}P_r(j)\tilde{\ell}_r(j)\right|\,\middle|\,E^c\right] \le \inf_{j\in[N]}\mathop{\mathbb{E}}_{Z_{1:T}}\left[\left|\sum_{r=1}^{\lfloor\frac{T}{\tau}\rfloor}\tilde{\ell}_r(j)\right|\,\middle|\,E^c\right] + \frac{\log(N)}{\eta} + \eta\mathop{\mathbb{E}}_{\mathcal{A},Z_{1:T}}\left[\left|\sum_{r=1}^{\lfloor\frac{T}{\tau}\rfloor}\sum_{j=1}^{N}P_r(j)\tilde{\ell}_r'(j)^2\right|\,\middle|\,E^c\right].$$

We now analyze each of the three terms with expectations separately. First,

$$\begin{aligned}\mathop{\mathbb{E}}_{\mathcal{A},Z_{1:T}}\left[\left|\sum_{r=1}^{\lfloor\frac{T}{\tau}\rfloor}\sum_{j=1}^{N}P_r(j)\tilde{\ell}_r(j)\right|\,\middle|\,E^c\right] &= \frac{1}{\tau}\mathop{\mathbb{E}}_{\mathcal{A},Z_{1:T}}\left[\left|\sum_{r=1}^{\lfloor\frac{T}{\tau}\rfloor}\sum_{s=(r-1)\tau+1}^{r\tau}\sum_{j=1}^{N}P_r(j)\sum_{i=1}^{K}\hat{\ell}_s(i)\mu_s^j(i)\right|\,\middle|\,E^c\right]\\[2mm]
&= \frac{1}{\tau}\mathop{\mathbb{E}}_{\mathcal{A},Z_{1:T}}\left[\left|\sum_{r=1}^{\lfloor\frac{T}{\tau}\rfloor}\sum_{s=(r-1)\tau+1}^{r\tau}\sum_{i=1}^{K}\left(\sum_{j=1}^{N}P_r(j)\mu_s^j(i)\right)\hat{\ell}_s(i)\right|\,\middle|\,E^c\right]\\[2mm]
&= \frac{1}{\tau}\mathop{\mathbb{E}}_{\mathcal{A},Z_{1:T}}\left[\left|\sum_{t=1}^{\tau\lfloor\frac{T}{\tau}\rfloor}\sum_{i=1}^{K}\left(\frac{Q_t(i)-\frac{\gamma}{K}}{1-\gamma}\right)\hat{\ell}_t(i)\right|\,\middle|\,E^c\right]\\[2mm]
&\ge \frac{1}{\tau(1-\gamma)}\mathop{\mathbb{E}}_{\mathcal{A},Z_{1:T}}\left[\left|\sum_{t=1}^{\tau\lfloor\frac{T}{\tau}\rfloor}\sum_{i=1}^{K}Q_t(i)\hat{\ell}_t(i)\right|\,\middle|\,E^c\right] - \frac{\gamma}{(1-\gamma)}\left\lfloor\frac{T}{\tau}\right\rfloor.\end{aligned}$$

Next,

$$\begin{aligned}\mathop{\mathbb{E}}_{\mathcal{A},Z_{1:T}}\left[\left|\sum_{r=1}^{\lfloor\frac{T}{\tau}\rfloor}\sum_{j=1}^{N}P_r(j)\tilde{\ell}_r'(j)^2\right|\,\middle|\,E^c\right] &= \mathop{\mathbb{E}}_{\mathcal{A},Z_{1:T}}\left[\left|\sum_{r=1}^{\lfloor\frac{T}{\tau}\rfloor}\sum_{j=1}^{N}P_r(j)(\tilde{\ell}_r(j)+Z_r^j)^2\right|\,\middle|\,E^c\right]\\[2mm]
&= \mathop{\mathbb{E}}_{\mathcal{A},Z_{1:T}}\left[\left|\sum_{r=1}^{\lfloor\frac{T}{\tau}\rfloor}\sum_{j=1}^{N}P_r(j)(\tilde{\ell}_r(j)^2+(Z_r^j)^2)\right|\,\middle|\,E^c\right]\\[2mm]
&\le \mathop{\mathbb{E}}_{\mathcal{A},Z_{1:T}}\left[\left|\sum_{r=1}^{\lfloor\frac{T}{\tau}\rfloor}\sum_{j=1}^{N}P_r(j)(\tilde{\ell}_r(j)^2+10\lambda^2\log^2(N\lfloor\tfrac{T}{\tau}\rfloor))\right|\,\middle|\,E^c\right]\\[2mm]
&= \mathop{\mathbb{E}}_{\mathcal{A},Z_{1:T}}\left[\left|\sum_{r=1}^{\lfloor\frac{T}{\tau}\rfloor}\sum_{j=1}^{N}P_r(j)\tilde{\ell}_r(j)^2\right|\,\middle|\,E^c\right] + 10\left\lfloor\frac{T}{\tau}\right\rfloor\lambda^2\log^2(N\lfloor\tfrac{T}{\tau}\rfloor)\end{aligned}$$

To bound the first of the two terms above, note that:

$$
\begin{aligned}
\underset{\mathcal{A}, Z_{1:T}}{\mathbb{E}} \left[ \sum_{r=1}^{\lfloor \frac{T}{\tau} \rfloor} \sum_{j=1}^{N} P_r(j) \tilde{\ell}_r(j)^2 \middle| E^c \right] &\leq \underset{\mathcal{A}, Z_{1:T}}{\mathbb{E}} \left[ \sum_{r=1}^{\lfloor \frac{T}{\tau} \rfloor} \sum_{j=1}^{N} P_r(j) \left( \frac{1}{\tau} \sum_{s=(r-1)\tau+1}^{r\tau} \hat{\ell}_s \cdot \mu_s^j \right)^2 \middle| E^c \right] \\
&\leq \underset{\mathcal{A}, Z_{1:T}}{\mathbb{E}} \left[ \sum_{r=1}^{\lfloor \frac{T}{\tau} \rfloor} \sum_{j=1}^{N} P_r(j) \frac{1}{\tau^2} \left( \sum_{s=(r-1)\tau+1}^{r\tau} \hat{\ell}_s \cdot \mu_s^j \right)^2 \middle| E^c \right] \\
&\leq \underset{\mathcal{A}, Z_{1:T}}{\mathbb{E}} \left[ \sum_{r=1}^{\lfloor \frac{T}{\tau} \rfloor} \sum_{j=1}^{N} P_r(j) \frac{1}{\tau} \sum_{s=(r-1)\tau+1}^{r\tau} \left( \hat{\ell}_s \cdot \mu_s^j \right)^2 \middle| E^c \right] \\
&\leq \frac{1}{\tau} \underset{\mathcal{A}, Z_{1:T}}{\mathbb{E}} \left[ \sum_{r=1}^{\lfloor \frac{T}{\tau} \rfloor} \sum_{j=1}^{N} P_r(j) \sum_{s=(r-1)\tau+1}^{r\tau} \hat{\ell}_s^2 \cdot \mu_s^j \middle| E^c \right] \\
&= \frac{1}{\tau} \underset{\mathcal{A}, Z_{1:T}}{\mathbb{E}} \left[ \sum_{r=1}^{\lfloor \frac{T}{\tau} \rfloor} \sum_{s=(r-1)\tau+1}^{r\tau} \sum_{j=1}^{N} P_r(j) \sum_{i=1}^{K} \hat{\ell}_s^2(i) \mu_s^j(i) \middle| E^c \right] \\
&= \frac{1}{\tau} \underset{\mathcal{A}, Z_{1:T}}{\mathbb{E}} \left[ \sum_{r=1}^{\lfloor \frac{T}{\tau} \rfloor} \sum_{s=(r-1)\tau+1}^{r\tau} \sum_{i=1}^{K} \left( \sum_{j=1}^{N} P_r(j) \mu_s^j(i) \right) \hat{\ell}_s^2(i) \middle| E^c \right] \\
&= \frac{1}{\tau} \underset{\mathcal{A}, Z_{1:T}}{\mathbb{E}} \left[ \sum_{t=1}^{\tau \lfloor \frac{T}{\tau} \rfloor} \sum_{i=1}^{K} \left( \frac{Q_t(i) - \frac{\gamma}{K}}{1 - \gamma} \right) \hat{\ell}_t^2(i) \middle| E^c \right] \\
&\leq \frac{1}{\tau(1 - \gamma)} \underset{\mathcal{A}, Z_{1:T}}{\mathbb{E}} \left[ \sum_{t=1}^{\tau \lfloor \frac{T}{\tau} \rfloor} \sum_{i=1}^{K} Q_t(i) \hat{\ell}_t^2(i) \middle| E^c \right].
\end{aligned}
$$

Finally,

$$
\begin{aligned}
\inf_{j \in [N]} \underset{Z_{1:T}}{\mathbb{E}} \left[ \sum_{r=1}^{\lfloor \frac{T}{\tau} \rfloor} \tilde{\ell}_r(j) \middle| E^c \right] &= \frac{1}{\tau} \inf_{j \in [N]} \underset{Z_{1:T}}{\mathbb{E}} \left[ \sum_{r=1}^{\lfloor \frac{T}{\tau} \rfloor} \sum_{s=(r-1)\tau+1}^{r\tau} \hat{\ell}_s \cdot \mu_s^j \middle| E^c \right] \\
&= \frac{1}{\tau} \inf_{j \in [N]} \underset{Z_{1:T}}{\mathbb{E}} \left[ \sum_{t=1}^{\tau \lfloor \frac{T}{\tau} \rfloor} \hat{\ell}_t \cdot \mu_t^j \middle| E^c \right] \\
&= \frac{1}{\tau} \inf_{j \in [N]} \sum_{t=1}^{\tau \lfloor \frac{T}{\tau} \rfloor} \ell_t \cdot \mu_t^j,
\end{aligned}
$$

where the last equality follows by the unbiasedness of $\hat{\ell}_t$. Putting all the bounds together, we get that

$$\frac{1}{\tau(1-\gamma)} \underset{\mathcal{A},Z_{1:T}}{\mathbb{E}} \left[ \sum_{t=1}^{\tau\lfloor\frac{T}{\tau}\rfloor} \sum_{i=1}^{K} Q_t(i)\hat{\ell}_t(i) \middle| E^c \right] \leq \frac{1}{\tau} \inf_{j\in[N]} \sum_{t=1}^{\tau\lfloor\frac{T}{\tau}\rfloor} \ell_t \cdot \mu_t^j + \frac{\log(N)}{\eta} + \frac{\gamma}{(1-\gamma)} \left\lfloor \frac{T}{\tau} \right\rfloor$$

$$+ \frac{\eta}{\tau(1-\gamma)} \underset{\mathcal{A},Z_{1:T}}{\mathbb{E}} \left[ \sum_{t=1}^{\tau\lfloor\frac{T}{\tau}\rfloor} \sum_{i=1}^{K} Q_t(i)\hat{\ell}_t^2(i) \middle| E^c \right]$$

$$+ 10\eta \left\lfloor \frac{T}{\tau} \right\rfloor \lambda^2 \log^2(N \left\lfloor \frac{T}{\tau} \right\rfloor).$$

Multiplying both sides by $\tau(1-\gamma)$, gives

$$\underset{\mathcal{A},Z_{1:T}}{\mathbb{E}} \left[ \sum_{t=1}^{\tau\lfloor\frac{T}{\tau}\rfloor} \sum_{i=1}^{K} Q_t(i)\hat{\ell}_t(i) \middle| E^c \right] \leq (1-\gamma) \inf_{j\in[N]} \sum_{t=1}^{\tau\lfloor\frac{T}{\tau}\rfloor} \ell_t \cdot \mu_t^j + \frac{\tau(1-\gamma)\log(N)}{\eta} + \tau\gamma \left\lfloor \frac{T}{\tau} \right\rfloor$$

$$+ \eta \underset{\mathcal{A},Z_{1:T}}{\mathbb{E}} \left[ \sum_{t=1}^{\tau\lfloor\frac{T}{\tau}\rfloor} \sum_{i=1}^{K} Q_t(i)\hat{\ell}_t^2(i) \middle| E^c \right]$$

$$+ 10\eta(1-\gamma)\tau \left\lfloor \frac{T}{\tau} \right\rfloor \lambda^2 \log^2(N \left\lfloor \frac{T}{\tau} \right\rfloor),$$

which implies that

$$\underset{\mathcal{A},Z_{1:T}}{\mathbb{E}} \left[ \sum_{t=1}^{\tau\lfloor\frac{T}{\tau}\rfloor} \sum_{i=1}^{K} Q_t(i)\hat{\ell}_t(i) \middle| E^c \right] \leq \inf_{j\in[N]} \sum_{t=1}^{\tau\lfloor\frac{T}{\tau}\rfloor} \ell_t \cdot \mu_t^j + \frac{\tau\log(N)}{\eta} + \gamma T$$

$$+ \eta \underset{\mathcal{A},Z_{1:T}}{\mathbb{E}} \left[ \sum_{t=1}^{\tau\lfloor\frac{T}{\tau}\rfloor} \sum_{i=1}^{K} Q_t(i)\hat{\ell}_t^2(i) \middle| E^c \right]$$

$$+ 10\eta T\lambda^2 \log^2(N \left\lfloor \frac{T}{\tau} \right\rfloor).$$

Using the fact that $\hat{\ell}_t$ is an unbiased estimator of $\ell_t$ gives that

$$\underset{\mathcal{A},Z_{1:T}}{\mathbb{E}} \left[ \sum_{t=1}^{\tau\lfloor\frac{T}{\tau}\rfloor} \sum_{i=1}^{K} Q_t(i)\ell_t(i) \middle| E^c \right] \leq \inf_{j\in[N]} \sum_{t=1}^{\tau\lfloor\frac{T}{\tau}\rfloor} \ell_t \cdot \mu_t^j + \frac{\tau\log(N)}{\eta} + \gamma T + \eta \underset{\mathcal{A},Z_{1:T}}{\mathbb{E}} \left[ \sum_{t=1}^{\tau\lfloor\frac{T}{\tau}\rfloor} \sum_{i=1}^{K} \ell_t^2(i) \middle| E^c \right] + 10\eta T\lambda^2 \log^2(N \left\lfloor \frac{T}{\tau} \right\rfloor).$$

By the boundedness of the loss, we have

$$\underset{\mathcal{A},Z_{1:T}}{\mathbb{E}} \left[ \sum_{t=1}^{\tau\lfloor\frac{T}{\tau}\rfloor} \sum_{i=1}^{K} Q_t(i)\ell_t(i) \middle| E^c \right] \leq \inf_{j\in[N]} \sum_{t=1}^{\tau\lfloor\frac{T}{\tau}\rfloor} \ell_t \cdot \mu_t^j + \frac{\tau\log(N)}{\eta} + \gamma T + \eta K\tau \left\lfloor \frac{T}{\tau} \right\rfloor + 10\eta T\lambda^2 \log^2(N \left\lfloor \frac{T}{\tau} \right\rfloor).$$

Bounding the regret in the last $\tau$ rounds by $\tau$, gives

$$\mathop{\mathbb{E}}_{\mathcal{A}, Z_{1:T}} \left[ \sum_{t=1}^{T} \sum_{i=1}^{K} Q_t(i)\ell_t(i) \middle| E^c \right] \leq \inf_{j \in [N]} \sum_{t=1}^{T} \ell_t \cdot \mu_t^j + \frac{\tau \log(N)}{\eta} + \gamma T + \eta TK + 10\eta T\lambda^2 \log^2(NT) + \tau$$

$$\leq \inf_{j \in [N]} \sum_{t=1}^{T} \ell_t \cdot \mu_t^j + \frac{\tau \log(N)}{\eta} + \gamma T + \eta TK + \frac{90\eta TNK^2 \log(\frac{1}{\delta}) \log^2(NT)}{\varepsilon^2 \gamma^2 \tau^2} + \tau$$

Using Equation 1, then gives that

$$\mathrm{R}(T, K, N) \leq \frac{\tau \log(N)}{\eta} + \gamma T + \eta TK + \frac{90\eta TNK^2 \log(\frac{1}{\delta}) \log^2(NT)}{\varepsilon^2 \gamma^2 \tau^2} + 2\tau$$

Since $\eta < 1$, we trivially have that

$$\mathrm{R}(T, K, N) \leq \frac{3\tau \log(N)}{\eta} + \gamma T + \eta TK + \frac{90\eta TNK^2 \log(\frac{1}{\delta}) \log^2(NT)}{\varepsilon^2 \gamma^2 \tau^2}.$$

Now, choosing $\gamma = \max \left\{ \frac{\eta^{1/3} N^{1/3} K^{2/3} \log^{2/3}(NT)}{\varepsilon^{2/3} \tau^{2/3}}, \frac{12\eta K \sqrt{N \log(\frac{1}{\delta})} \log(NT)}{\varepsilon \tau} \right\}$, gives

$$\mathrm{R}(T, K, N) \leq \frac{3\tau \log(N)}{\eta}$$
$$+ 90 \max \left\{ \frac{\eta^{1/3} (N \log(\frac{1}{\delta}))^{1/3} K^{2/3} \log^{2/3}(NT)}{\varepsilon^{2/3} \tau^{2/3}}, \frac{\eta K \sqrt{N \log(\frac{1}{\delta})} \log(NT)}{\varepsilon \tau} \right\} T + \eta TK.$$

Choosing $\eta = \frac{(N \log(\frac{1}{\delta}))^{1/6} \log^{1/3}(NT) \log^{1/3}(N)}{T^{1/3} K^{1/2} \varepsilon^{1/3}}$ and $\tau = \frac{(N \log(\frac{1}{\delta}))^{1/3} \log^{2/3}(NT) T^{1/3}}{\varepsilon^{2/3} \log^{1/3}(N)}$ gives

$$\mathrm{R}(T, K, N) \leq \frac{95(N \log(\frac{1}{\delta}))^{1/6} K^{1/2} \log^{1/3}(NT) \log^{1/3}(N) T^{2/3}}{\varepsilon^{1/3}}$$
$$+ \frac{(95N \log(\frac{1}{\delta}))^{1/3} K^{1/2} \log^{2/3}(NT) \log^{2/3}(N) T^{1/3}}{\varepsilon^{2/3}}$$
$$\leq \frac{100N^{1/6} K^{1/2} T^{2/3} \log^{1/6}(\frac{1}{\delta}) \log^{1/3}(NT) \log^{1/3}(N)}{\varepsilon^{1/3}}$$
$$+ \frac{N^{1/2} \log(\frac{1}{\delta})^{1/2} \log(NT) \log(N)}{\varepsilon}.$$

which completes the proof. ∎

*Proof.* (of Privacy in Theorem 4.5) Fix $\varepsilon, \delta \in (0, 1]$. Note that the sequence of actions played by Algorithm 2 are completely determined by the noisy loss vectors $\tilde{\ell}'_1, \ldots, \tilde{\ell}'_{\frac{T}{\tau}}$. Thus, by post-processing it suffices to show that these vectors are output in a $\varepsilon$-differentially private manner. From this perspective, Algorithm 2 can be viewed as the adaptive composition $M$ of the sequence of mechanisms $M_1, \ldots, M_{\lfloor \frac{T}{\tau} \rfloor}$, where $M_1 : ([K] \times \Pi([K]))^\tau \times \ell_{1:T} \to \mathbb{R}^N$ is defined as

$$M_1(I_{1:\tau}, \mu_{1:T}^{1:\tau}, \ell_{1:T}) = (\tilde{\ell}'_1(1), \ldots, \tilde{\ell}'_1(N))$$

for $\tilde{\ell}'_1(j)$ defined as in Line 10 of Algorithm 2. Likewise, for $s \in \{2, \ldots, \lfloor \frac{T}{\tau} \rfloor\}$, define $M_s : (\mathbb{R}^N)^{s-1} \times (\Pi([K]) \times [K])^\tau \times \ell_{1:T} \to \mathbb{R}^N$ such that

$$M_s(\tilde{\ell}'_{1:s-1}, \mu^{1:N}_{(s-1)\tau+1:s\tau}, I_{(s-1)\tau+1:s\tau}, \ell_{1:T}) = (\tilde{\ell}'_{s\tau}(1), \ldots, \tilde{\ell}'_{s\tau}(N)).$$

We will prove that $M(\ell_{1:T})$ and $M(\ell'_{1:T})$ are $(\epsilon, \delta)$-indistinguishable. To do so, fix two neighboring data sets $\ell_{1:T}$ and $\ell'_{1:T}$. Let $t'$ be the index where the two datasets differ. Let $r' \in \{1, \ldots, \lfloor \frac{T}{\tau} \rfloor\}$ be the batch in where $t'$ lies. Fix a sequence of outcomes $x_{1:\frac{T}{\tau}} \in (\mathbb{R}^N)^{\frac{T}{\tau}}$, $\mu^{1:N}_{1:T} \in \Pi([K])^{NT}$, and $I_{1:T} \in [K]^T$.

For all $r < r'$, we have that the random variables $M_r(x_{1:r-1}, \mu^{1:N}_{(r-1)\tau+1:r\tau}, I_{(r-1)\tau+1:r\tau}, \ell_{1:T})$ and $M_r(x_{1:r-1}, \mu^{1:N}_{(r-1)\tau+1:r\tau}, I_{(r-1)\tau+1:r\tau}, \ell'_{1:T})$ are 0-indistinguishable. We now show that $M_{r'}(x_{1:r'-1}, \mu^{1:N}_{(r'-1)\tau+1:r'\tau}, I_{(r'-1)\tau+1:r'\tau}, \ell_{1:T})$ and $M_{r'}(x_{1:r'-1}, \mu^{1:N}_{(r'-1)\tau+1:r'\tau}, I_{(r'-1)\tau+1:r'\tau}, \ell'_{1:T})$ are $(\varepsilon, \delta)$-indistinguishable. On input $x_{1:r'-1}, \mu^{1:N}_{(r'-1)\tau+1:r'\tau}$, and $I_{(r'-1)\tau+1:r'\tau}$, the mechanism $M_{r'}(\cdot, \ell_{1:T})$ computes $\tilde{\ell}'_{r'\tau}(j) = \tilde{\ell}_{r'\tau}(j) + Z^j_{r'}$, where $Z^j_{r'} \sim \text{Lap}\left(0, \frac{3K\sqrt{N\log(\frac{1}{\delta})}}{\gamma\tau\varepsilon}\right)$ and

$$\tilde{\ell}_{r'\tau}(j) = \frac{1}{\tau} \sum_{m=(r'-1)\tau+1}^{r'\tau} \sum_{i=1}^{K} \frac{\mu^j_m(i)\ell_m(i)\mathbb{I}\{I_m = i\}}{Q_m(i)},$$

for every $j \in [N]$. Note that $x_{1:r'-1}$ complete determines $Q_m(i)$. Moreover, the global sensitivity of $\tilde{\ell}_{r'\tau}(j)$, with respect to neighboring datasets, is at most $\frac{K}{\gamma\tau}$ since $Q_t(i) \geq \frac{\gamma}{K}$ for all $t \in [T]$. Accordingly, by the Laplace Mechanism and advanced composition, we have that the outputs of $M_{r'}(\cdot, \ell_{1:T})$ and $M_{r'}(\cdot, \ell'_{1:T})$ are $(\varepsilon, \delta)$-indistinguishable.

To complete the proof, it suffices to show that for all $r \geq r' + 1$, we have that the outputs of, $M_r(\cdot, \ell_{1:T})$ and $M_r(\cdot, \ell'_{1:T})$ are 0-indistinguishable. However, this follows from the fact that for every $r \geq r' + 1$, we have that $\ell_{(r-1)\tau+1:r\tau+1} = \ell'_{(r-1)\tau+1:r\tau}$ and that mechanism $M_r$ does not access the true data $\ell_{1:(r-1)\tau}$, but only the privatized, published outputs of the previous mechanisms $M_1, \ldots, M_{r-1}$. Thus, by advanced composition, we have that the entire mechanism $M$ is $(\varepsilon, \delta)$-differentially private. ∎

## G. Barriers to Private Adversarial Bandits

### G.1. Privacy leakage in EXP3

To better understand its per-round privacy loss, it is helpful to view EXP3 as the adaptive composition of $T - 1$ mechanisms $M_2, \ldots, M_T$ where $M_t : [K]^{t-1} \times \ell_{1:T} \to [K]$. For every $t \in \{2, \ldots, T\}$, the mechanism $M_t$, given as input the previously selected actions $I_1, \ldots, I_{t-1}$ and the dataset $\ell_{1:T}$, computes the distribution

$$P_t(i) = (1 - \gamma)\frac{w_t(i)}{\sum_{j=1}^{K} w_t(j)} + \frac{\gamma}{K}$$

where $w_t(j) = \exp\{-\eta \sum_{s=1}^{t-1} \hat{\ell}_s(j)\}$ and $\hat{\ell}_s(j) = \frac{\ell_s(j)\mathbb{I}\{I_s=j\}}{P_s(j)}$. Then, $M_t$ samples an action $I_t \sim P_t$. The mechanism $M_t$ is $\varepsilon_t$-differentially private if for any pair of neighboring data sets $\ell_{1:T}$ and $\ell'_{1:T}$, we have that

$$\sup_{I_1,\ldots,I_{t-1}\in[K]^{t-1}} \sup_{i\in[K]} \frac{\mathbb{P}[M_t(I_{1:t-1}, \ell_{1:T}) = i]}{\mathbb{P}[M_t(I_{1:t-1}, \ell'_{1:T}) = i]} \leq e^{\varepsilon_t}.$$

Now, consider two neighboring datasets $\ell_{1:T}$ and $\ell'_{1:T}$ that differ at the first time point $t = 1$. Let $P_1, \ldots, P_T$ denote the sequence of probabilities output by the mechanisms when run on $\ell_{1:T}$ and let $P'_1, \ldots, P'_T$ denote the same for $\ell'_{1:T}$. Since $\ell_1 \neq \ell'_1$, we have that $\hat{\ell}_1 \neq \hat{\ell}'_1$. Accordingly, $P_2 \neq P'_2$. The key insight now is that because $P_2 \neq P'_2$, we have that $\hat{\ell}_2 \neq \hat{\ell}'_2$, and so $P_3 \neq P'_3$. Continuing this process gives that $P_t \neq P'_t$ and $\hat{\ell}_t \neq \hat{\ell}'_t$ for all $t \geq 2$. Unfortunately, this difference in probabilities can cause the privacy loss to grow with $t$. To get some intuition, fix some $t \geq 2$ and sequence $I_1, \ldots, I_{t-1} \in [K]^{t-1}$. Consider the ratio

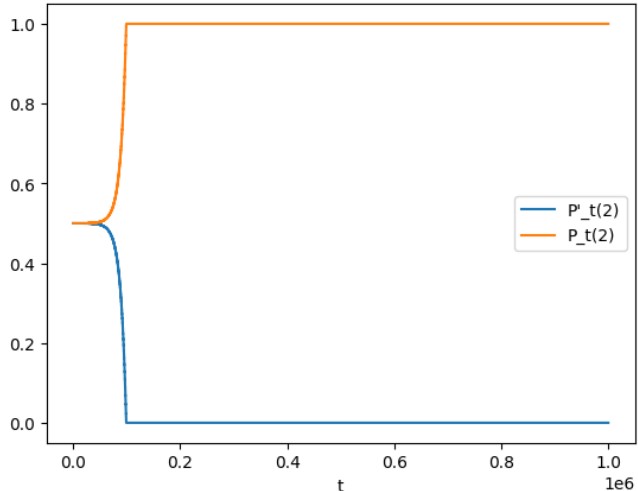

*Figure 1.* Probabilities on action 2 assigned by EXP3 when run with $\gamma = \eta = 0.0001$, and $T = 100 \cdot \frac{1}{\eta}$ on datasets $\ell_{1:T}$ and $\ell'_{1:T}$.

$$\sup_{i \in [K]} \frac{P_t(i)}{P'_t(i)} \approx \sup_{i \in [K]} \frac{w_t(i)}{w'_t(i)} \frac{\sum_{j=1}^{K} w'_t(j)}{\sum_{j=1}^{K} w_t(j)} \approx \sup_{i \in [K]} \frac{w_t(i)}{w'_t(i)}.$$

Observe that

$$\sup_{i \in [K]} \frac{w_t(i)}{w'_t(i)} = \sup_{i \in [K]} \exp\left\{\eta \sum_{s=1}^{t-1} \hat{\ell}'_s(i) - \hat{\ell}_s(i)\right\}$$

$$= \sup_{i \in [K]} \exp\left\{\eta \frac{(\ell'_1(i) - \ell_1(i))\mathbb{I}\{I_1 = i\}}{P_1(i)} + \eta \sum_{s=2}^{t-1} \ell_s(i)\mathbb{I}\{I_s = i\}\left(\frac{1}{P'_s(i)} - \frac{1}{P_s(i)}\right)\right\}.$$

Since $P'_s(i) \neq P_s(i)$ for every $s \leq t - 1$, we can pick two neighboring sequences of losses and a sequence of actions $I_1, \ldots, I_T$ such that $\sup_{i \in [K]} \frac{w_s(i)}{w'_s(i)}$ grows very quickly with $s$. For example, the following choices for neighboring datasets and sequences of actions will do. Let $K = 2$ and pick $\ell_{1:T}$ such that $\ell_1(1) = 1$, $\ell_1(2) = 0$, and $\ell_t(1) = \ell_t(2) = 1$ for all $t \in \{2, \ldots, T\}$. Pick neighboring dataset $\ell'_{1:T}$ such that $\ell'_t(1) = \ell'_t(2) = 1$, for all $t \in [T]$. Finally, consider the sequence of actions $I_1, \ldots, I_T$ such that $I_t = 2$ if $t$ is odd and $I_t = 1$ if $t$ is even. That is, the sequence of actions $I_1, \ldots, I_T$ alternates between 2 and 1, starting with action 2.

We claim that for this choice of neighboring datasets and sequences of actions, $P_t(2)$ and $P'_t(2)$ diverge rapidly with $P_t(2)$ approaching $1 - \frac{\gamma}{2}$ and $P'_t(2)$ approaching $\frac{\gamma}{2}$. To see why, fix $t \geq 2$ and suppose that $I_t = 1$. Then, by definition, for the loss sequence $\ell_{1:T}$, we have that $w_{t+2}(1) = w_{t+1}(1) = w_t(1) \exp \frac{-\eta}{P_t(1)}$ and $w_{t+2}(2) = w_t(2) \exp \frac{-\eta}{1 - P_{t+1}(1)}$. Since $I_t = 1$, we know that $P_{t+1}(1) < P_t(1)$ and therefore $w_{t+2}(2) > w_t(2) \exp \frac{-\eta}{1 - P_t(1)}$. Now, if $P_t(1) < \frac{1}{2}$, then $w_{t+2}(1)/w_{t+2}(2) < w_t(1)/w_t(2) < 1$. Accordingly, $P_{t+2}(1) < 1/2$, and repeating the analysis would eventually show that $w_t(1)/w_t(2) \to 0$, implying that $P_t(2) \to 1 - \frac{\gamma}{2}$. A symmetric argument shows that if $P_t(1) > 1/2$, then $w_t(2)/w_t(1) \to 0$, and therefore $P_t(2) \to \frac{\gamma}{2}$. Since the two loss sequences $\ell_{1:T}$ and $\ell'_{1:T}$ are identical after time point $t = 2$, an identical argument holds for $\ell'_{1:T}$. To complete the proof sketch, note that for loss sequence $\ell_{1:T}$, $I_4 = 1$ and $P_4(1) < \frac{1}{2}$, thus $P_t(2) \to 1 - \frac{\gamma}{2}$. On the other hand, for loss sequence $\ell'_{1:T}$, $I_2 = 1$ and $P'_2(1) > \frac{1}{2}$, giving that $P'_t(2) \to \frac{\gamma}{2}$.

We also verify this claim empirically in Figure 1, which gives a better sense of the rate of divergence between $P_t(2)$ and $P'_t(2)$. The code generating the figure above is provided below.

```python
import numpy as np
import matplotlib.pyplot as plt

eta = 0.0001
T = 100 * int(1/eta)
gamma = eta

# Execute EXP3 on loss sequence l_1, \dots, l_T
w_1 = 1
w_2 = 1
P_2 = 0
P_2_hist = []

for t in range(T):
    Q_2 = (w_2/(w_2 + w_1)) #unmixed prob.
    P_2 = (1-gamma) * Q_2  + gamma/2 #mixed prob.
    P_2_hist.append(P_2)
    if t == 0:
        w_2 = w_2 * np.exp(0*eta/(P_2))
    elif t % 2 == 0:
        w_2 = w_2 * np.exp(-1*eta/(P_2)) #pull action 2 in even rounds
    else:
        w_1 = w_1 * np.exp(-1*eta/((1-P_2))) #pull action 1 in odd rounds

plt.plot(P_2_hist, label= "P_t(2)")

# Execute EXP3 on loss sequence l'_1, \dots, l'_T
w_1 = 1
w_2 = 1
P_2 = 0
P_2_hist = []

for t in range(T):
    Q_2 = (w_2/(w_2 + w_1))
    P_2 = (1-gamma) * Q_2  + gamma/2
    P_2_hist.append(P_2)
    if t % 2 == 0:
        w_2 = w_2 * np.exp(-1*eta/(P_2)) #pull action 2 in even rounds
    else:
        w_1 = w_1 * np.exp(-1*eta/((1-P_2))) #pull action 1 in odd rounds

plt.plot(P_2_hist, label= "P'_t(2)")

plt.xlabel("t")
plt.legend()
plt.show()
```

We note that the authors of (Tossou & Dimitrakakis, 2017) acknowledge that this issue was overlooked when stating Theorem 3.3 in (Tossou & Dimitrakakis, 2017). Therefore, we are unable to verify the Theorem 3.3. Unfortunately, (Tossou & Dimitrakakis, 2017) use Theorem 3.3 in the proof of Corollary 3.3, which claims to give a private adversarial bandit algorithm with expected regret $\tilde{O}\left(\frac{T^{2/3}\sqrt{K\ln(K)}}{\varepsilon^{1/3}}\right)$, ignoring log factors in $\frac{1}{\delta}$. Thus, we are unable to verify whether Corollary 3.3 is correct.

### G.2. Proof of Lemma 5.1

*Proof.* (of Lemma 5.1) Let $\mathcal{A}$ be any $\varepsilon$-differentially private algorithm (for $\varepsilon \leq 1$) that satisfies condition (1) and (2) with parameters $\gamma \in [0, \frac{1}{2}]$, $\tau < \frac{T}{2}$ and $2\gamma\tau \leq p \leq \gamma(T-\tau)$. Consider the alternate sequence of loss functions $\ell'_1, \ldots, \ell'_T$ such that $\ell'_{1:\tau} = \ell_{1:\tau}$ but $\ell'_{\tau+1:T}$ is such that $\ell'_t(2) = 0$ and $\ell'_t(1) = \frac{1}{2}$ for all $t \in \{\tau+1, \ldots, T\}$.

It suffices to show that

$$\mathbb{P}(I'_1, \ldots, I'_T \notin E^p_{\gamma,\tau}) \leq e^{\varepsilon p} \cdot \mathbb{P}(I_1, \ldots, I_T \notin E^p_{\gamma,\tau}) \leq \frac{1}{2} e^{\varepsilon p} \tag{2}$$

where $I_{1:T}$ and $I'_{1:T}$ are the random variables denoting the selected actions of $\mathcal{A}$ when run on $\ell_{1:T}$ and $\ell'_{1:T}$ respectively. Indeed, when $I'_1, \ldots, I'_T \in E^p_{\gamma,\tau}$, we have that the regret of $\mathcal{A}$ when run on $\ell'_{1:T}$ is at least $\frac{p}{2\gamma} - \frac{p}{2} - \frac{\tau}{2}$. On the other hand, if $I_{1:T} \in E_\gamma$, we have that the regret of $\mathcal{A}$ on $\ell_{1:T}$ is at least $\frac{\gamma}{2}T$. So with probability $\frac{1}{2}$, the regret of $\mathcal{A}$ on $\ell_{1:T}$ is $\frac{\gamma}{2}T$ and with probability at least $1 - \frac{1}{2}e^{\varepsilon p}$, the regret of $\mathcal{A}$ on $\ell'_{1:T}$ is at least $\frac{p}{2\gamma} - \frac{p}{2} - \frac{\tau}{2} \geq \frac{p}{4\gamma} - \frac{\tau}{2}$, where the inequality follows from the fact that $\gamma \leq \frac{1}{2}$. Therefore, the worst-case *expected* regret is at least

$$\max\left\{ \frac{1}{2} \cdot \frac{\gamma T}{2}, (1 - \frac{1}{2}e^{\varepsilon p})\left(\frac{p}{4\gamma} - \frac{\tau}{2}\right) - \left(\frac{1}{2}e^{\varepsilon p}\right)\frac{\tau}{2} \right\} \geq \max\left\{ \frac{\gamma T}{4}, \left(1 - \frac{1}{2}e^{\varepsilon p}\right)\frac{p}{4\gamma} - \frac{\tau}{2} \right\}.$$

To prove Equation 2, recall that we may write any randomized algorithm $\mathcal{A}$ as a deterministic function of an input $x$ and an infinite sequence of bits $b_1, b_2, \ldots$ generated uniformly at random. From this perspective, we can think of a randomized bandit algorithm $\mathcal{A}$ as a deterministic mapping from a sequence of losses $\ell_{1:T}$ and an infinite sequence of bits $b \in \{0, 1\}^{\mathbb{N}}$ to a sequence of $T$ actions. That is,

$$\mathcal{A} : \{0, 1\}^{\mathbb{N}} \times \left([0, 1]^K\right)^T \to [K]^T.$$

Using this perspective, Equation 2 is equivalent to showing that:

$$\mathbb{P}_{b \sim \{0,1\}^{\mathbb{N}}}(\mathcal{A}(b, \ell'_{1:T}) \notin E^p_{\gamma,\tau}) \leq e^{\varepsilon p} \mathbb{P}_{b \sim \{0,1\}^{\mathbb{N}}}(\mathcal{A}(b, \ell_{1:T}) \notin E^p_{\gamma,\tau}).$$

Consider the following sequence of losses parameterized by $S \subset \{\tau + 1, \ldots, T\}, |S| \leq p$:

$$\ell^S_t(i) = \begin{cases} 1/2, & \text{if } i = 1 \\ 0, & \text{if } i = 2 \text{ and } t \in S \\ 1, & i = 2 \text{ and } t \notin S \end{cases}$$

Let $\mathcal{L} := \{\ell^S_{1:T} : S \subset \{\tau + 1, \ldots, T\}, S \leq p \}$ be the collection of all such sequences of loss functions. Note that every $\ell^S_{1:T} \in \mathcal{L}$ differs from $\ell_{1:T}$ only at time points $t \in S$. Thus, by group privacy (see Lemma A.8), we have that

$$\sup_{\ell^S_{1:T} \in \mathcal{L}} \mathbb{P}_{b \sim \{0,1\}^{\mathbb{N}}}(\mathcal{A}(b, \ell^S_{1:T}) \notin E^p_{\gamma,\tau}) \leq e^{\varepsilon p} \mathbb{P}_{b \sim \{0,1\}^{\mathbb{N}}}(\mathcal{A}(b, \ell_{1:T}) \notin E^p_{\gamma,\tau}).$$

Now, fix the sequence of random bits $b \in \{0, 1\}^{\mathbb{N}}$. Let $i'_{1:T} = \mathcal{A}(b, \ell'_{1:T})$. Define $S' := \{t \geq \tau + 1 : i'_t = 2\}$ and $S'_{\leq p}$ be the first $p$ such time points. Let $i^{S'_{\leq p}}_{1:T} = \mathcal{A}(b, \ell^{S'_{\leq p}}_{1:T})$. Let $t' = \max\{t \geq \tau + 1 : \sum_{s=\tau+1}^t \mathbb{I}\{i'_s = 2\} \leq p\}$ and $t^{S'_{\leq p}} = \max\{t \geq \tau + 1 : \sum_{s=\tau+1}^t \mathbb{I}\{i^{S'_{\leq p}}_s = 2\} \leq p\}$. Because bandit algorithms only observe the losses of the selected action, we have that $t' = t^{S'_{\leq p}}$. In addition, we have that $i'_{1:T} \in E^p_{\gamma,\tau}$ if and only if $t' \geq \tau + \frac{p}{\gamma}$, and likewise for $i^{S'_{\leq p}}_{1:T}$. Therefore,

$$\mathbb{I}\{i'_{1:T} \in E^p_{\gamma,\tau}\} = \mathbb{I}\{i^{S'_{\leq p}}_{1:T} \in E^p_{\gamma,\tau}\}$$

and therefore

$$\mathbb{I}\{i'_{1:T} \notin E^p_{\gamma,\tau}\} = \mathbb{I}\{i^{S'_{\leq p}}_{1:T} \notin E^p_{\gamma,\tau}\}.$$

Taking expectation on both sides with respect to $b \sim \{0, 1\}^{\mathbb{N}}$, gives that

$$
\begin{aligned}
\mathbb{P}_{b\sim\{0,1\}^{\mathbb{N}}}(\mathcal{A}(b,\ell'_{1:T}) \notin E^p_{\gamma,\tau}) &= \mathbb{P}_{b\sim\{0,1\}^{\mathbb{N}}}(\mathcal{A}(b,\ell^{S'_{\leq p}}_{1:T}) \notin E^p_{\gamma,\tau}) \\
&\leq \sup_{\ell^S_{1:T}\in\mathcal{L}} \mathbb{P}_{b\sim\{0,1\}^{\mathbb{N}}}(\mathcal{A}(b,\ell^S_{1:T}) \notin E^p_{\gamma,\tau}) \\
&\leq e^{\varepsilon p}\mathbb{P}_{b\sim\{0,1\}^{\mathbb{N}}}(\mathcal{A}(b,\ell_{1:T}) \notin E^p_{\gamma,\tau}),
\end{aligned}
$$

completing the proof. ∎

