# OpenReview forum: "Faster Rates for Private Adversarial Bandits"
_ICML.cc/2025/Conference — ICML 2025 poster_

### Official Review · Reviewer_faFK · 2025-03-13

**Overall Recommendation:** 3

**Summary:**

This work studies the adversarial bandit problem where the reward function can vary across stages and considers a differential privacy guarantee. The author proposes a novel algorithm within a batched learning framework, adding Laplace noise to the average reward in each batch to ensure differential privacy. Compared with previous work, this approach achieves a better dependency on the differential privacy parameter $\epsilon$ while still maintaining a $\sqrt{T}$ regret.

**Claims And Evidence:**

The author provides a clear claim of the result in the theorems and includes a proof sketch to outline the key steps of the theoretical analysis.

**Essential References Not Discussed:**

This paper provides a comprehensive discussion of related work in adversarial bandits and private bandits.

**Experimental Designs Or Analyses:**

The main contribution of this work focuses on the theoretical analysis of regret, with the simulation serving as an auxiliary tool to support its efficiency, which is reasonable and sufficient. Therefore, there is no need to evaluate the soundness or validity of the experimental designs or analyses.

**Methods And Evaluation Criteria:**

The main contribution of this work focuses on the theoretical analysis of regret, with the simulation serving as an auxiliary tool to support its efficiency, which is reasonable and sufficient.

**Other Comments Or Suggestions:**

The definition of differential privacy should be refined. For instance, in Theorem 4.5, the algorithm provides a $(\epsilon, \delta)$ privacy guarantee, but the earlier definition only involves $\epsilon$, making it unclear what $\delta$ represents. Additionally, in lines 236-237, there is a contradiction between the stated upper bound and the previously established lower bound. This discrepancy appears to stem from different definitions of privacy (local vs. non-local differential privacy). However, the paper does not clearly define local differential privacy or explain what specifically accounts for the gap in the regret bound. Clarifying these aspects would strengthen the theoretical consistency of the work.

**Other Strengths And Weaknesses:**

Compared with previous works in the adversarial bandit problem, this work relies on a stronger assumption of an oblivious adversary, where all reward functions are chosen at the beginning and cannot adapt based on the agent's previous actions. Therefore, it is not entirely fair to directly compare the results with prior work, as the improvement may stem from this assumption. It would be beneficial to explicitly highlight this difference in the introduction and include it in the comparison table.

**Questions For Authors:**

The results for Bandits with Experts are somewhat confusing, and several questions arise:

1. While the Generic Conversion method is well-motivated, it is unclear which of the three different results for Bandits with Experts is the best, and what specific advantages the other algorithms provide. A clearer comparison of these results would be helpful.

2. Based on the proof sketch, the proposed method appears to be a general framework applicable to any algorithm. However, it is unclear why this method fails to achieve a $1/\sqrt{\epsilon}$ dependency in the second and third regret guarantees. A more detailed explanation of this limitation would be beneficial.

3. Regarding the Adversarial Bandit result and the first result for Bandits with Experts, the proof sketch suggests that the only stated improvement concerns $\epsilon$. However, it remains unclear whether the batch learning framework with a batch size of $1/\epsilon$ can effectively remove the logarithmic dependency on $T$ compared to previous work. Clarifying this aspect would improve the theoretical discussion.

**Relation To Broader Scientific Literature:**

This work mainly focuses on providing a differential privacy guarantee for the adversarial bandit problem, which may be related to other works on bandit analysis.

**Theoretical Claims:**

The author provides a clear proof sketch outlining the key steps of the theoretical analysis. The critical step involves transitioning from the Non-Private to the Private setting. With the batched learning framework, it becomes reasonable to reduce the magnitude of the Laplace noise added to the average reward function, saving a factor of $\epsilon$. On the other hand, the batched updating in an adversarial environment introduces an additional dependency of $\sqrt{\epsilon}$. Therefore, the improvement in the dependency on $\epsilon$ appears reasonable.

However, a concern regarding correctness arises with the logarithmic dependency on the number of rounds $T$. Based on the proof sketch, the only stated improvement concerns $\epsilon$, and it remains unclear whether the batch learning framework with a batch size of $1/\epsilon$ can effectively remove the logarithmic dependency on $T$ compared to previous work.

---

> ### Author Rebuttal · Authors · 2025-03-30
>
> We thank the reviewer for their comments. We address the reviewer's main concern below and hope that they will reevaluate their score accordingly.
>
> > Therefore, it is not entirely fair to directly compare the results with prior work, as the improvement may stem from this assumption.
>
> While adaptive adversaries are well-studied in non-private bandit literature, oblivious adversaries have been the main focus in the private bandit literature. In particular, there are only two previous works that study private adversarial bandits, namely Tossou \& Dimitrakakis (2017) and  Agarwal \& Singh (2017). In the last two paragraphs of Section 2.1,  Tossou \& Dimitrakakis (2017) make explicit that they first consider an oblivious adversary. They then note that their guarantees can be apply to an $m$-bounded adaptive adversary. As for Agarwal \& Singh (2017), Theorem 4.1 makes it clear that they only consider an oblivious adversary as the sequence of loss vectors is fixed before the game begins. Nevertheless, if one considers an adaptive adversary, then the lower bound from Asi et al. (2023) shows that sublinear regret is not possible under pure differential privacy even for a constant $\epsilon$. Moreover, for approximate differential privacy, the lower bounds from Asi et al. (2023) show that sub linear regret is not possible if $\epsilon \leq \frac{1}{\sqrt{T}}.$ Thus, the adaptive adversary case is not as interesting because existing algorithms are already optimal (up to log factors).
>
> > The definition of differential privacy should be refined...
>
> Our definitions of differential privacy (see Definitions 2.3 and 2.4) do include $\delta$, and this is the standard $\delta$ which people use to capture failure probability.
>
> > Additionally, in lines 236-237, there is a contradiction between the stated upper bound and the previously established lower bound.
>
> Our algorithm obtaining the upper bound in Corollary 3.2  satisfies the notion of central differential privacy. Therefore, the lower bound from Basu et al. (2019) does **not** apply here as it hold for the more strict notion of local differential privacy. We will clarify this in the paper. We will also make sure to include an explicit definition of local differential privacy in the camera-ready version.
>
> > (Q1) While the Generic Conversion method is well-motivated, it is unclear which of the three different results for Bandits with Experts is the best, and what specific advantages the other algorithms provide.
>
> None of the regret guarantees of our three algorithms strictly dominate the other. Below, we highlight three regimes and the corresponding algorithm that obtains the best regret bound in that regime. We will make sure to include this detailed comparison in the final version.
>
> **Low-dimension and High Privacy ($N \leq K$):** In this regime, our upper bound $O(\frac{\sqrt{NT}}{\sqrt{\epsilon}})$ is superior.
>
> **High-dimension and Low Privacy ($N \geq K$ and $\epsilon \geq \frac{K}{N}$):** In this regime, our second upper bound $O(\frac{\sqrt{KT \log N} \log(KT)}{\epsilon})$ is superior.
>
> **High-dimension and High Privacy ($N \geq K$ and $\epsilon \leq \frac{1}{\sqrt{T}}$):** In this regime, our third upper bound $O(\frac{N^{1/6}K^{1/2}T^{2/3}\log(NT)}{\epsilon^{1/3}} + \frac{N^{1/2}\log(NT)}{\epsilon})$ is superior.
>
> These are all regimes that are interesting in practice, so its important to get the best rates for each one of these regimes.
>
> > (Q2) However, it is unclear why this method fails to achieve a $1/\sqrt{\epsilon}$ dependency in the second and third regret guarantees.
>
> The second guarantee (Theorem 4.4) actually does not use the batching method. Instead, it simply adds Laplace noise to each loss vector. Thus, one should not expect to get a $\frac{1}{\sqrt{\epsilon}}$ here, as this approach actually gives a stronger, local DP guarantee. We will make this more explicit in the camera ready version. As for the third guarantee, this is more subtle and occurs because more noise needs to be added to privatize the batched expert losses. We will include a more complete discussion in the camera-ready version.
>
> > (Q3) However, it remains unclear whether the batch learning framework with a batch size of  can effectively remove the logarithmic dependency on  compared to previous work...
>
> Our technique allows us to improve upon existing regret bounds in terms of both $\epsilon$ and $T$. As the reviewer noted, our batching with noise technique allows to improve the dependence on $\epsilon$ from $1/\epsilon$ to $1/\sqrt{\epsilon}.$ However, our technique also allows us to *expand* the set of bandit algorithms that we can privatize beyond EXP3. In particular, we are now able to privatize *heavy-tailed* bandit algorithms which do not have $\log{T}$ terms in their regret bounds, one of which is the HTINF algorithm from Huang et al. (2022). Because our reduction carries over the regret bound from the base bandit algorithm, we are able to remove a log factor in $T.$

---

> > ### Comment · Reviewer_faFK · 2025-04-03
> >
> > Thanks for the rebuttal and it addresses my concern regarding the definitions of local differential privacy and centralized differential privacy. I will maintain my positive score.

---

### Official Review · Reviewer_NE5W · 2025-03-13

**Overall Recommendation:** 3

**Summary:**

The paper presents novel differentially private algorithms for adversarial bandits and bandits with expert advice. The primary contribution is an efficient conversion method that transforms any non-private bandit algorithm into a differentially private one, leading to improved regret bounds. The proposed algorithms achieve regret upper bounds of O(\sqrt{KT/ϵ}) (the sota had an additional log term) for adversarial bandits, which improves on existing results. Additionally, for bandits with expert advice, the paper introduces the first differentially private algorithms with various regret bounds that cater to different settings of actions, experts, and privacy parameters.

## update after rebuttal"
I have read the discussions and I keep my assessment unchanged

**Claims And Evidence:**

The paper claims an improvement over the previous best regret bound. This improvement is significant because it ensures sublinear regret even when ϵ≤\sqrt{T}​, which was not previously achieved. The claim is supported by mathematical proofs demonstrating regret upper bounds under different adversarial conditions.
The study establishes a fundamental separation between central and local differential privacy in adversarial bandits, proving that sublinear regret is possible for ϵ∈o(1/\sqrt{T}) under central differential privacy, whereas this is not achievable under local differential privacy.
Three different private algorithms are introduced, achieving different regret bounds with various techniques.

**Essential References Not Discussed:**

The interest of the contribution is justified by the application of bandits to "online advertising, medical trials, and recommendation systems" but the relevance of adversarial bandits in these settings would deserve references.

**Experimental Designs Or Analyses:**

nothing here

**Methods And Evaluation Criteria:**

The paper primarily employs theoretical analysis to evaluate the effectiveness of the proposed algorithms. No numerical experiments are reported.

**Other Comments Or Suggestions:**

It would be useful to contrast the results with private stochastic bandits to understand trade-offs.

**Other Strengths And Weaknesses:**

+ The proposed framework (Alg 1) applies to any non-private bandit algorithm, it is very simple and natural.
+ The introduction of batching and noise addition significantly improves private bandit performance.
- No experimental results are presented to support theoretical findings.
- The paper focuses purely on theoretical settings without discussing real-world implementations.

**Questions For Authors:**

How do the proposed algorithm compare experimentally to the SOTA?
Could alternative privacy mechanisms improve regret rates?
How does the proposed approach generalize to contextual bandits?

**Relation To Broader Scientific Literature:**

The paper builds upon and extends several foundational works in adversarial bandits and differential privacy, with no remarkably original mechanism or proof scheme.

**Theoretical Claims:**

Generic Conversion Framework:
    The paper establishes a (very simple) framework to convert any non-private bandit algorithm into a private one while maintaining sublinear regret.
    The core technique involves batching rounds and adding Laplace noise to the observed losses.

Lower Bounds and Limits:
    The authors prove that it is difficult to achieve an additive separation between ϵϵ and TT in the adversarial bandit setting, unlike in full-information settings.
    They also show that standard EXP3-based approaches are challenging to privatize effectively due to compounding privacy loss over multiple rounds.

---

> ### Author Rebuttal · Authors · 2025-03-30
>
> We thank the reviewers for their comments. We address their concerns below and hope they will reevaluate their score accordingly.
>
> > No experiments.
>
> We acknowledge the concern regarding the lack of experiments. However, our work is intentionally theoretical, aimed at understanding fundamental rates and limits of private adversarial bandits. Theoretical contributions often provide key insights that guide future empirical research. While experimental validation is certainly valuable, we believe that our results are meaningful in their own right and align with the norms of theoretical research in this area.
>
> > (Q1) How do the proposed algorithm compare experimentally to the SOTA?
>
> Our paper is mainly theoretical in nature. That said, we agree with the reviewer that this is an important direction of future work.
>
> > Could alternative privacy mechanisms improve regret rates?
>
> This is a great question. We believe that one way to improve our regret bounds is through the use of the Binary Tree Mechanism, which has been used to obtain private regret guarantees in the full-information setting. However, our current attempts at doing have been unsuccessful. In particular, some subtleties arise in the bandit setting as one needs to deal with unbiased estimates of the true loss, whose sensitivity can be massive. We will make sure to discuss this in the camera-ready version.
>
> > How does the proposed approach generalize to contextual bandits?
>
> The bandits with expert advice setting can be viewed as the contextual version of the traditional adversarial bandit setting [1]. Accordingly, our results in Section 4 show that our techniques do extend to the contextual setting. Table 1 summarizes the exact regret bounds we achieve for the bandits with expert advice setting.
>
> [1] Auer, Peter, et al. "The nonstochastic multiarmed bandit problem." SIAM journal on computing 32.1 (2002): 48-77.

---

### Official Review · Reviewer_w61y · 2025-03-14

**Overall Recommendation:** 4

**Summary:**

This paper presents novel differentially private (DP) adversarial bandit algorithms with improved dependency on the DP parameter $\epsilon$. It improves the regret bound from $O(\sqrt{KT\log KT}/\epsilon)$ to $O(\sqrt{KT}/\epsilon)$, ensuring no-regret even when $\epsilon \sim 1/\sqrt{T}$. Moreover, the paper's construction is general in that it provides a framework for constructing DP adversarial bandit algorithms from a baseline adversarial bandit algorithm with unbounded and negative losses.

**Claims And Evidence:**

This paper is mainly theoretical and its claims are supported by proofs.

**Essential References Not Discussed:**

All essential references are discussed to the best of my knowledge.

**Experimental Designs Or Analyses:**

N/A

**Methods And Evaluation Criteria:**

N/A

**Other Comments Or Suggestions:**

In **Line 1850**,  Shouldn't "...$M_t:[K]^{i-1}$...” be “...$M_t:[K]^{t-1}$...”?

**Other Strengths And Weaknesses:**

**Strengths**

- This paper provides a simple conversion technique from adversarial bandit algorithms with unbounded losses over the reals to a DP adversarial bandit algorithm with improved rates. This could have a broader impact due to its simplicity and its independence from the details of the baseline (host) algorithm.
- They provide a lower bound of $\Omega(\sqrt{T/\epsilon})$ for a class of algorithms, including EXP3 and its batched variants.

**Weakness**

- While they have shown, through experiments and quantitative arguments, that Theorem 3.3 of Tossou & Dimitrakakis (2017) is problematic, it is unclear whether Lemma 5.1 in this paper definitively rules out one of Tossou & Dimitrakakis (2017)'s algorithms claiming a regret of $\widetilde{O}\left(\frac{T^{2/3}\sqrt{K\log K}}{\epsilon^{1/3}}\right)$ (This is also my question: Does Lemma 5.1 rule it out?).

**Questions For Authors:**

Please refer to the strengths & weaknesses section above.

**Relation To Broader Scientific Literature:**

This paper extends the line of research on DP online learning, particularly DP adversarial bandit algorithms. Their results demonstrate that no-regret learning is achievable for any privacy parameter $\epsilon \in \omega(1/T)$ in the oblivious adversary setting under the central DP model. This setting is optimal in the sense that it is known that no-regret learning is impossible for $\epsilon \in o(1/\sqrt{T})$ in either the adaptive adversary setting or the local DP setting. Their improved dependence on $\epsilon$ implies a better trade-off between privacy and utility, advancing the frontier of privacy-preserving algorithms.

**Theoretical Claims:**

I could not find any specific issues with their theorems, particularly their main results, Theorem 3.1 and Theorem 4.2, which provide a general conversion from a class of bandit algorithms to differentially private ones.

---

> ### Author Rebuttal · Authors · 2025-03-30
>
> We thank the reviewer for their comments and noting that our work advances the frontier of privacy-preserving algorithms and could have a broader impact. We address the reviewer's main concern below and hope they reevaluate their score accordingly.
>
> > Does Lemma 5.1 rule it out?
>
> The reviewer is correct in that Lemma 5.1 does not rule out the possibility that the algorithm from Tossou \& Dimitrakakis (2017) obtains a regret of $O(\frac{T^{2/3} \sqrt{K \log K}}{\epsilon^{1/3}})$ as this upper bound is larger than our lower bound $O(\sqrt{T/\epsilon})$ in Lemma 5.1. Nevertheless, even if one considers the upper bound $O(\frac{T^{2/3} \sqrt{K \log K}}{\epsilon^{1/3}})$, our upper bound  of $O(\sqrt{KT/\epsilon})$ in Corollary 3.2 is strictly better for all $\epsilon \leq 1$.
>
> > In Line 1850, Shouldn't...”?
>
> Yes, this is a typo and the reviewer is correct. We will make sure to fix this in the camera-ready version.

---

> > ### Comment · Reviewer_w61y · 2025-04-09
> >
> > Thank you for the detailed rebuttal. After consideration, I have raised my score. I believe achieving a better privacy-accuracy trade-off through a relatively simpler method is a valuable contribution to the community.

---

### Official Review · Reviewer_124Y · 2025-03-23

**Overall Recommendation:** 2

**Summary:**

This paper studies adversarial bandit problems and bandit problems with expert advice, and introduces a differentially private algorithm that achieves better regret bounds than previous approaches.

**Claims And Evidence:**

The writing is not clear enough. See the weakness part for details.

**Essential References Not Discussed:**

The paper investigated most of the related work.

**Experimental Designs Or Analyses:**

There is no experiment.

**Methods And Evaluation Criteria:**

They make sense.

**Other Comments Or Suggestions:**

See the Questions part for more details.

**Other Strengths And Weaknesses:**

Strengths:
The problem in the paper is interesting. And it is good to give the lower bound.

Weaknesses:
1. The writing is not clear enough. For example, in lines 43-44, "Motivated by this gap", which gap? In lines 71- 73, "it is well known that
this is not the case for local differential privacy." Then what is your contribution? What is the logic of the sentence?
2. There are no experimental results.
3. See the Questions part for more details.

**Questions For Authors:**

1.  The regret of $O(\sqrt{KT\log K}/\epsilon)$ is for local differential privacy model. You consider classical DP in your paper, i.e., central differential privacy. How can you compare your CDP results with LDP since they are different DP models?
2. You give 3 different DP private bandit algorithms and get 3 regret bounds. What are the same points and differences among the three algorithms? What is your motivation to design them?
3. You define Definition 2.3 on history $\{\mathcal{H}_t\}$. What is the exact information you want to protect? Loss function or action? Which part is sensitive information?
4. Why did you choose pure differential privacy, not approximation DP for your work?
5. Compare your results with related work. For example, compare the result of Corollary 3.2 with non-private  HTINF and explain the effect of privacy.
6. There is a gap of the factor of $\sqrt{K}$ between your upper bound and lower bound. Could you please explain it?

**Relation To Broader Scientific Literature:**

Their results improve upon previous regret bounds.

**Theoretical Claims:**

I didn't check the details of the proofs. Most of them seem true.

---

> ### Author Rebuttal · Authors · 2025-03-30
>
> We thank the reviewer for their comments. We address their concerns below and hope they will reevaluate their score accordingly.
>
> > in line 43-44, "Motivated by this gap", which gap?
>
> By this phrase, we are referring to our comment in the previous sentence on lines 38-40: "it was not known how large $\epsilon$ needs to be to obtain sublinear expected worst-case regret." We will clarify this in the final version.
>
> > "it is well known that this is not the case for local differential privacy." then what is your contribution?
>
> In this paper, we study the weaker notion of *central differential privacy*. Our results (Corollary 3.2) shows that under central differential privacy constraints, sub-linear regret is possible even when $\epsilon \leq \frac{1}{\sqrt{T}}.$ This is not the case under the strict notion of *local differential privacy*, where sublinear regret is **not** possible if $\epsilon \leq \frac{1}{\sqrt{T}}.$ Hence, our results establish a separation between central and local differential privacy in terms of when sublinear regret can be achieved.
>
> > No experiments
>
> We acknowledge the reviewer's concern. However, our work is intentionally theoretical, aimed at understanding fundamental rates and limits of private adversarial bandits.
>
> >  (Q1) The regret of $O(\sqrt{KT \log K}/\epsilon)$ is for the local differential privacy model...
>
> Since local differential privacy is stronger than central differentially privacy, any $\epsilon$-locally differentially private algorithm is also an $\epsilon$-centrally differentially private algorithm. Accordingly, comparing our regret bounds against $O(\sqrt{KT \log K}/\epsilon)$ *is still meaningful since this is the only known regret bound for adversarial bandits under central differential privacy*. That is, we don't intend to compare our CDP results with LDP; it just happens to be the case that the previous best known regret bound for adversarial bandits under CDP is $O(\sqrt{KT \log K}/\epsilon)$.
>
> > (Q2) What are the same points and differences among the three algorithms?
>
> For bandits *with expert advice*, we give three different algorithms/regret bounds to cover different combinations of small and large $K, N$ and $\epsilon.$ In particular, amongst the three algorithms we give, none of their regret bounds strictly dominate the other -- there are certain choices of $K, N$ and $\epsilon$, where the regret bound of one algorithm is better than the other. Below, we highlight three regimes and the corresponding algorithm that obtains the best regret bound in that regime. We will include this detailed comparison in the final version.
>
> **Low-dimension and High Privacy ($N \leq K$):** Our upper bound $O(\frac{\sqrt{NT}}{\sqrt{\epsilon}})$ is superior.
>
> **High-dimension and Low Privacy ($N \geq K$ and $\epsilon \geq \frac{K}{N}$):** Our second upper bound $O(\frac{\sqrt{KT \log N} \log(KT)}{\epsilon})$ is superior.
>
> **High-dimension and High Privacy ($N \geq K$ and $\epsilon \leq \frac{1}{\sqrt{T}}$):** Our third upper bound $O(\frac{N^{1/6}K^{1/2}T^{2/3}\log(NT)}{\epsilon^{1/3}} + \frac{N^{1/2}\log(NT)}{\epsilon})$ is superior.
>
> These regimes are all interesting in practice, so it's important to get the best rates for each one.  With regards to the algorithms, two out of the three algorithms, namely those obtaining the guarantees in Corollary 4.3 and Theorem 4.5, use the batching plus noise technique. The last algorithm, the one obtaining the guarantee in Theorem 4.4, does not batch and adds noise to each loss vector.
>
> > (Q3) What is the exact information you want to protect?
>
>  We are taking *only* the sequence of loss functions as the sensitive information. This is made explicit in our definition of differential privacy in Definition 2.4 and is standard in the private bandit literature.
>
> > (Q4) Why did you choose pure differential privacy?
>
> Pure differential privacy is a stronger privacy guarantee and often easier to derive minimax rates for. Surprisingly, and unlike under full-information online learning, we found that even under pure differential privacy, the minimax rates for adversarial bandits were not known. This motivated our study of pure differential privacy.
>
> > (Q5) Compare your results with related work.
>
> We will make sure to include a comparison in the final version. For HTINF, the non-private version achieves a regret bound of $O(\sqrt{TK})$ whereas our private version achieves a regret bound of $O(\frac{\sqrt{TK}}{\sqrt{\epsilon}} + \frac{1}{\epsilon}).$ So, the blow is by a factor of $O(\frac{1}{\sqrt{\epsilon}}).$
>
> > (Q6) There is a gap of the factor of $\sqrt{K}$ between your upper bound and lower bound.
>
> This gap comes from the fact that in our lower bound we only consider a setting with two arms. It should be possible to upgrade of lower bound with a factor of $K$ using the standard strategy for proving lower bounds for adversarial bandits, but we leave this for future work.

---

### Decision · Program_Chairs · 2025-05-01

**Decision:**

Accept (poster)

**Comment:**

This paper presents a **central differentially private (CDP)** framework for the adversarial bandit problem. Its main contribution is a generic conversion method that transforms any non-private adversarial bandit algorithm into one satisfying CDP, achieving a regret bound of $O\left(\frac{\sqrt{K T}}{\sqrt{\varepsilon}}\right)$. Additionally, the work extends this approach to bandits with expert advice, proposing the first DP algorithms for this setting with regret bounds that adapt to the number of actions, experts, and privacy parameters. A key insight of the results is that, unlike **local differential privacy (LDP)**, where sublinear regret is impossible when $\varepsilon \leq \frac{1}{\sqrt{T}}$, the proposed CDP framework maintains sublinear regret even in this regime. This demonstrates a fundamental separation between CDP and LDP in adversarial bandits.

This paper was escalated to the SAC, who also reviewed the paper in detail. Overall, we believe the paper offers a versatile and theoretically sound approach to differentially private adversarial bandit learning, providing a better understanding of privacy-regret trade-offs in such problems.

However, one reviewer pointed out that LDP differs significantly from CDP. Therefore, while the dependence of regret on $\varepsilon$ in this work is tighter than that in previous studies on bandits with LDP, it may not be entirely accurate to claim that the regret in this work directly improves upon the existing upper bound of $O\left(\frac{\sqrt{K T \log (K T)}}{\varepsilon}\right)$, as stated, for example, in lines 19-21 of the abstract in the current version. Thus, the authors are strongly advised to carefully check and clarify all such comparisons for better accuracy and clarity in the camera-ready version of this work.